# Visualization of translation and protein biogenesis at the ER membrane

Max Gemmer[1], Marten L. Chaillet[1], Joyce van Loenhout[1], Rodrigo Cuevas Arenas[1], Dimitrios Vismpas[1], Mariska Gröllers-Mulderij[1], Fujiet A. Koh[2], Pascal Albanese[3,4], Richard A. Scheltema[3,4], Stuart C. Howes[1], Abhay Kotecha[2], Juliette Fedry[1✉] & Friedrich Förster[1✉]

The dynamic ribosome–translocon complex, which resides at the endoplasmic reticulum (ER) membrane, produces a major fraction of the human proteome[1,2]. It governs the synthesis, translocation, membrane insertion, N-glycosylation, folding and disulfide-bond formation of nascent proteins. Although individual components of this machinery have been studied at high resolution in isolation[3–7], insights into their interplay in the native membrane remain limited. Here we use cryo-electron tomography, extensive classification and molecular modelling to capture snapshots of mRNA translation and protein maturation at the ER membrane at molecular resolution. We identify a highly abundant classical pre-translocation intermediate with eukaryotic elongation factor 1a (eEF1a) in an extended conformation, suggesting that eEF1a may remain associated with the ribosome after GTP hydrolysis during proofreading. At the ER membrane, distinct polysomes bind to different ER translocons specialized in the synthesis of proteins with signal peptides or multipass transmembrane proteins with the translocon-associated protein complex (TRAP) present in both. The near-complete atomic model of the most abundant ER translocon variant comprising the protein-conducting channel SEC61, TRAP and the oligosaccharyltransferase complex A (OSTA) reveals specific interactions of TRAP with other translocon components. We observe stoichiometric and sub-stoichiometric cofactors associated with OSTA, which are likely to include protein isomerases. In sum, we visualize ER-bound polysomes with their coordinated downstream machinery.

In mammalian cells, the vast majority of membrane proteins, secreted proteins and soluble proteins of most organelles are synthesized at the ER membrane. A cleavable N-terminal signal peptide emerging from the ribosome targets most secretory pathway proteins to the ER[1,2], where the nascent chain elongation is continued, concomitant with its translocation across or insertion into the ER membrane. During its elongation cycle, the ribosome recruits aminoacyl-tRNAs (aa-tRNAs) matching the mRNA codons in the aminoacyl (A)-site, forms the peptide bond between the amino acid and the nascent chain, and translocates the mRNA–tRNA moiety[3]. The GTP-dependent eEF1a and eEF2 support the required tRNA movements and motions of the ribosomal small subunit (SSU) with respect to the large subunit (LSU).

Ribosomes bind to the dynamic ER translocon complex[2]. Its invariant core module—the heterotrimeric protein-conducting channel SEC61—faces the ribosomal exit tunnel. To facilitate protein transport and to accommodate the signal peptide, SEC61 can switch from closed to open conformations[8,9]. SEC61 associates with distinct cofactors that reflect the requirements of different substrates. The translocon-associated protein complex (TRAP), a hetero-tetrameric transmembrane protein complex supporting the insertion of many signal peptides[10,11], is a near-stoichiometric ER translocon component[12]. Low-resolution studies revealed interactions of TRAP with ribosomal RNA (rRNA) expansion segments and ribosomal subunit protein 38e[13] (RPL38e), but the molecular details remain unresolved given the absence of an atomic model of TRAP. OSTA, which is responsible for co-translational N-glycosylation of substrates, is observed in at least 50% of translocon particles in mammalian cells[14]. Although the structure of OSTA and its specific association with the ribosome and SEC61 have been studied extensively[13], its native interactions, including those with biogenesis cofactors such as ER chaperones remain unknown. In addition to the SEC61–TRAP and SEC61–TRAP–OSTA translocons, a ribosome-bound translocon specialized in the insertion of multipass transmembrane proteins, has recently been isolated and analysed structurally[15,16]. Here we have used cryo-electron tomography (cryo-ET) to visualize the elongating ribosome at the ER membrane and its downstream translocation and biogenesis machinery.

[1]Structural Biochemistry, Bijvoet Center for Biomolecular Research, Utrecht University, Utrecht, The Netherlands. [2]Thermo Fisher Scientific, Eindhoven, The Netherlands. [3]Biomolecular Mass Spectrometry and Proteomics Group, Utrecht Institute for Pharmaceutical Sciences, Utrecht University, Utrecht, The Netherlands. [4]Netherlands Proteomics Center, Utrecht University, Utrecht University, Utrecht, The Netherlands. ✉e-mail: j.m.m.fedry@uu.nl; f.g.forster@uu.nl

## Subtomogram analysis of ribosome complexes

To analyse the elongation cycle of ER-bound ribosomes and the associated ER translocon complex we rapidly (within about 1 h) isolated ER-derived vesicles (microsomes) from HEK 293F cells for subsequent cryo-ET imaging (Extended Data Fig. 1 and Supplementary Fig. 1). We acquired a large dataset (869 tilt series) of frozen hydrated vesicles and used a regularized single-particle analysis approach to analyse the membrane-associated ribosome particles[17] (Extended Data Fig. 1). Extensive subtomogram analysis reveals the most abundant ribosome and translocon states. Altogether, we distinguish ten ribosomal intermediate states and four translocon variants, as well as two translocon-bound chaperones at resolutions ranging from 4 to 10 Å, which allows the identification of ribosomal intermediate states on the basis of high-resolution structures of isolates (Extended Data Fig. 2 and Supplementary Table 1).

## Ribosomal intermediates and 3D distribution

We first dissected the translational states of the ribosome pool consisting of membrane-bound and residual soluble particles. Focusing on the orientation of the SSU and association of tRNAs and elongation factors, we classified the particles into ten distinct states (Extended Data Figs. 1c and 2). To assess their translational activity, we examined the relative 3D distribution of the particles from the classes using a reciprocal neighbourhood probability analysis, which is indicative of integration into polysomes (Fig. 1a–c and Extended Data Fig. 3). Particles from eight classes (89%) show probability hotspots proximal to the ribosomal mRNA entrance and exit (E)-sites characteristic for membrane-bound and cytosolic ribosomes and consistent with previous lower-resolution analyses[18,19]. By contrast, two classes show a featureless neighbour distribution, implying that these particles are not involved in polysomes. The reconstructions of these two classes do not have tRNA bound in the peptidyl (P)-site and resemble known hibernating ribosome complexes bound to eEF2[20].

To assess the physiological relevance of our preparation, we analysed the distribution of ribosomal intermediate states in situ using focused ion beam (FIB) milled human cells. Although the lower yield of this approach resulted in substantially fewer particles (5,818) and reduced classification depth, it confirmed the high abundance of factor-bound classes (around 70%), and their presence on polysomes (Extended Data Fig. 4). The approximately 66% factor-bound ribosome complexes ex vivo exceed the abundance in previous ribosomal purification from HEK cells involving size-exclusion chromatography (around 8% in ref. [21]) (Extended Data Fig. 4f). Consistent with this previous cryo-electron microscopy (cryo-EM) analysis[21] and the high abundance of eEF1a and eEF2 in proteomics data of the sample (Supplementary Fig. 2), we identify eEF1a and eEF2 as ribosome-binding factors. Nevertheless, we stress that lysis and the isolation conditions may affect intermediate complexes and their abundances, which may eventually be overcome when higher resolution is achievable for FIB cryo-ET studies of human cells.

## Elongation cycle intermediates

To further analyse the polysome-associated ribosomal classes, we attempted to position them in the context of the elongation cycle as modelled on the basis of knowledge from previous in vitro reconstitution studies[3,21–24] (Fig. 1d). Although one class could not be conclusively assigned functionally (Supplementary Fig. 3), the remaining seven states are consistent with previous structural or biochemical data. The elongation cycle model commences with delivery of aa-tRNAs to the ribosome by GTP-bound eEF1a (Fig. 1d, decoding state). Approximately 22% of ribosomes in our data adopt an unrotated state, with clear densities for the tRNAs in the P- and E-sites and the eEF1a–tRNA ternary complex, which we assigned to a decoding population[25] (Fig. 1e). The position of eEF1a in

our decoding map differs slightly from a previously reported decoding state in polysomes purified from HEK cells[21], which may be a result of differences in the preparation protocols. The position of eEF1A in our decoding complex rather resembles a codon sampling state obtained by inhibiting eEF1a GTP hydrolysis[25] (Extended Data Fig. 5a,b). We speculate that the decoding population observed in our data may be explained by ribosomes testing non-cognate tRNAs that do not trigger GTP hydrolysis and occur more frequently than cognate tRNAs in the cell.

Next, we observe a highly abundant intermediate (33%) that has not been described previously: whereas the tRNA is accommodated in the canonical A-site and the SSU 'rolls' into the classical pre-configuration, eEF1a is bound to the ribosome in an extended conformation, which matches crystal structures of purified eEF1A–GDP[26] and its bacterial homologue EF-Tu–GDP[27] (Fig. 1e, Extended Data Fig. 5c,d and Supplementary Video 2). To analyse the pre+ state at higher resolution, we rapidly isolated soluble ribosomes and imaged them with cryo-EM single-particle analysis (SPA). Approximately 30% of particles were in the pre+ state, yielding a focused reconstruction of eEF1a with specific side chains of domain 3 (approximately 3.5 Å resolution) unambiguously identifying eEF1a (Extended Data Fig. 6a–g). In the classical pre+ state, eEF1a domains 1 and 3 interact with the sarcin–ricin loop (SRL) of the 28S rRNA (Extended Data Fig. 6h,i), whereas domain 2 blocks the A/T site and contacts the A-site tRNA. In a model of the human elongation cycle, we propose that the classical pre+ state may follow the decoding state, in which eEF1a still adopts a compact conformation. Although we cannot rule out that other factors observed at this site in situ could have been displaced by eEF1A (Extended Data Fig. 4) during the purification, the occurrence of the eEF1A bound classical pre+ state in purified samples indicates the possibility that eEF1a may remain bound to the ribosome during conformational switching to the extended form. This observation is different from bacteria, where no factors are observed in situ on the abundant pre-like A/P state[28] and suggests differences in eukaryote post-hydrolysis proofreading, possibly involving eEF1A[29–31]. The functional relevance of a possible eEF1a-bound classical pre+ state remains to be further investigated with complementary methods.

Next, we observe a previously described classical pre state, which we propose to occur after eEF1a fully dissociates from the ribosome, as the SSU and tRNAs remain unchanged (3%). We then identified two rotated states in our data: the rotated-1 pre state resulting from dissociation of a tRNA (4%), and the much more highly populated rotated-2 pre state with the tRNAs in hybrid A/P and P/E positions (17%). In contrast to previous studies of cytosolic polysomes[21], we found 5% of ribosomes in a state resembling a translocation intermediate associated with eEF2 and tRNAs in the canonical P- and E-sites. GTP hydrolysis seems to have occurred as indicated by the disordered switch I loop (Supplementary Fig. 4). This state resembles the late translocation intermediate post-3 state[24], which would be consistent with kinetic studies in the bacterial system[32]. Finally, we observe a similar state with tRNAs in the P- and E-sites and without eEF2, which is in good agreement with the post translocation (post) state (Fig. 1d).

Finally, we note that the assigned positions of the three most abundant states we observe are consistent with the elongation rate-limiting steps: decoding and pre+ correspond to proofreading steps, whereas rotated-2 precedes translocation.

## Hibernating ribosomes and ER stress

Membrane-bound hibernating ribosomes group into two major populations (Fig. 1f). A non-rotated state with a tRNA bound at the exit (E)-site and the protein CCDC124 occupying the P-site (7%) differs from a similar structure of the cytosolic hibernating ribosome[20] by eEF2 binding. We also detected a second rotated ribosome state (5%), which features eEF2 and from which CCDC124 is absent, analogous to the cytosolic hibernating ribosome[20]. To investigate the physiological

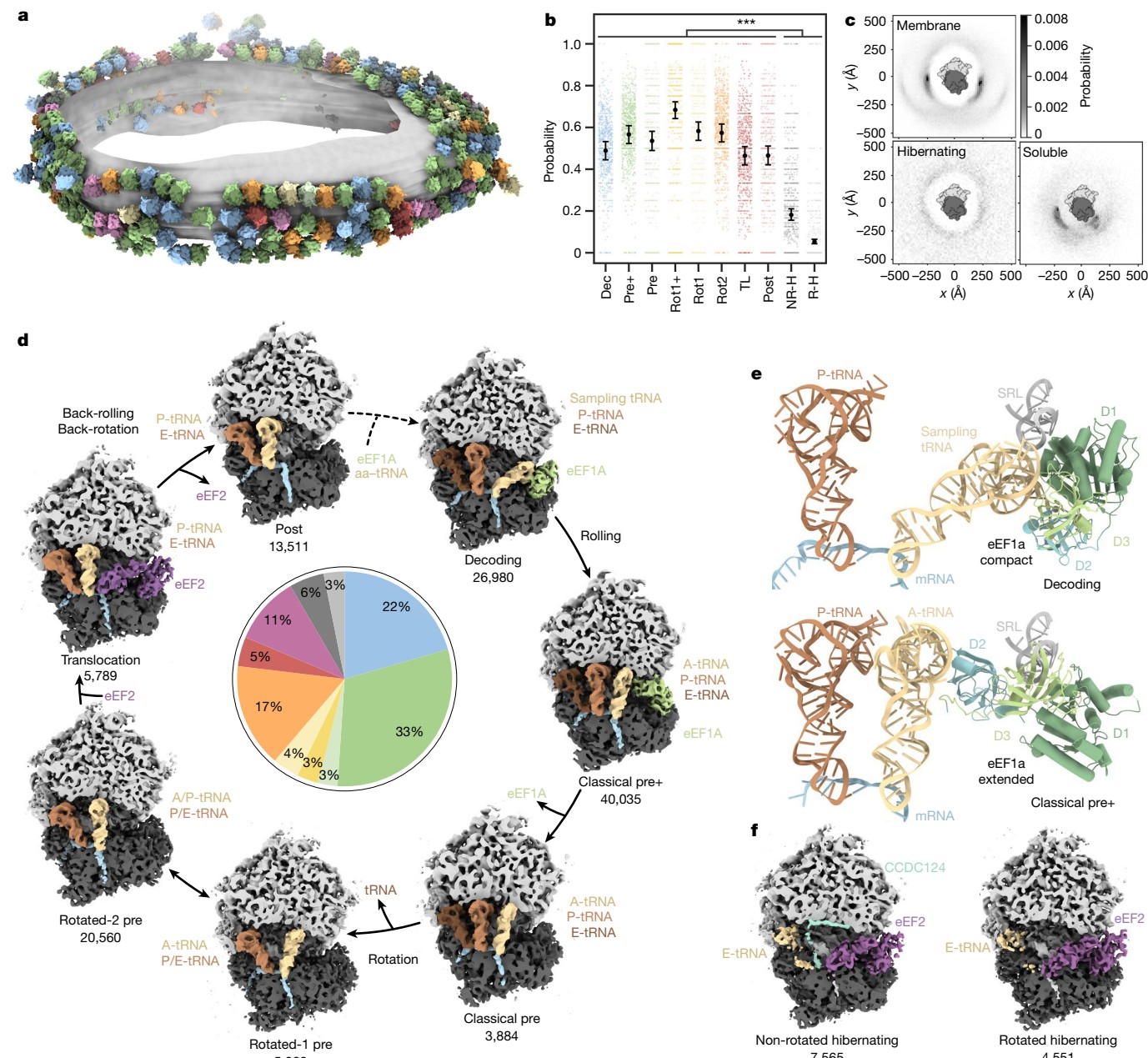

**Fig. 1 | Captured human ribosomal states and spatial distribution.**
**a**, Different ribosome states mapped back onto one exemplary ER-derived vesicle (*n* = 869 tomograms from one experiment, two independent replicates; Extended Data Fig. 4). **b**, Probabilities of ribosome states being present in polysomes. Black circles show the modelled mean and error bars represent the 95% confidence interval (CI) (*n* = 132,371 ribosomes with the 869 tomograms included as a random effect). Hochberg-adjusted *P* values were determined using a two-sided Wald-test. *P* values for comparison between hibernating and elongating states were all smaller than $2 \times 10^{-16}$. The small scattered points represent the frequencies of events per tomogram. TL, translocation; NR-H, non-rotated hibernating; R-H, rotated hibernating. **c**, Neighbour distribution of ER membrane-bound, hibernating and soluble ribosome particles. The membrane resides in the plane of the image. **d**, Observed active intermediates positioned in a model of the human elongation cycle. All reconstructions were filtered to 7 Å resolution. The ribosome is clipped for visualization. The tRNAs are colour-coded with respect to a complete cycle. The abundance of each state is indicated in the pie chart, colour-coded as in **a**. **e**, Close-up views of ribosome-bound compact eEF1a in the decoding-sampling state (PDB: 4CXG) and in the classical pre+ state (4C0S). D1 to D3 indicate the extended eEF1a domains 1 to 3. **f**, Reconstructions of two distinct ribosome states lacking tRNA at the P-site (hibernating states).

role of hibernating ribosomes, we also imaged microsomes from dithiothreitol (DTT)-treated HEK cells, in which elongation activity should be reduced[33]. Notably, we observed almost exclusively hibernating ribosomes upon treatment with DTT (Supplementary Fig. 5a,b). Thus, the abundance of ER-bound hibernating ribosomes depends strongly on cell state, and possibly also on cell density, as observed for cytosolic hibernating ribosomes[20]. We cannot rule out induction of some

hibernating ribosomes by lysis, which must be considered when interpreting the relative abundances.

## Native ER translocon distribution

We then grouped the particles according to their structural features near the ribosomal exit tunnel into five different classes (Fig. 2a): one

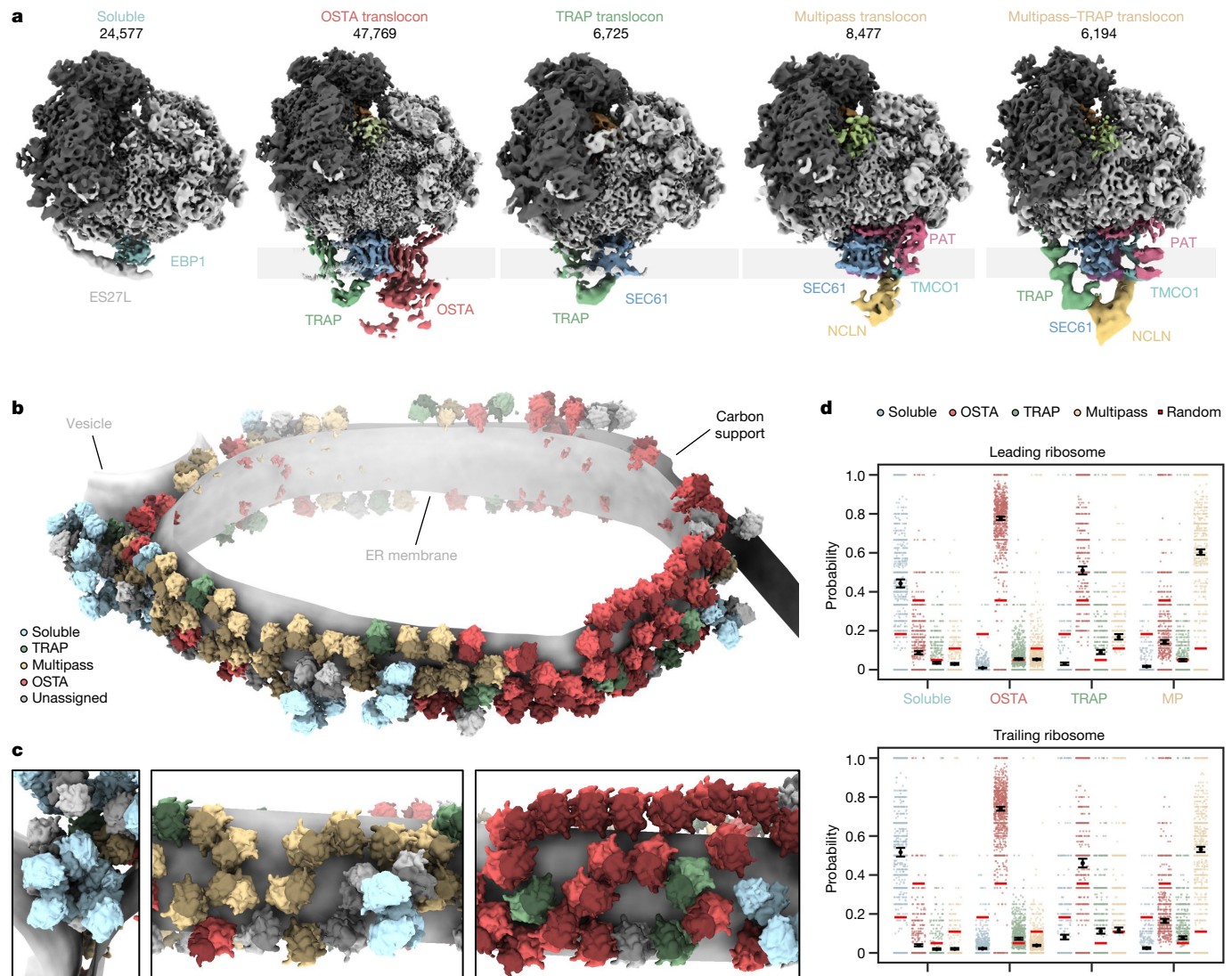

**Fig. 2 | Organization of soluble and ER membrane-associated ribosome populations. a**, Locally filtered reconstructions of different soluble and ER membrane-associated ribosome populations resulting from 3D classification focused near the exit tunnel. **b**, Segmented representation of one tomogram of an ER-derived vesicle (n = 869 tomograms from 1 experiment). Populations from **a** are mapped back into the reconstruction and coloured accordingly. **c**, Close-up views of the segmentation from **b**. **d**, The probability of encountering

ER-associated ribosomes from **a** as leading or trailing neighbour. Black circles show the modelled mean and error bars represent the 95% CI (n = 45,751 ribosomes with the 869 tomograms included as a random effect). The small, scattered points represent the frequencies of events per tomogram. The random association probability (bright red lines) is the overall abundance of the ribosome populations. MP, multipass.

soluble ribosome class and four membrane-bound ribosome classes. Approximately 30% of particles, mostly 'top views', were not assigned to any of these distinct five classes owing to insufficient signal or the missing wedge (Extended Data Fig. 1a). Soluble ribosomes are associated with EBP1 embraced by expansion segment 27L (ES27L) at the exit tunnel[34] (Supplementary Fig. 6), whereas membrane-bound ribosomes (64,208 particles) contact four distinct ER translocon complexes (Fig. 2a): the most populated SEC61–OSTA–TRAP (69% of ER-bound particles) and SEC61–TRAP translocons (10%) have previously been identified in cryo-ET datasets of dog pancreatic ER-derived microsomes[14]. The ER translocons in the remaining two classes (21%) have a common larger component, with one of them also harbouring TRAP. The common density has been observed but not identified previously in ER microsomes from HEK 293T cells upon knockout of OSTA subunit STT3a[7]. This translocon component resembles a recently discovered transmembrane protein complex responsible for insertion of multipass transmembrane proteins[15]. In addition to SEC61, the multipass

translocon comprises the insertase TMCO1, the PAT complex and the nicalin–TMEM147–NOMO complex[16,35]. To confirm the assignment of our density to the multipass (TMCO1–PAT–nicalin–TMEM147–NOMO) translocon, we knocked out CCDC47, a component of the PAT complex. Indeed, cryo-ET data of the ΔCCDC47 microsomes did not display the density at the position of the protein in the isolated multipass translocon[15] (Supplementary Fig. 7a,b). Thus, the major translocon types in wild-type HEK ER microsomes are SEC61-multipass, SEC61-multipass–TRAP, SEC61–OSTA–TRAP and SEC61–TRAP (Fig. 2a).

Mapping back the particles of these different ribosome–translocon populations in the original tomograms indicate clustering according to their translocon type (Fig. 2c). To further examine their polysomal organization, we used our neighbour probability analysis in the context of leading and trailing ribosome neighbours, which reflect late and early stages of translation, respectively (Fig. 2d and Extended Data Fig. 7). This statistical approach indicates a strong segregation of ribosomes bound to OSTA-containing and multipass translocons, as well as soluble

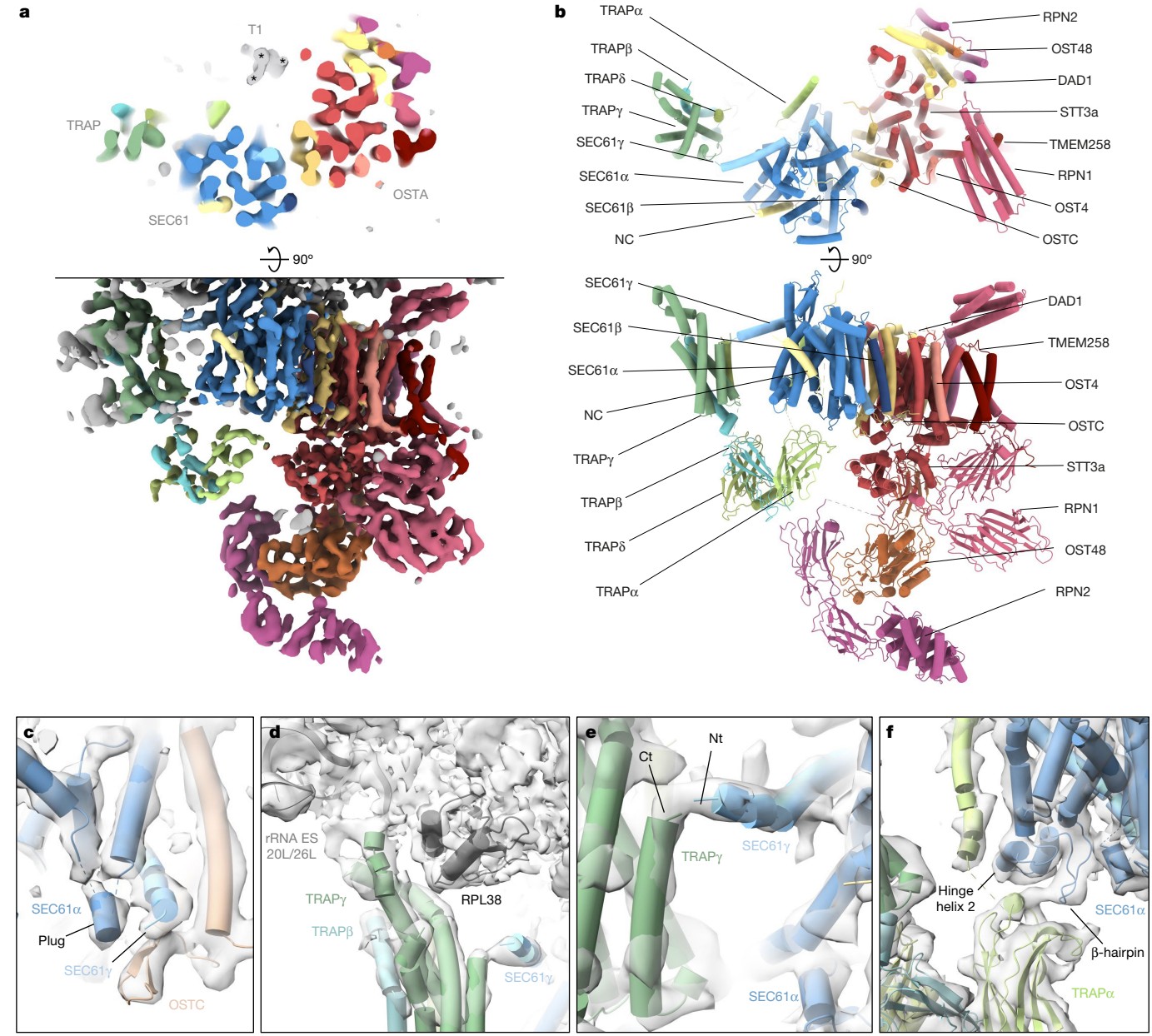

**Fig. 3 | Atomic model of the most abundant ER translocon. a**, Top view (top) and side view (bottom) of the translocon-centred reconstruction of SEC61–TRAP–OSTA. **b**, Atomic model of the ER translocon built from cryo-EM structures (PDB: 3JC2 and 6S7O) and AlphaFold predictions. **c–f**, Close-up views showing the molecular model placed into the segmented density maps. **c**, The plug helix of SEC61α contacts the SEC61γ C terminus and the luminal OSTC β-hairpin. SEC61α transmembrane helix 4 (TMH4) and SEC61β were removed for clarity. **d**, The cytosolic TRAPγ domain associates with rRNA expansion segments ES20L and ES26L and the ribosomal protein L35. **e**, The TRAPγ C terminus contacts the N terminus of SEC61γ. Ct, C terminus; Nt, N terminus. **f**, The luminal TRAPα domain interacts with a β-hairpin of the SEC61α hinge region and the TRAPα transmembrane helix contacts the second helix of the hinge region. SEC61α TM7 to TM10 were removed for clarity.

EBP1 (Fig. 2d). SEC61–TRAP translocons have less tendency to pair among themselves. They also neighbour OSTA-containing and multipass translocons, where they are preferably found as a trailing polysome neighbour (Extended Data Fig. 7c,d). Thus, nascent peptides preferentially encounter SEC61–TRAP translocons early in their biogenesis. Later, the membrane-bound translocon machineries specialize—this is consistent with recent studies on different model substrates[36].

## Architecture of the SEC61–OSTA–TRAP translocon

Ribosome-centred refinement of the most abundant population, the SEC61–OSTA–TRAP translocon, yielded a 4.2 Å-resolution structure (focused on the LSU) with poorly resolved transmembrane helices (TMHs) (7–10 Å resolution). Recentring on the ER luminal domains resolved those at improved resolution (6–8 Å) (Fig. 3a,b and Extended Data Fig. 8a–d). A composite of both densities enabled us to build a near-complete atomic model using AlphaFold[37].

The SEC61 channel opens its lateral gate to the lipid membrane[8,9] (Fig. 3a,b and Extended Data Fig. 8d). As in previous cryo-ET studies[12], the lateral gate accommodates a pronounced helical density, which matches the position of the signal peptide in isolates[6,38] and may represent an average of the signal peptides of the different proteins synthesized at the ER membrane. Moreover, a density is discernable near the ribosomal exit tunnel that may correspond to an average of nascent

chains (Extended Data Fig. 8e). The luminal part of SeC61 reveals a short α-helix, which we assigned to the SEC61α plug (Fig. 3c and Extended Data Fig. 8f,g). This hallmark feature of SEC61 was not resolved in lower-resolution cryo-ET studies[12] and higher-resolution structures of solubilized ribosome–SEC61 complexes[6,7]. Here, we observe the plug in a displaced conformation stabilized by SEC61γ and the oligosaccharyltransferase complex subunit OSTC. This arrangement resembles the yeast post-translocon, in which SEC63 stabilizes the plug[39].

To investigate the structural deviations of SEC61–TRAP–OSTA when bound to hibernating ribosomes, we reconstructed the inactive SEC61–TRAP–OSTA from the DTT-stressed microsomes. Although the SEC61 plug closes in the inactive complex, the density reveals an open lateral gate accommodating a helical density (Supplementary Fig. 5c–f). Since signal peptides can be cleaved co-translationally[40], this helix might correspond to a pool of cleaved signal peptides or to an unknown specific peptide.

An AlphaFold-based model of TRAP could be fitted unambiguously into the SEC61–OSTA–TRAP translocon map, requiring only minor repositioning of single transmembrane helices and removal of low-confidence segments (Extended Data Fig. 9a–c). The assembly model does not display notable clashes and density in the lumen coincides with predicted *N*-glycosylation sites of TRAPα and TRAPβ, further supporting our assignment (Extended Data Fig. 9d).

As previously observed[16], the cytosolic domain of TRAPγ tethers TRAP to RPL38e and the rRNA expansion segments ES20L and ES26L (Fig. 3d). Our results reveal the position of the TRAPα transmembrane helix, separated by a 2–3.5 nm lipid density from the major transmembrane part of TRAP, which comprises TRAPγ, TRAPβ and TRAPδ. In addition, we visualize a contact between the previously unresolved C terminus of TRAPγ and the amphipathic SEC61γ N-terminal helix at the cytosolic face of the membrane (Fig. 3e and Extended Data Fig. 9e). The fibronectin fold domains of TRAPα, TRAPβ and TRAPδ form the luminal part of TRAP, where they may interact with nascent proteins in a confined space. Near the luminal end of its transmembrane helix, the FG and BC loops of the TRAPα fibronectin-like domain associate with the SEC61α hinge region (Fig. 3f), which bridges the pseudo-symmetric N- and C-terminal halves of SEC61. Finally, we observe that TRAP association is not restricted to laterally open SEC61 as the SEC61-multipass–TRAP translocon displays a closed lateral gate both in the presence of CCDC47, as recently shown[41], as well as in the absence of CCDC47 (Supplementary Fig. 7h,i).

Cellular and biochemical studies indicate that TRAP is required for the biogenesis of proteins that exhibit signal peptides with weak helical propensity due to glycine and proline residues[11]. Preproteins with pronounced hydrophobic helical signal peptides are subject to stronger pulling forces than TRAP-dependent preproteins of, for example, prion protein[42], presumably owing to the lower affinity of their signal peptides for the lateral gate. Although the structure of TRAP–SEC61 does not provide an obvious mechanism of action for the TRAP complex, it enables to formulate a hypothesis. When signal peptides traverse SEC61 head-on and enter the lumen, they contact the luminal TRAPα domain[43]. We speculate that the growing nascent chain pushing against the SEC61 hinge-bound TRAPα domain might then open the SEC61 lateral gate via an allosteric mechanism and expose its hydrophobic surface to accommodate the signal peptide. Alternatively, it was suggested during the revision of this work that lipid bilayer modulation induced by TRAP, which can indeed be observed in our membrane-embedded structure, could promote insertion of signal peptides[44]. Further studies will be required to evaluate the mechanistic function of the TRAP interactions revealed in this study.

## Native OSTA and its associated factors

The cryo-ET structure is in excellent agreement with the cryo-EM SPA structure of solubilized OSTA[4], which lacks the RPN2 N-terminal domain

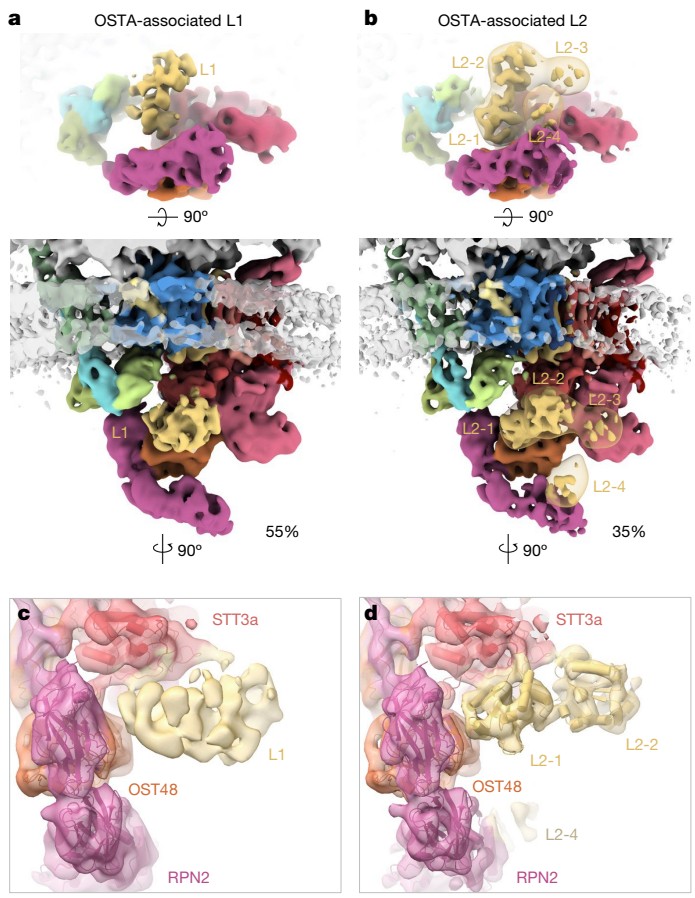

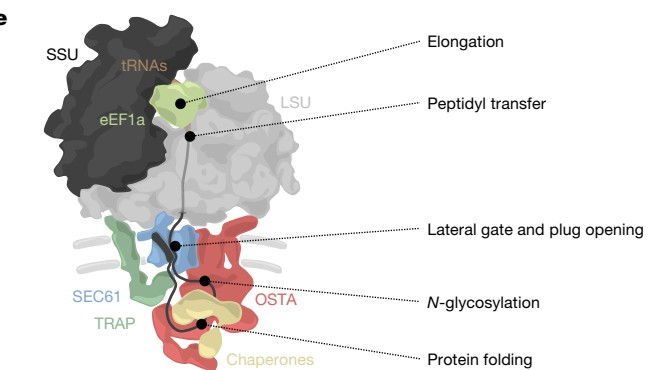

**Fig. 4 | Co-translational ER biogenesis factors and summary. a,b**, Top view (top) and front view (bottom) of the accessory factors L1 (**a**) and L2 (**b**) associated with the SEC61–TRAP–OSTA translocon. The transparent map represents L2 filtered to a resolution of 20 Å. **c,d**, Close-up view of the interaction site between STT3a and L1 (**c**) or L2 (**d**). Domains a and b of the L2 candidate protein PDIR were placed into domain L2-1 and L2-2, respectively. **e**, Model of the main protein translation and translocation machinery at the ER membrane.

(Extended Data Fig. 10a). To complete the atomic model, we fitted the corresponding AlphaFold models into the most membrane-distal part of our map (Extended Data Fig. 10b,c). However, the SEC61–TRAP–OSTA model does not explain a transmembrane helix structure (T1)—approximately 15 kDa in size—comprising three transmembrane helices and a characteristic amphipathic helix facing the cytosol (Fig. 3a and Extended Data Fig. 10d–f). T1 is intercalated between STT3a TMH9 and the C terminus of the TRAPα TMH, resulting in the formation of a lipid-filled cavity near the hinge region of SEC61. In line with OSTA association and cavity formation, T1 was observed only in the

OSTA-containing ER translocon (Extended Data Fig. 10g,h). The glucosyltransferases acting upstream of OSTA or the dolichyldiphosphatase I acting downstream of OSTA are candidates for T1[45], but neither atomic model provides an acceptable fit. Thus, further investigation will be required to determine the molecular identity of T1.

We observed weak density associated with the luminal domain of STT3a. For higher-resolution insights into possible sub-stoichiometric binding partners, we performed classification focused on the SEC61-proximal luminal face of the OSTA, which revealed three distinct populations: (1) OSTA without accessory factors (11%), (2) OSTA in complex with a globular density of approximately 35 kDa (L1, 54%), and (3) OSTA in complex with a density of approximately 60 kDa (L2, 35%) (Fig. 4a,b and Extended Data Fig. 10i–l). L1 associates with negatively charged residues at the C terminus of STT3a (amino acids 667–676). L2 comprises four approximately equally sized domains (Fig. 4b), of which domains L2-1 and L2-2 compete for the same binding site with L1. Domain L2-4 binds the N-terminal domain of RPN2, and L2-3 does not interact with OSTA. Whereas L2-1 and L2-2 reveal secondary structure elements, L2-3 and L2-4 bind the flexible RPN2 N-terminal domain and are poorly resolved.

To our knowledge, L1 and L2 have not previously been observed in OSTA complexes purified from HEK cells[4]; they are likely to represent transiently binding proteins. The ER contains many chaperones that assist in protein biogenesis, which are prime candidates for L1 and L2[46]. Among the ER chaperones, prolyl isomerase cyclophilin B is most abundant in the sample (Supplementary Table 2) and shows the best agreement in shape and size with L1, which is, however, too small and globular for unambiguous assignment. Protein disulfide isomerases[47] (PDIs) show a good fit with the characteristic four-domain structure of the larger L2 well (Fig. 4d and Extended Data Fig. 10m–p). The negative charges at the interacting site of STT3a would be consistent with the interaction pattern observed for PDIs with calnexin and calreticulin[48] (Extended Data Fig. 10q). PDIs are highly abundant in the sample (Supplementary Table 2) with glycoprotein-specific family member PDIA3 (also known as ERp57), probably representing L2[49]. The transient recruitment of an oxidoreductase to OSTA is plausible, as its post-translational counterpart OSTB features a constitutive oxidoreductase[50] (N33 (also known as Tusc3)).

## Conclusions

In summary, extensive classification of cryo-ET data visualizes the process of ER-associated translation and the dynamic recruitment protein biogenesis factors in the context of polysomes (Fig. 4c). This study on the ensemble of secretory proteins synthesized in the cell complements biochemical analyses[36,41] and forms the basis for future investigation of the biogenesis of specific proteins and the change of the machinery in distinct cellular states and diseases.

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

## Methods

### CRISPR–Cas9 knockout of CCDC47

FreeStyle 293-F cells (Thermo Fisher Scientific, R79007) were transfected with the plasmid pSpCas9(BB)-2A-Puro (PX459) V2.0 from the F. Zhang laboratory (Addgene plasmid 62988) containing the 20-bp single guide RNA (sgRNA) target sequence 5′-CACCGGTACACGGTGAACTCGTGCG-3′, PAM: AGG or 5′-CACCGGGAGGAAGCGGGCGAGGTGC-3′, PAM:GGG. Transfection was performed using Lipofectamine 2000 (Thermo Fisher Scientifiic, 11668019) according to the manufacturer's instructions using 1 µg DNA per ml of culture at a cell density of $1 \times 10^6$ cells per ml. Cells were cultured for 48 h in FreeStyle 293 expression medium (Thermo Fisher Scientific, 12338018) on an orbital shaker (120 RPM) at 37 °C and supplemented with 5% $CO_2$. Two days after transfection, cells were collected and resuspended in complete Dulbecco's Modified Eagle Medium (DMEM) (Thermo Fisher Scientific, 11966025) (supplemented with 10% fetal bovine serum (Thermo Fisher Scientific, 10100147) and GlutaMAX-I (Thermo Fisher Scientific, 35050061)) with 0.5 µg ml⁻¹ Puromycin (InvivoGen, ant-pr-1). Subsequently, cells were plated in T175 flasks (Thermo Fisher Scientific, 159910) and grown for 7 days in complete DMEM with 0.5 µg ml⁻¹ Puromycin with periodical medium exchange or sub-culturing when confluency was reached.

After 7 days of Puromycin selection, surviving cells were dislodged, collected, and resuspended at 5 cells per ml in conditioned complete DMEM. One-hundred and fifty microlitres per well of cell suspension was plated into sterile 96-well plates and cultured for 14 days. Cell colonies derived from single cells were used for further cell expansion. After 14 days in culture, conditioned complete DMEM was exchanged for FreeStyle medium and cell colonies transferred into 24-well plates. Subsequently, cells were grown to confluency and further expanded into 6-well plates and 10-cm dishes before analysis.

### Cell culture

HeLa and U2OS cells (from ATCC, CVCL_0042 and CVCL_0030 in Cellosaurus.org, respectively) were grown in standard tissue culture conditions (37°, 5% CO2) in DMEM Glutamax (Gibco). HEK 293F cells (Thermo Fisher Scientific, R79007) were grown in suspension in FreeStyle medium with 120 rpm agitation. Cell lines were not authenticated and were tested for negative mycoplasma.

### ER-vesicle preparation

HEK 293F wild-type or CCDC47 knockout cells ($0.5–1 \times 10^6$ cells per ml, 50 ml) were collected and washed (3 times with PBS, at 300$g$, 5 min, 4 °C). HEK 293F cells used for ER stress studies were treated with 10 mM DTT for 2 h before collection. Cells were resuspended in lysis buffer (2–4 ml, 10 mM HEPES-NaOH pH 7.4, 250 mM sucrose, 2 mM $MgCl_2$, 0.5 mM DTT, protease inhibitor cocktail (Roche)) and lysed using a Isobiotec cell cracker (5–10 passes, 14 µm clearance, on ice). The lysate was cleared (1,500$g$, 2–3 × 5 min, 4 °C, in 2 ml tubes) using a cooled tabletop centrifuge. Vesicles were pelleted (10,000$g$, 10 min, 4 °C), and washed with resuspension buffer (10 mM HEPES, 250 mM sucrose, 1 mM $MgCl_2$, 0.5 mM DTT). The pellet was resuspended at a concentration of ~50 mg ml⁻¹ determined by $A_{280}$, frozen in liquid nitrogen and stored at −80 °C until further use. The supernatant was used for proteomics as control.

Twenty micrograms of microsomes were used for SDS–PAGE followed by immunoblotting using antibodies against SEC61α (Abcam, ab15575; 1:1,000), TRAPγ (Sigma Aldrich, hpa014906; 1:1,000) and CCDC47 (Abcam, ab241608; 1:1,000).

### Mass spectrometry data acquisition

Approximately 100 µg of the isolated ER-microsome and cytosolic fraction (supernatant) were digested using an S-Trap micro-MS column (protifi) according to the vendor's protocol.

Proteins were solubilized in lysis buffer (10% SDS, 100 mM Tris, pH 8), reduced (100 mM TCEP), alkylated (400 mM CAA in isopropanol) and denatured (27.5% phosphoric acid). For protein trapping, samples were flown over an S-Trap micro spin column, (10,000$g$, 30 s) and further washed with binding buffer (100 mM triethylammonium bicarbonate (TEAB) buffer, in 90% methanol). Protein digestion was achieved with an overnight incubation at 37 °C using a water bath (Grant Instruments, JB Academy) after the addition of digestion buffer (10% trypsin, 2% lysine, 50 mM Tris). Protein peptides were retrieved by washing with elution buffer (50 mM Tris), using a tabletop centrifuge (10,000$g$, 1 min).

Eluted peptides were lyophilized and dissolved in 2% formic acid prior to liquid chromatography–mass spectrometry (LC–MS/MS) data acquisition. MS data were acquired using an Ultimate 3000 RSLC nano system (Thermo Scientific) coupled to an Exploris 480 (Thermo Scientific). Three technical replicates of each sample were measured. Peptides were first trapped in a pre-column (Dr. Maisch Reprosil C18, 3 µm, 2 cm × 100 µm) prior to separation on the analytical column packed in-house (Poroshell EC-C18, 2.7 µm, 50 cm × 75 µm), both columns were kept at 40 °C in the built-in oven. Trapping was performed for 10 min in solvent A (0.1% v/v formic acid in water), and the elution gradient profile was as follows: 0–10% solvent B (0.1% v/v formic acid in 80% v/v acetonitrile) over 5 min, 13–44% solvent B over 37 min, 44–100% solvent B over 4 min, and finally 100% B for 4 min before re-equilibration in 100% A for 8 min. The mass spectrometer was operated in a data-dependent mode. Full-scan mass spectra were collected in a mass range of $m/z$ 350–1,300 Thomson (Th) in the Orbitrap at a resolution of 60,000 after accumulation to an AGC target value of $10^6$ with a maximum injection time of 50 ms. In-source fragmentation was activated and set to 15 eV. The cycle time for the acquisition of MS/MS fragmentation scans was set to 1 s. Dynamic exclusion properties were set to n = 1 and to an exclusion duration of 10 s. HCD fragmentation (MS/MS) was performed with a fixed normalized collision energy of 27% and the mass spectra acquired in the Orbitrap at a resolution of 30,000 after accumulation to an AGC target value of $10^5$ with an isolation window of $m/z$ = 1.4 Th.

Raw data were processed using the MaxQuant software[51] version 2.0.1.0 with standard settings applied. In brief, the extracted peak lists were searched against the reviewed Human UniProtKB database (date 15 July 2021; 20,353 entries), with an allowed precursor mass deviation of 4.5 ppm and an allowed fragment mass deviation of 20 ppm. Cysteine carbamidomethylation was set as static modification, and methionine oxidation, N-terminal acetylation as variable modifications (maximum 5 modifications per peptide allowed). Both LFQ quantification and 'match between runs' were enabled. The iBAQ values in Supplementary Fig. 4b are approximate absolute abundances of the identified proteins derived by the normalization of the summed peptide intensities by the number of theoretically observable peptides for a given protein. Raw data were processed using the MaxQuant software[51] version 2.0.1.0 with standard settings applied. In brief, the extracted peak lists were searched against the reviewed Human UniProtKB database (date 15 July 2021; 20,353 entries), with an allowed precursor mass deviation of 4.5 ppm and an allowed fragment mass deviation of 20 ppm. Cysteine carbamidomethylation was set as static modification, and methionine oxidation, N-terminal acetylation as variable modifications (maximum five modifications per peptide allowed). Both LFQ quantification and 'match between runs' were enabled. The iBAQ values in Supplementary Fig. 4b are approximate absolute abundances of the identified proteins derived by the normalization of the summed peptide intensities by the number of theoretically observable peptides for a given protein.

### Grid preparation

ER vesicles were diluted in resuspension buffer to a concentration of 2–3 mg ml⁻¹ and 2 µl were applied onto a glow-discharged lacey carbon grid (Quantifoil). Four m,icrolitres of BSA-conjugated gold beads (10 nm, UMC Utrecht) diluted in resuspension buffer without sucrose were added and mixed with the sample on grid. Grids were immediately

blotted from the backside for 5–6 s and plunged into a mix of liquid ethane and propane using a manual plunger.

For the adherent cell lines (Hela and U2OS), cells were seeded on R2/2 holey carbon on gold grids (Quantifoil) coated with fibronectin in a Mattek dish and incubated for 24 h. The suspension HEK 293F cells were grown to mid-log phase, and the cells were then directly pipetted onto glow-discharged R2/1 Carbon on Copper grids (Quantifoil). Grids were immediately blotted from the back for 10 s and plunged into liquid ethane propane mix using a manual plunger.

### Lamella preparation
Lamellae were prepared using an Aquilos FIB-SEM system (Thermo Fisher Scientific). Grids were sputtered with an initial platinum coat (10 s) followed by a 10 s gas injection system (GIS) to add an extra protective layer of organometallic platinum. Samples were tilted to an angle of 15° to 22° and 12 μm wide lamellae were prepared. The milling process was performed with an ion beam of 30 kV energy in 3 steps: (1) 500 pA, gap 3 μm with expansion joints, (2) 300 pA, gap 1 μm, (3) 100 pA, gap 500 nm. Lamellae were finally polished at 30–50 pA with a gap of 200 nm.

### Data acquisition
We acquired 869 tilt series on a Talos Arctica (Thermo Fisher Scientific) operated at an acceleration voltage of 200 kV and equipped with a K2 summit direct electron detector and energy filter (Gatan). Images were recorded in movies of 7–8 frames at a target defocus of 3 μm and an object pixel size of 1.72 Å. Tilt series were acquired in SerialEM (3.8)[52] using a grouped dose-symmetric tilt scheme[53] covering a range of ±54° with an angular increment of 3°. The cumulative dose of a series did not exceed 80 e$^-$ Å$^{-2}$.

Lamella data used in this analysis has been collected in one session on a pool of grids of human cell lines. Twenty-seven tilt series were acquired on six different lamellae on a Talos Arctica (same instrument as above). Images were recorded in movies of 5–8 frames at a target defocus of 4 μm and an object pixel size of 2.17 Å. Tilt series were acquired in SerialEM using a grouped dose-symmetric tilt scheme covering a range of ±60° with a pre tilt of ±10° and an angular increment of 3°. The cumulative dose of a series did not exceed 70 e$^-$ Å$^{-2}$.

### Reconstruction and particle localization
Video files of individual projection images were motion-corrected in Warp (1.0.9)[54] and combined into stacks of tilt series with the determined contrast transfer function (CTF) parameters. The combined stacks were aligned using the gold fiducials in IMOD (4.10.25)[55]. Per-tilt CTF estimation for entire tilt series was performed in Warp and full deconvoluted tomograms were reconstructed by weighted back projection at a pixel size of 20 Å. Ice thickness was determined manually for a subset of 50 tomograms and results in an average thickness of 156 nm. Particle coordinates were determined by template matching against a reconstruction of a human 80S ribosome filtered to 40 Å and downsampled to match the tomogram pixel size (20 Å) using pyTOM (0.994)[56]. Most false-positive hits were manually removed in pyTOM. The determined positions of ribosomes were used to extract subtomograms and their corresponding CTF volumes at a pixel size of 3.45 Å (2× binned) in Warp. Video files of individual projection images were motion-corrected in Warp[54] and combined into stacks of tilt series with the determined CTF parameters. The combined stacks were aligned using the gold fiducials in IMOD[55]. Per-tilt CTF estimation for entire tilt series was performed in Warp and full deconvoluted tomograms were reconstructed by weighted back projection at a pixel size of 20 Å. Ice thickness was determined manually for a subset of 50 tomograms and results in an average thickness of 156 nm. Particle coordinates were determined by template matching against a reconstruction of a human 80S ribosome filtered to 40 Å and downsampled to match the tomogram pixel size (20 Å) using pyTOM[56]. Most false-positive hits

were manually removed in pyTOM. The determined positions of ribosomes were used to extract subtomograms and their corresponding CTF volumes at a pixel size of 3.45 Å (2× binned) in Warp.

Lamellae data were processed as above with slight variations. Video files of individual projection images were motion- and CTF-corrected in Warp and combined into stacks of tilt series. The combined stacks were aligned using patch tracking in IMOD. CTF estimation for entire tilt series was performed in Warp and full tomograms were reconstructed by weighted back projection at a pixel size of 17.36 Å. Ice thickness was determined manually and was found to be <200 nm for all lamellae. Particle coordinates were determined by template matching against a reconstruction of a human 80S ribosome filtered to 40 Å using downsampled to match the tomogram pixel size (17.36 Å) pyTOM. The determined positions of ribosomes were used to extract subtomograms and corresponding CTF volumes at a pixel size of 8.68 Å (4× binned) in Warp.

### Subtomogram analysis
The extracted subtomograms were aligned in RELION (3.1.1)[57] using a spherical mask with a diameter of 300 Å against a reference of an 80S ribosome obtained from a subset of the same data. The extracted subtomograms were aligned in RELION (3.1.1)[57] using a spherical mask with a diameter of 300 Å against a reference of an 80S ribosome obtained from a subset of the same data. The aligned particles were refined in M (1.0.9)[17] using the reconstructions of the two half maps as a reference and a tight soft mask focused on the LSU at a pixel size of 3.45 Å. Particles were subjected to 2–3 rounds of refining image warp grid, particle poses, stage angles, volume warp grid, defocus and pixel size. After refinements, new subtomograms and their corresponding CTF volumes were extracted at a pixel size of 6.9 Å (4× binned) and subjected to 3D classification (without mask, without reference, $T = 4$ and classes = 50) to sort out remaining false positives, poorly aligned particles, and lone LSUs. The remaining 134,350 particles were used for subsequent focused classification steps to dissect ribosomal intermediate states or translocon variants.

### Classification of ER ribosome populations
All 134,350 particles were subjected to 3D classification (without reference, with soft mask, $T = 4$, classes = 20) in RELION, focused on the area at the ribosomal tunnel exit including the membrane and translocon. Particles were sorted into SEC61–TRAP-bound, SEC61–TRAP–OST-bound, SEC61-multipass-bound and EBP1-bound ribosomes and a combined class of ribosomes with ambiguous densities. Ribosomes with ambiguous densities were subjected to two further classification rounds and sorted the respective class from above until no further separation could be achieved. Ribosomes that associated with the EBP1 were designated 'soluble', ribosomes associated with translocon variants were designated 'membrane-bound' and ribosomes associated with ambiguous densities were designated 'unidentified'.

Subtomograms of the multipass translocon were recentered by 17 nm from the centre of the ribosome towards SEC61 and extracted in M at a voxel size of 6.9 Å. Subsequently, subtomograms were classified focused on the luminal domains of TRAP and NCLN (with reference of all multipass translocons, with soft mask, $T = 4$, classes = 3) or focused on the cytosolic domain of CCDC47 (with reference, with mask, $T = 3$, classes = 2). The TRAP-multipass translocon was further refined using local angular searches in RELION or, to obtain ribosome-centred reconstructions of the multipass translocon populations, subtomograms were recentered again by 17 nm towards the centre of the ribosome in M and subjected to another round of refinement.

### Refinement of the OST translocon
The 42,215 best-correlating particles (5,554 particles were poorly aligned) of the OST-bound ribosome were used for refinement focused on the LSU in M using the same parameters as above at a pixel size of 1.72 Å (unbinned), which resulted in a reconstruction at an overall

resolution of ~4 Å. However, densities of OST or TRAP in the ER lumen were poorly resolved. To improve local resolution of the translocon components, the reconstruction was recentered by 19.5 nm from the centre of the ribosome towards the OST translocon and subtomograms were extracted in M at a pixel size of 3.45 Å. The particles were aligned in RELION using the average of the recentered reconstruction of the OST translocon as reference and a tight soft mask focused on SEC61, TRAP and OST. Subsequently, the aligned particles were refined in M as above at a pixel size of 1.72 Å resulting in a reconstruction at an overall resolution of 8 Å. Local resolutions estimated using M[17] ranged from 6–7 Å for the OST and 8–9 Å for TRAP and the N-terminal domain of RPN2, indicating flexibility. Local refinement focused on the TRAP complex did not improve its resolution, presumably because the protein complex was too small to provide sufficient signal for reliable refinement.

After refinement in M, translocon-centred OST-particles were extracted at a pixel size of 6.9 Å and subjected to classification in RELION (without reference, with mask, $T = 10$, classes = 4) focused on the chaperone binding site. The resulting classes were refined in M as above using masks focusing on SEC61, TRAP, OST and chaperone.

## Classification of ribosomal intermediates

Ribosomal intermediate states were obtained by hierarchical classification focused on the rotation of the SSU and on the tRNA and elongation factor binding sites. First, all 134,350 particles were classified into classes of ribosomes with non-rotated and rotated SSU (with reference, with soft tight mask focused on SSU, $T = 4$, classes = 2). Subsequently, non-rotated and rotated particles were each subjected to two rounds of classification (with reference, with mask focused on tRNA and elongation factor binding site, $T = 10$–$20$, classes = 10–20). Classes with fragmented densities, such as pre/pre+, rotated−1/rotated−1+, non-rotated idle/translocation, were separated in the second round of classification (with reference, with mask focused on tRNA and elongation factor binding site, $T = 10$–$20$, classes = 2–4).

Classification of intermediate states was first performed for individual populations of ER translocon-bound or soluble ribosomes, which revealed similar results for each population. However, to improve performance of classification, especially for translocon-associated populations with a low number of particles, we pooled all translocon and soluble populations and performed classification of intermediates on the entire dataset. Subsequently, particle sets of individual intermediate states were dissected according to the translocon-associated and soluble ribosome populations.

The classification workflow was repeated four times to assess the technical uncertainties of 3D classification, which was determined at 5% to 15% and correlates inversely with class size. To assess experimental reproducibility, we combined two smaller datasets of ER-derived vesicles (31 tomograms, 6,101 particles; 58 tomograms, 3,836 particles) with the large dataset (869 tomograms, 134,350 particles) and processed them as described above. After obtaining classes of intermediate states, particle numbers were determined for each dataset and class.

The classification workflow was applied to in situ data with slight variations: extracted subtomograms were used for 3D classification with image alignment against a low pass filtered 80S ribosome map as reference in RELION to exclude false positive. The remaining 5,818 ribosome subtomograms were refined in RELION and re-extracted in Warp at a pixel size of 4.34 Å (2× binned). Two times-binned subtomograms were refined in RELION with a mask on the LSU prior to a first round of 3D classification without image alignment with a mask on the SSU to separate rotated from non-rotated ribosomes. A second round of classification was performed using a mask positioned on the tRNA and elongation factors sites, optimizing the mask extension and class number to this data in order to yield stable classes despite limited resolution and particle number. The different classes were finally subjected to iterative refinement in M.

## Refinement of intermediate states

Classes of ribosomal intermediate states were simultaneously refined in M at a pixel size of 1.72 Å (unbinned) using tight masks focused on the entire 80S ribosome, tRNAs and elongation factors, which were individually generated for each intermediate. Refinement of image warp grid, particle poses, stage angles, volume warp grid, defocus and pixel size were performed iteratively (2–3 iterations). Globally or locally filtered and sharpened maps were generated by M and used for visualization or model building.

## Model building

Initial models for each chain of SEC61 and the OST were downloaded from the Alphafold database[58]. A polyalanine helical stretch was manually built to account for the plug density. The OSTA chains were manually docked into the higher-resolution OSTA SPA map EMD-10110, followed by refinement through an iterative cycling between phenix (1.20.1) refine[59], isolde (1.0b5)[60] and Coot (0.9.8.2)[61]. The initial model for TRAP was built using AlphaFold Colab[37] and Coot[61]. The initial model for TRAP was built using AlphaFold Colab for multimeric complexes[62] and was divided into the transmembrane part and the luminal part. Each model was manually fitted into our subtomogram average (STA) density in UCSF Chimera (1.14.0)[63], followed by normal-mode guided refinement using iMODFIT (1.51)[64]. Long flexible loops not visible in our density were manually removed from the models. SEC61, OSTA and luminal TRAP domains were fitted and refined into a STA centred on the OST, while the TRAP transmembrane helices were fitted and refined into the original ribosome-centred STA, in which they were better defined. Each model was refined using iterative cycling between phenix refine, Isolde and Coot. Models were then combined for one last round of refinement together in the OST centred STA. Validation was performed using Molprobity (4.5.1)[65]. UCSF ChimeraX (1.3.0)[63] was used for visualization of all models and reconstructions.

## Single-particle analysis

Suspension HEK 293F cells were grown to mid-log phase ($0.5$–$1 \times 10^6$ cells per ml, 50 ml). Cells were pelleted at 500 g for 5 min and washed twice in ice cold PBS and resuspended in 10 mM Hepes KOH, pH 7.5, 250 mM sucrose, 2 mM magnesium acetate, 0.5 mM DTT, 0.5 mM PMSF, protease inhibitor tablets). Cells were lysed with 30 passages through a 21-gauge needle. The lysate was cleared by centrifugation steps at 1,000 g for 10 min, 1,500 g for 15 min and 20,000 g for 20 min. The final supernatant was loaded onto a 1 M sucrose cushion and spun at 300,000 g for 1 h. The final ribosomal pellet was resuspended in lysis buffer and snap frozen in liquid nitrogen. For grid preparation, 3.5 µl of the ribosome preparation was pipetted onto glow-discharged R 3.5/1 2 nm C holey grids (Quantifoil) and blotted for 2.5 s at force 0 using a Vitrobot (Thermo Fisher Scientific) before subsequent plunging into liquid ethane.

Single-particle cryo-EM data were acquired on a Titan Krios (Thermo Fisher Scientific) equipped with a cold FEG, Falcon 4i detector and Selectris X energy filter 10 eV slit at a pixel size of 0.729 Å per pixel. A total of 17,000 movies was acquired with EPU 3 (Thermo Fisher Scientific) in EER format. A cumulative dose of 40 e$^-$ Å$^{-2}$ was used.

The data was processed in Relion 3.1.1. Movies were motion-corrected and CTF was estimated. Particles were picked with the logpicker and reconstructed at a pixel size of 6 Å per pixel for subsequent 2D classification, followed by 3D classification with image alignment to exclude false-positive and low-quality particles. A total of 66,000 particles was then subjected to 3D classification without image alignment using a mask on the A tRNA site and the GTPase centre. 19,000 particles were selected in a class corresponding to the classical pre+ state, refined, re-extracted at 1.0 Å per pixel and refined again. CtfRefine was performed followed by another round of refinement. Masks on the A-site tRNA site and elongation factor, as well as on the peptidyl transferase

centre were used for particle subtraction and focused refinements to improve the quality of the maps in these regions.

For model building, a previous crystallographic structure of eEF1A in the extended GDP bound conformation (PDB 4C0S) was used as starting model and was first briefly refined in real space in the higher-resolution crystallographic electron density map using Isolde and phenix refine, in order to improve the starting geometry of the model. The resulting model was then refined in our map through iterative cycling between phenix refine[59], Isolde[60] and Coot[61]. The model was validated using Coot[61] and Molprobity[65].

### Sequence conservation
The degree of sequence conservation was determined using the ConSurf server[66] using 150 homologous sequences with a sequence identity ranging from 35%–95%. The conservation score was plotted onto the surface of the respective protein model in UCSF Chimera.

### Polysome analysis
For the neighbourhood analysis, ribosome positions and orientations were read from the RELION star files resulting from subtomogram alignment in a python script (Python 3.8.11, Numpy 1.20.3, Scipy 1.7.1). For each ribosome we determined distance vectors between itself and its $n$ closest neighbours ($n = 4$), excluding neighbours further than 100 Å. The vectors were rotated with the inverse orientation of the respective ribosome, resulting in the coordinates of neighbours in the coordinate system of an ER-bound ribosome with the $xy$ plane corresponding to the ER membrane. These vectors were sampled on a 3D-histogram with voxels corresponding to $15^3$ Å$^3$ and divided by the total number of analysed neighbours to indicate the probability of finding a neighbouring ribosome particle in each voxel. The plots were projected on the $xy$ plane to visualize the density of neighbours surrounding ER-bound and soluble ribosomes.

A threshold was chosen to identify clusters for trailing and leading neighbours. For ER-bound neighbours a binary mask was created in the 3D-histogram above a probability of $P = 0.0005$, while for soluble ribosomes the threshold was put at $P = 0.0003$. Both masks were dilated by two voxels. The soluble and ER-bound trailing masks were combined in a trailing mask for the whole dataset, and the same procedure was performed for the leading mask. The masks were used to annotate associations of ribosome pairs in a polysome. A trailing–leading connection was confirmed if the neighbour localized in the trailing–leading mask area and the analysed ribosome also positioned in the leading–trailing area of the respective neighbour (that is, the inverse calculation).

The trailing/leading states of neighbours were used in R to fit a multinomial mixed-effects logistic regression model (mclogit 0.9.4.2[67] in R 3.6.1). The ribosome's state was used to predict probabilities of leading and trailing states, where the tomogram index was used as a random effect to account for sample and imaging variation. We used the same model to predict probabilities of translation states in polysome chains. For visualization, the probabilities were extracted with their 95% confidence interval, representing the region of 95% certainty that the modelled mean is the population mean. Variation between tomograms was shown by calculating the frequency of certain events per tomogram—for example, the 42nd tomogram might have 7 pre+ ribosomes of which 6 are associated in polysomes resulting in a frequency of 0.86. Random association probability was calculated by fractional abundance of each state in the dataset. For the plots showing the fold increase, the modelled mean and confidence interval lower and upper bounds were divided by the random association probability and displayed with logarithmic $y$-axis. Statistical significance for the fitted logistic parameters was determined with a two-sided Wald-test (as reported by mclogit) and used to annotate plots. $P$ values were adjusted for multiple comparisons with the Hochberg method as implemented in R with p.adjust (method='hochberg').

### Previously published data
We made use of previously published atomic models from the PDB (accession codes 5AJO, 4CXG, 4UJE, 6Y0G, 6Y57, 6GZ5, 6Z6L, 6Z6M, 5LZS, 4C0S, 5LZT, 5IZK, 6O85, 5LZZ, 6GZ3, 6GZ4, 6GZ5, 6SXO, 1BN5, 6W6L, 6ENY, 6S7O, 3JC2) and the AlphaFold Protein Structure Database (AF-O00178, AF-P30101). Moreover, we used the following EM densities from the EMDB for analyses: EMDB-2904, EMDB-2908.

### Reporting summary
Further information on research design is available in the Nature Portfolio Reporting Summary linked to this article.

## Data availability
Data generated in this study have been deposited at the Electron Microscopy Data Bank (www.ebi.ac.uk/emdb) under accessions EMD-15870, EMD-15871, EMD-15872, EMD-15873, EMD-15874, EMD-15875, EMD-15876, EMD-15877, EMD-15878, EMD-15879, EMD-15880, EMD-15884, EMD-15885, EMD-15886, EMD-15887, EMD-15888, EMD-15889, EMD-15890, EMD-15891, EMD-15892, EMD-15893 and the Protein Data Bank (www.rcsb.org) under accessions 8B6Z and 8B6L. The mass spectrometry proteomics data have been deposited to the ProteomeXchange Consortium via the PRIDE[68] partner repository with the dataset identifier PXD035475.

## Code availability
Python code for polysome analysis is available at https://github.com/McHaillet/polysome-stats.

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

**Acknowledgements** This work was supported by the European Research Council under the European Union's Horizon2020 Program (ERC Consolidator Grant Agreement 724425–BENDER) and the Nederlandse Organisatie voor Wetenschappelijke Onderzoek (Vici 724.016.001 to F.F., Veni 212.152 to J.F., research program TA 741.018.201 to R.A.S. and F.F., and National Roadmap for Large-Scale Research Infrastructure NEMI 184.034.014). We thank G. van der Schot, M. Vanevic and R. Englmeier for help with data processing; R. Hermsen for advice on statistical analysis; and S. Pfeffer, B. Janssen, M. Feige, E. Schmitt, Y. Mechulam, S. Lang and R. Zimmermann for stimulating discussions and critical comments on the manuscript.

**Author contributions** M.G., J.F. and F.F. conceived the project. M.G. performed microsome sample preparation, cryo-ET data acquisition and image analysis. M.G. and J.F. carried out model building. M.L.C. analysed polysomes. J.v.L., S.C.H. and J.F. performed in situ cryo-ET of human cells. R.C.A. and M.G.-M. cloned CRISPR–Cas9 constructs and generated monoclonal cells. D.V., M.G., P.A. and R.A.S. analysed samples using mass spectrometry. J.F. performed cryo-EM SPA sample preparation and data processing, F.A.K. and A.K. collected cryo-EM SPA data. M.G., M.L.C., J.F. and F.F. analysed the data and wrote the manuscript.

**Competing interests** A.K. and F.A.K. are employees of Thermo Fisher Scientific. The other authors declare no competing interests.

**Additional information**
**Correspondence and requests for materials** should be addressed to Juliette Fedry or Friedrich Förster.

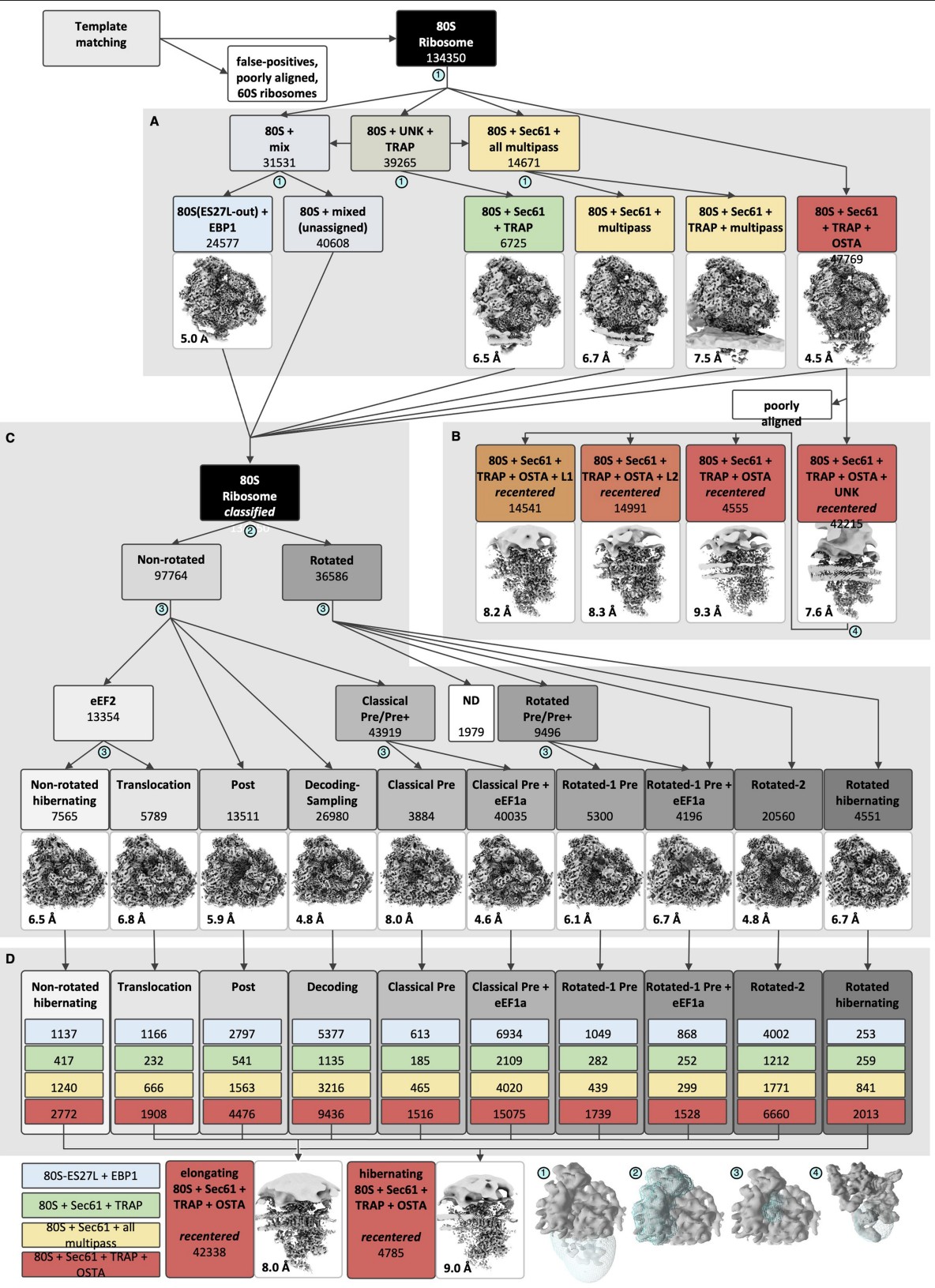

**Extended Data Fig. 1** | See next page for caption.

**Extended Data Fig. 1 | Cryo-ET data analysis workflow.** Template matching in PyTom[69] generates candidates for ribosomal particles, which are further analyzed in RELION[57] and M[17]. Initial coarse 3D classification allowed removal of false positives, poorly aligned particles, and isolated LSUs. (**A**) The remaining ~135,000 80S ribosome subtomograms were subjected to focused classification on the area at the ribosomal tunnel exit (mask 1). Repeated classification is required to distinguish subtle differences of Sec61-multipass-, Sec61-multipass-TRAP translocon, and Sec61-TRAP. (**B**) The center of the reconstruction of the ribosome-Sec61-TRAP-OSTA population was shifted to the center of the translocon. After refinement, recentered subtomograms were subjected to 3D classification focused on a luminal mask near OSTA (mask 4). (**C**) To obtain the best statistics for analysis of ribosomal processing states all subtomograms were pooled again. The particles were hierarchically classified, first according to the rotation state of the SSU (mask 2) and then further focused using masks including the tRNA and eEF binding sites (mask 3). A minor population of <2k particles could not be assigned unambiguously to a translation state (ND = not defined). (**D**) Previously annotated particles from classification focused on the translocon (**A**) were extracted from classes obtained by classification of ribosomal intermediate states (**C**).

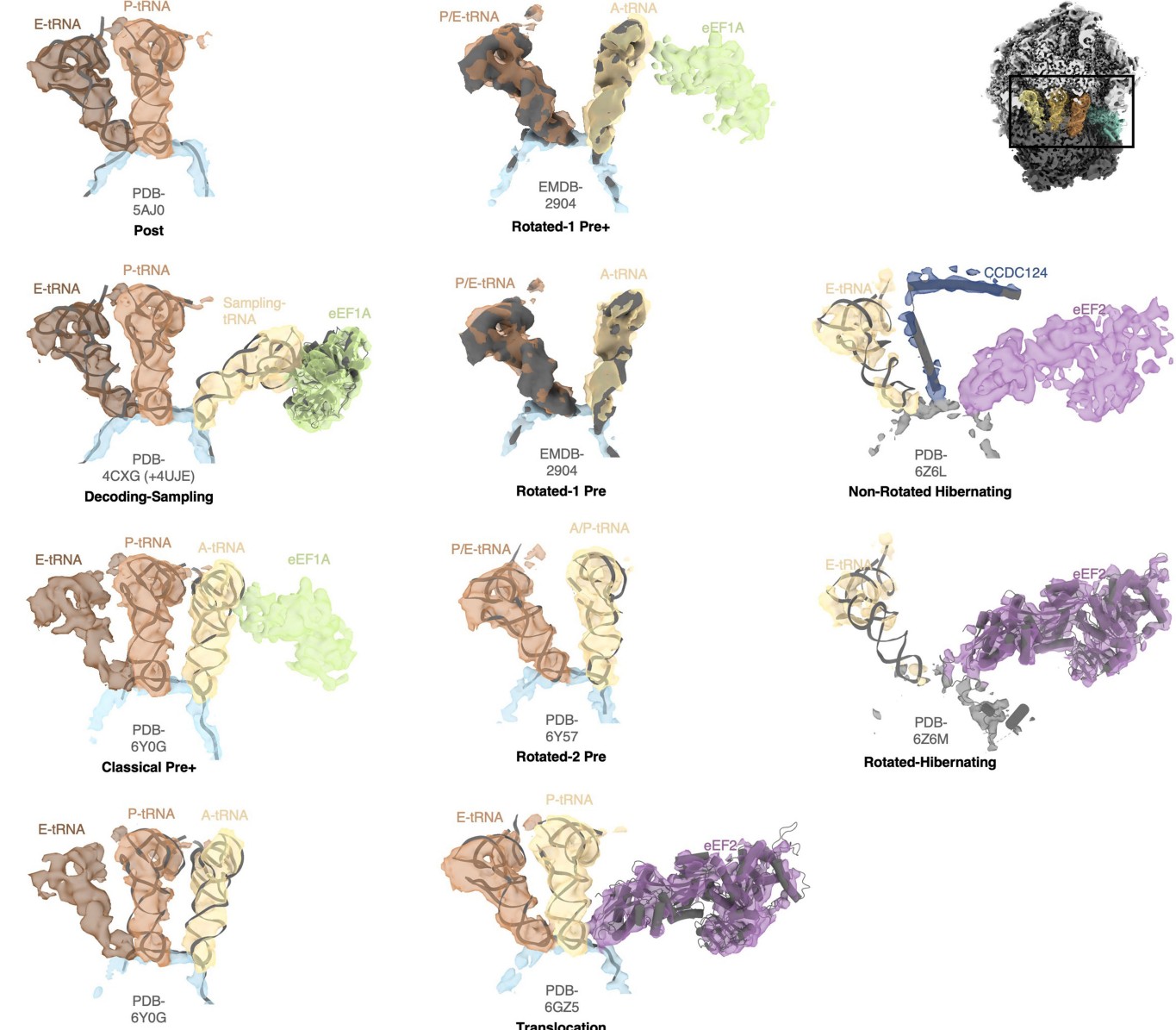

**Extended Data Fig. 2 | Identification of ribosomal intermediate states.**
Large ribosomal subunits of models or maps of previously characterized
intermediate states were fitted into our reconstructions from Fig. 1A, of which
we only show the tRNAs and elongation factors for clarity. Structures of
mRNAs, tRNAs, elongation factors and the small ribosomal subunit from the
models indicated by their PDB or EMDB codes are superposed onto the
respective segmented densities from our reconstructions.

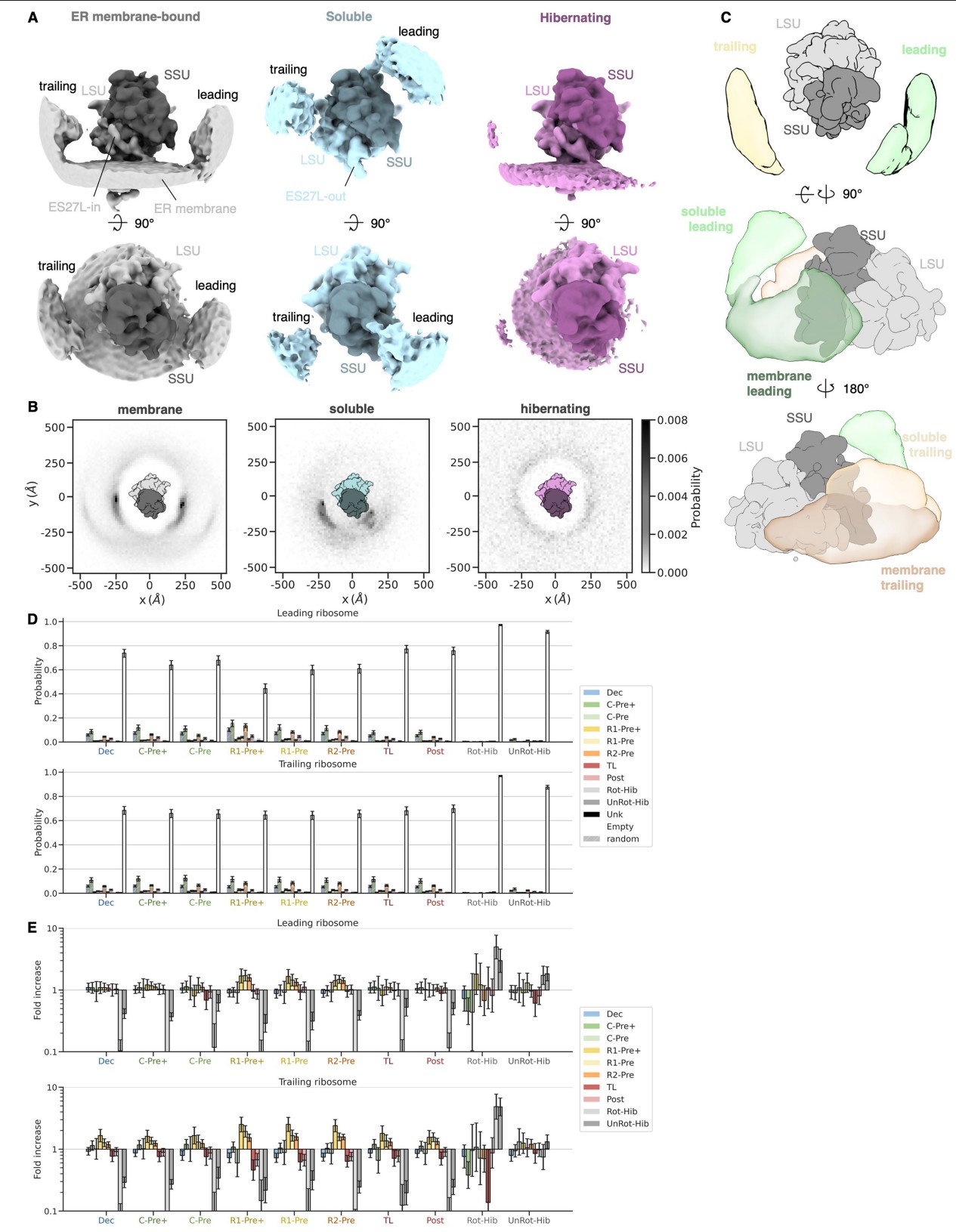

**Extended Data Fig. 3** | See next page for caption.

**Extended Data Fig. 3 | Neighborhood analysis of ER membrane-bound and soluble ribosomes and their intermediate states.** (**A**) Side view (top panels) and top view (bottom panels) of filtered reconstructions of ER-membrane bound, soluble and hibernating ribosome populations depicted at low contour level. Densities of leading and trailing ribosome neighbors are visible adjacent to the centered ribosome. (**B**) Neighborhood analysis illustrates the arrangement of ribosomes and is consistent with the subtomogram averages from (**A**). Neighborhood analysis was performed in 3D, whereas 2D heat maps show the results projected onto a plane parallel to the membrane. (**C**) Masks were generated in 3D from results of the neighborhood analysis of membrane-bound and soluble populations combined. (**D**) Columns represent the modelled mean neighbor probability with 95% confidence interval as error bars analysis based on the neighborhood analysis from (**B,C**) for each ribosomal intermediate state. Statistics determined from n = 132,371 ribosomes with the 869 tomograms included as a random effect. The random association probability (gray hatched bars) is the overall abundance of the ribosome populations. (**E**) Columns represent the mean logarithmic fold increase of observed vs. random probability with 95% confidence interval as error bars of the data from (**D**).

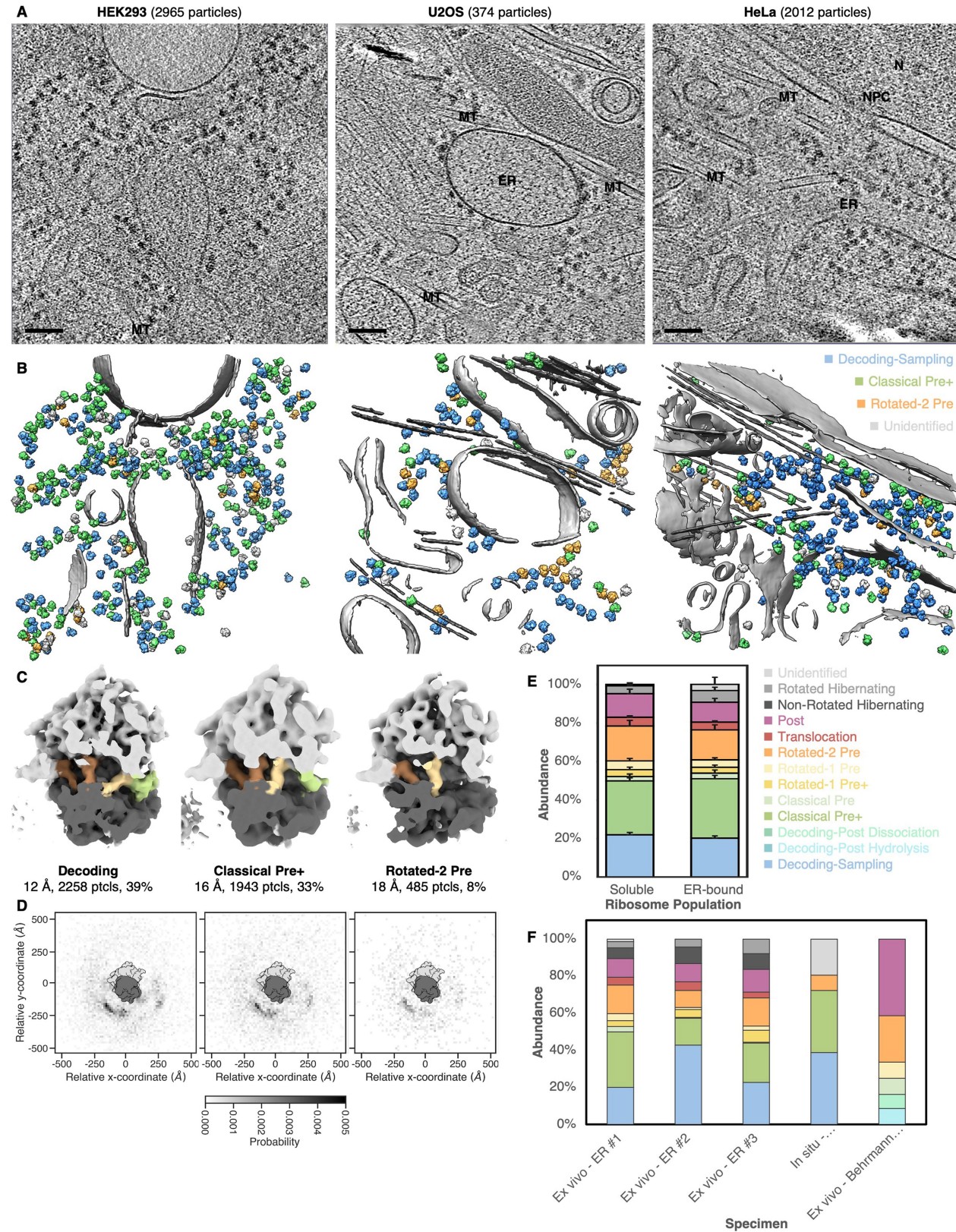

**Extended Data Fig. 4** | See next page for caption.

**Extended Data Fig. 4 | Ribosome states *in situ* and comparison to *ex vivo* abundances.** (**A**) Central slice (thickness 1.7 nm) of representative tomograms of cryo-FIB milled HEK293, U2OS and HeLa cells. Scale bar: 100 nm. (**B**) Segmented representation of tomograms from (**A**). Subtomogram averages of the ribosome were mapped back into the reconstruction and color-coded according to their ribosomal state. (**C**) Ribosomal states obtained by 3D classification of *in situ* data. (**D**) Neighborhood analysis of the intermediate states from (**C**). (**E**) Distribution of ribosomal states from soluble or membrane-bound ribosomes. Statistics determined from n = 132,371 ribosomes with 869 tomograms modeled as random effect. Stacked columns show the modelled mean with the 95% confidence interval as error bars. (**F**) Distribution of ribosomal states from 3 separate ER vesicles preparations (*ex vivo* - ER #1-3), *in situ* data, and cytosolic polysomes from Behrmann et al[21]. n(ER #1) = 132,731 particles in 869 tomograms, n(ER #2) = 6,101 particles in 31 tomograms, n(ER #3) = 3,836 particles in 58 tomograms, each from 1 experiment, n(in situ) = 5,351 (HEK293 = 2,965, U2OS = 374, HeLa = 2,012) particles in 27 tomograms from 3 independent experiments.

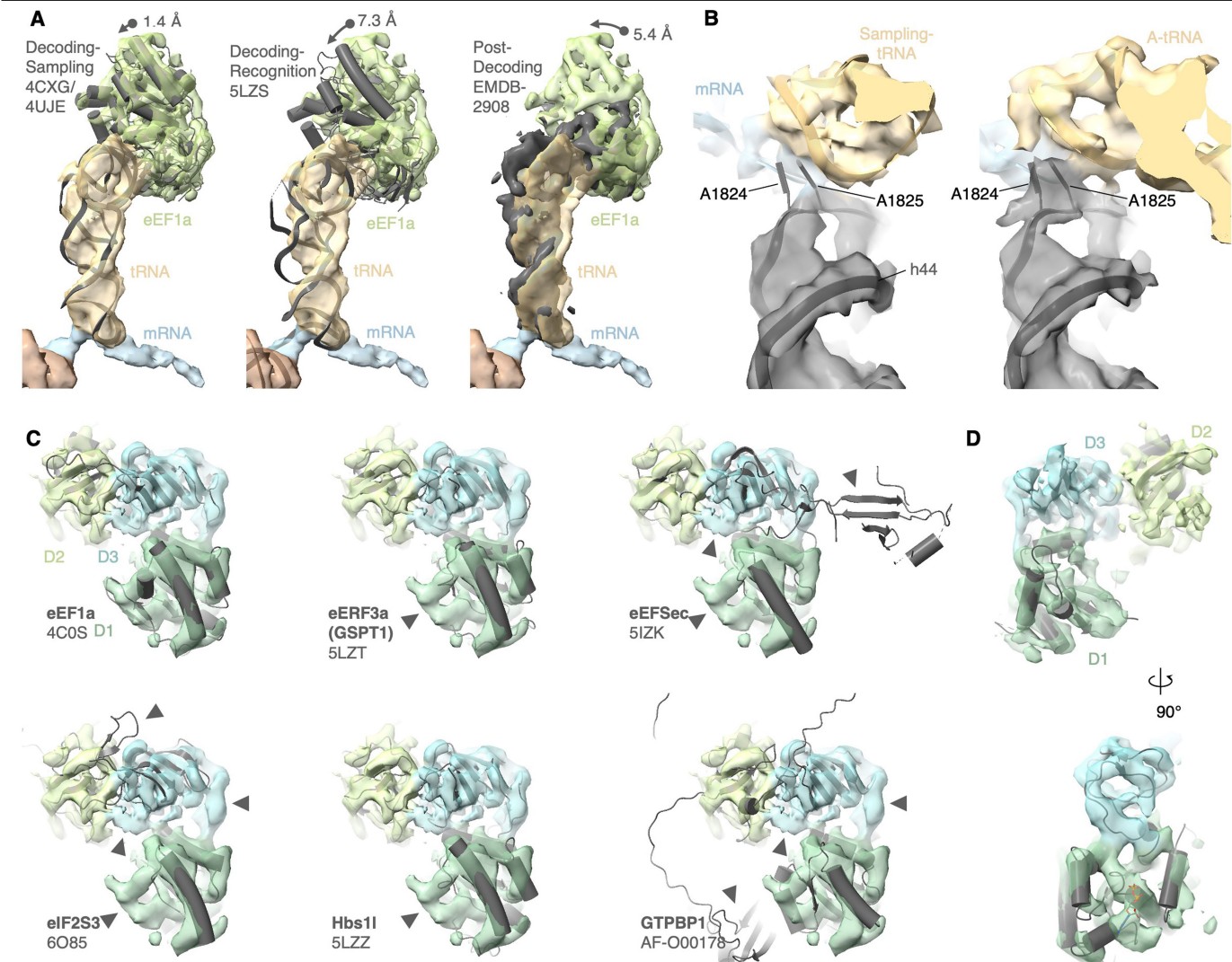

**Extended Data Fig. 5 | Identification of elongation factor-bound ribosomal intermediate states.** (**A**) Superposition of the decoding-sampling (4CXG+4UJE), decoding-recognition (5LZS) and post-decoding (EMDB-2908) state (dark grey cartoon representations) onto our reconstruction (semi-transparent colored maps) of the ribosome-bound eEF1a-tRNA ternary complex. Arrows indicate structural differences. (**B**) Close-up of the decoding center of the decoding-recognition state (5LZS) superposed onto our segmented reconstructions (semi-transparent maps) of our decoding state (left) or the subsequent classical pre state (right) for comparison. Densities of the nucleobases A1824 and A1825 are clearly visible in the flipped-out conformation in the classical pre state (right) but flipped-in in the decoding state (left), indicating that tRNA recognition has not yet occurred. tRNA, mRNA, and 18S rRNA segment h44 were segmented and tRNAs were clipped for better overview. (**C**) Comparison of eEF1a and structurally related candidates fitted into the segmented density of the classical pre+ state. Arrowheads indicate structural differences. (**D**) Structure of eEF1A in extended conformation (4C0S) fitted into the segmented density of the classical pre+ state. Domain 1, 2 and 3 (D1-3) were fitted individually.

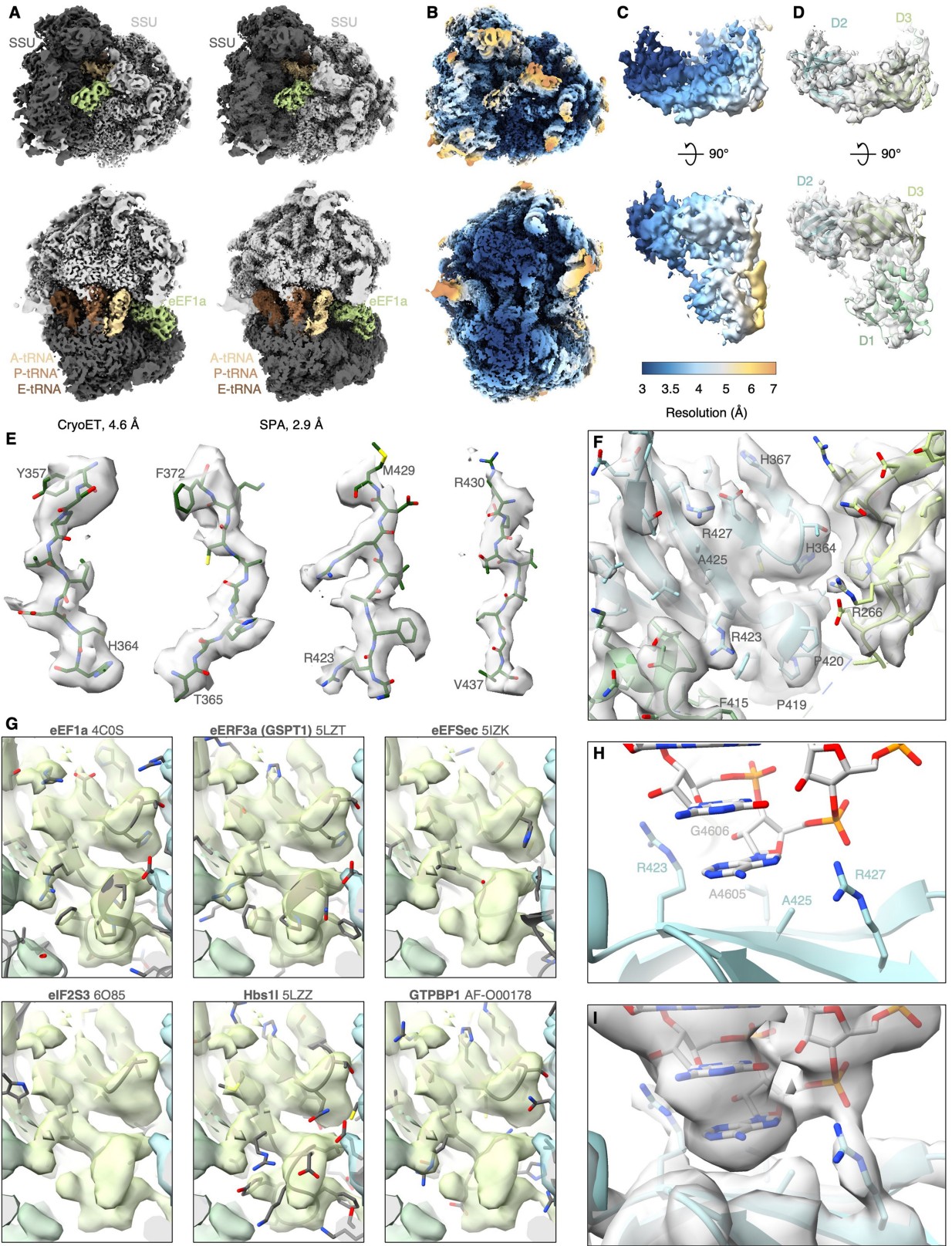

**Extended Data Fig. 6 | Single particle analysis of the ribosome in the classical pre+ state.** (**A**) Comparison of cryo-ET and SPA reconstructions of the ribosome in the classical pre+ state filtered to local resolution. Ribosomes were clipped in top views (bottom panels). (**B**) SPA reconstruction color-coded according to local resolution. (**C**) Close-up view of eEF1a color-coded according to local resolution explained in the color bar. (**D**) Refined atomic model of eEF1a placed into the SPA density map. Domains 1-3 (D1-3) are indicated. (**E**) Segments of eEF1a superposed on density maps with well-resolved side chains. (**F**) Refined model of eEF1a fitted into the locally refined reconstruction of domain 3. The SRL is not depicted for clarity. (**G**) Candidate GTPases fitted into the high-resolution density. The SRL binding site of domain 3 is displayed. (**H**) Interaction site of eEF1a with the SRL of the LSU. (**H**) Same view as in (**H**) with the density map.

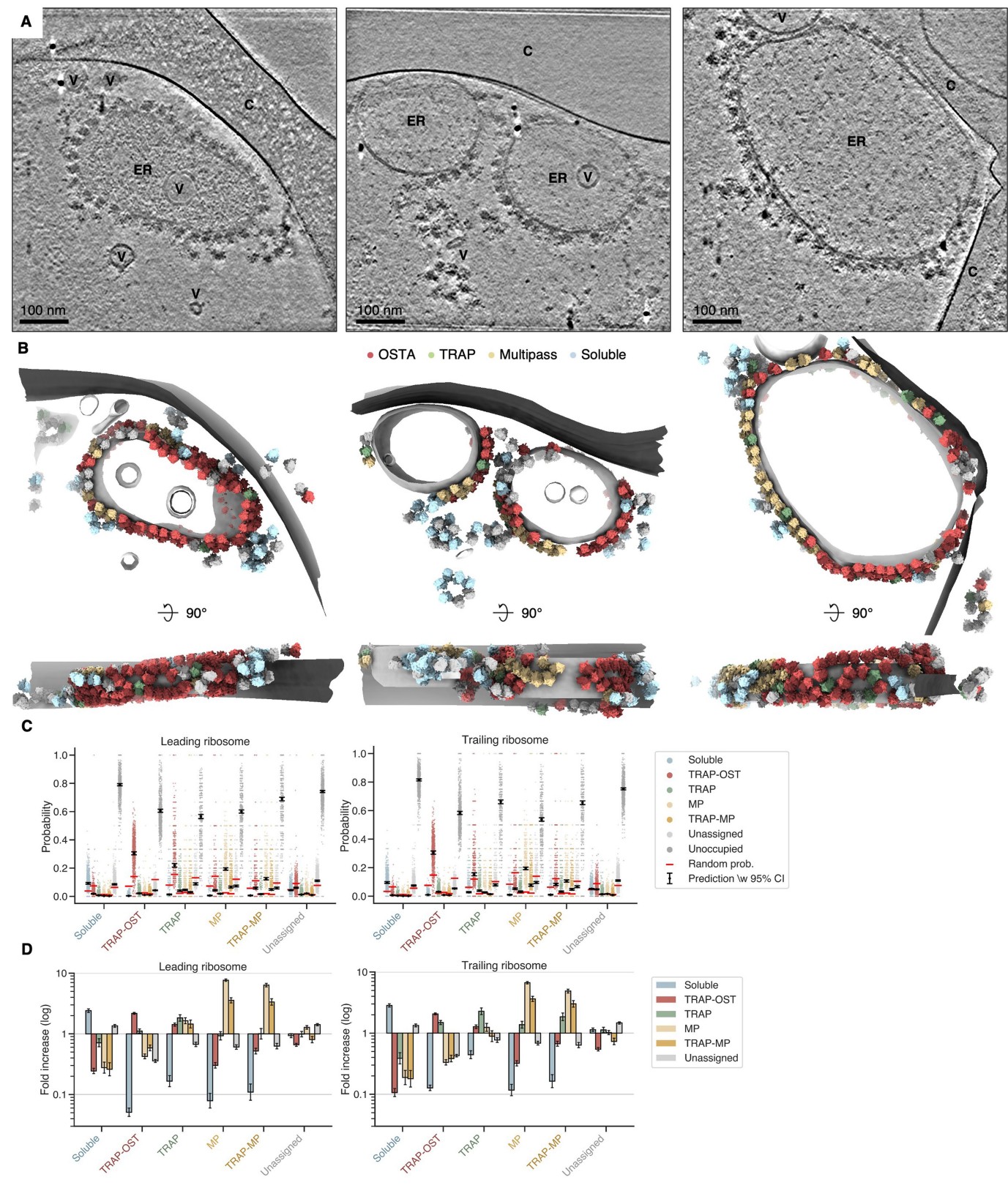

**Extended Data Fig. 7 |** See next page for caption.

**Extended Data Fig. 7 | Neighbor probability analysis of soluble and ER translocon populations.** (**A**) Central slices from representative filtered tomograms of ER-derived vesicles. ER (endoplasmic reticulum), V (vesicle), C (carbon support). (**B**) Segmented representation of tomograms from (**A**), including the ER membrane (grey), carbon support (black) and subtomogram averages of different ribosome populations mapped back into the tomogram. Ribosomes are color-coded according to their binding partners at the exit tunnel: soluble (blue), OSTA-translocon (red), TRAP-translocon (green), multipass-translocon (yellow), unassigned (grey); large ribosomal subunit (LSU, lighter shade), small ribosomal subunit (SSU, darker shade). (**C**) Probability of encountering soluble or ER-associated ribosomes from as leading or trailing neighbor. The black circles show the modelled mean with the 95% confidence interval as error bars fitted to n = 134,350 ribosomes with the 869 tomograms included as a random effect. The small scattered points represents the frequencies of events per tomogram. The random association probability (bright red lines) is the overall abundance of the ribosome populations corrected for unoccupied positions. Neighbors are defined as 'unoccupied' if there is no particle in the defined neighborhood mask or its potential neighbor (e.g., a particle must have a trailing neighbor, which has this particle as a leading neighbor). (**D**) Columns represent the mean fold increase of observed vs random probability with 95% confidence interval as error bars of the data from (**C**).

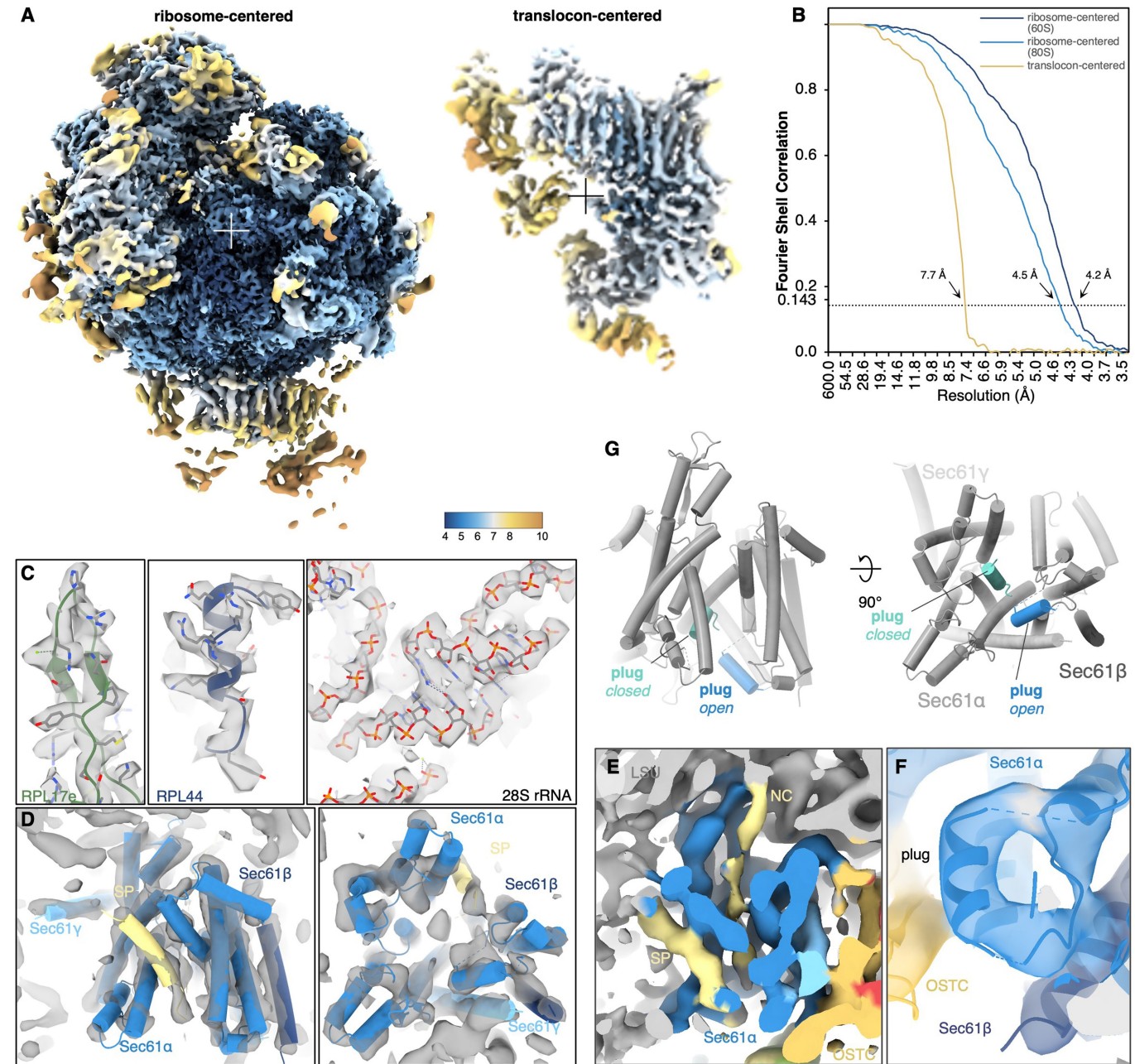

**Extended Data Fig. 8 | Reconstruction of the Sec61-TRAP-OSTA-translocon.**
(**A**) Ribosome- and translocon-centered reconstruction of the ribosome-Sec
61-TRAP-OSTA-translocon color-coded by local resolution (color bar in Å).
Centers of the respective reconstructions are indicated. (**B**) FSC curves of the
ribosome- and translocon-centered reconstructions of the ribosome-Sec
61-TRAP-OSTA-translocon. (**C**) Examples of 60S ribosomal proteins and 28S
rRNA fitted into the ribosome-centered reconstruction filtered to local
resolution of up to 3.5-Å. (**D**) Cryo-EM structures of Sec61 (3JC2) fitted into the
translocon-centered reconstruction. (**E**) Density of the nascent chain (NC,
light-yellow) is visible at the ribosomal tunnel exit, the Sec61 pore and in the
lateral gate as signal peptide (SP, light-yellow). The front side of the ribosome
and membrane were clipped for visualization purposes. (**F**) Close-up of the
Sec61 plug placed into the density of the translocon-centered reconstruction.
(**G**) Superposition of the plug in the closed (cyan, 3J7Q) and open (blue)
conformation.

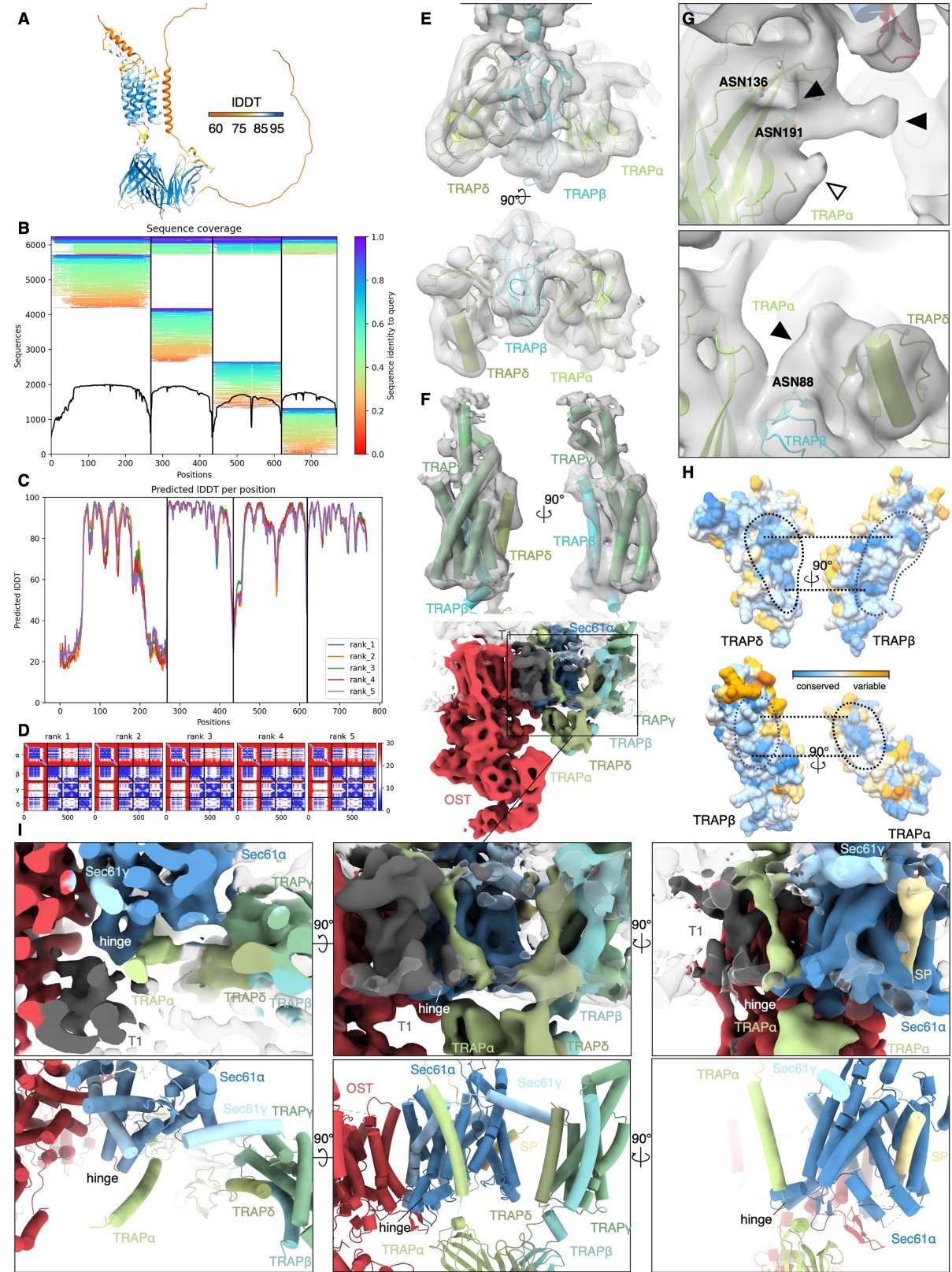

**Extended Data Fig. 9** | See next page for caption.

**Extended Data Fig. 9 | Model building of the TRAP complex.** (**A**) Prediction model of TRAP (P43307, P43308, Q9UNL2, P51571) obtained by Colabfold (v1.4)[70] using MMseqs2 and Alphafold2-multimer (v2)[62] color-coded according to predicted local distance difference test (pLDDT) score. Signal peptides were removed prior to prediction. (**B**) Sequence coverage obtained by sequence alignments generated by MMseqs2. (**C**) pLDDT scores per position of five model predictions. (**D**) Predicted aligned error (PAE) of five model predictions. (**E**) Prediction models of TRAPαβδ placed into the density of the locally filtered translocon-centered reconstruction. (**F**) Alphafold models of TRAPβγδ placed into the segmented density of the locally filtered ribosome-centered reconstruction. (**G**) Additional densities which are not explained by the prediction models reside near disordered terminal regions (white arrowhead) or glycosylation sites of TRAPαβ indicating partially ordered glycans (black arrowheads). Asparagine residues are displayed as ball/stick models and annotated according to residue number. (**H**) Sequence conservation score plotted onto the surface of TRAP subunits (blue: high conservation, orange: low conservation). Evolutionary conserved residues reside primarily at the interface areas, whereas peripheral residues are variable. The luminal TRAPα, TRAPβ, and TRAPδ domains possess large interaction interfaces (TRAPα-TRAPβ: 695 Å$^2$, TRAPβ-TRAPδ: 985 Å$^2$). (**I**) Top, back and side view of the reconstruction of the Sec61-TRAP-OSTA-translocon (top panels). Semi-transparent densities originate from residual membrane signal. Models generated from the density map at the same view (bottom panels).

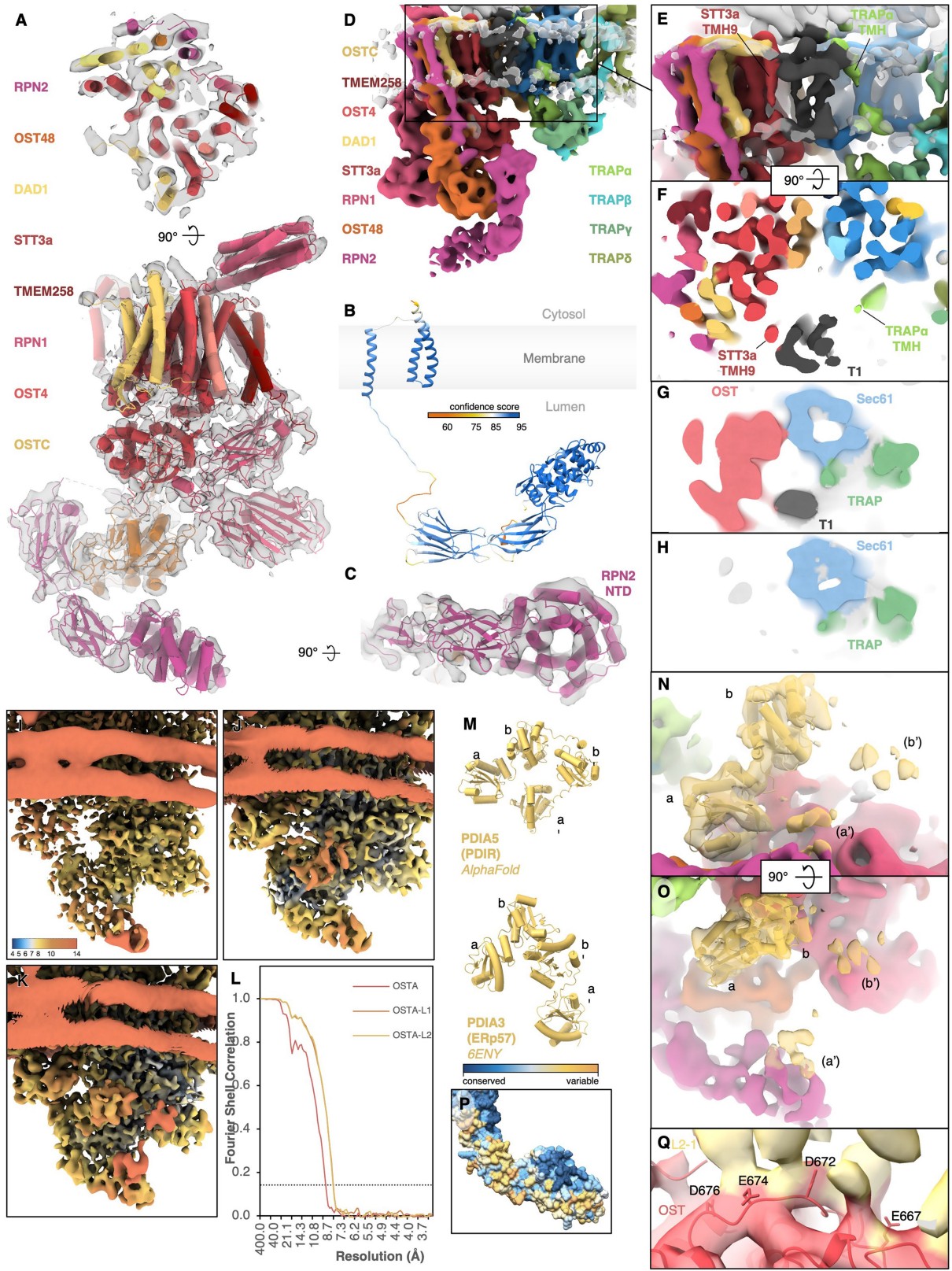

**Extended Data Fig. 10** | See next page for caption.

**Extended Data Fig. 10 | Native OSTA and its accessory factors.** (**A**) View from cytosol (top) and side view (bottom) of the OSTA complex (PDB 6S7O, AlphaFold P04844) fitted into the segmented map of the translocon-centered reconstruction of the OSTA-translocon. (**B**) AlphaFold model of RPN2 (P04844). The model is color-coded according to confidence score as indicated. (**C**) Close-up view of the N-terminal domain (NTD) of the RPN2 prediction model fitted into the reconstruction as in (A). (**D**) Side view of the OSTA-translocon opposite to the lateral gate. (**E,F**) Close-up side view (**E**) and top view from the cytosol (**F**) of T1 intercalated between TMHs of STT3a and TRAPα. (**G,H**) Membrane-resident translocon components (same view as in (**F**)) of the ribosome-centered reconstructions of the Sec61-TRAP-OSTA-translocon (**G**) and the Sec61-TRAP-translocon (**H**) filtered to a resolution of 15 Å. (**I–K**) Reconstructions of the OSTA-translocon without (**I**) or with accessory factor L1 (**J**) or L2 (**K**) color-coded according to local resolution as indicated. (**L**) FSC curves of the reconstructions from (**I–K**). (**M**) Models of L2-candidate proteins PDIA3 (6ENY) and PDIA5 (Q14554). Catalytic (a, a') and non-catalytic (b, b') thioredoxin domains are indicated. (**N,O**) PDI domains a and b fitted into the reconstruction of OSTA-L2. (**P**) Sequence conservation plotted onto the surface model of the RPN2 NTD. Highly conserved residues reside at the binding site of the a'-domain of PDI or other OST subunits. (**Q**) Close-up view of the interaction site of STT3A and L2-1.

# Reporting Summary

## Statistics

For all statistical analyses, confirm that the following items are present in the figure legend, table legend, main text, or Methods section.

| n/a | Confirmed | |
|---|---|---|
| ☐ | ☒ | The exact sample size (*n*) for each experimental group/condition, given as a discrete number and unit of measurement |
| ☐ | ☒ | A statement on whether measurements were taken from distinct samples or whether the same sample was measured repeatedly |
| ☐ | ☒ | The statistical test(s) used AND whether they are one- or two-sided<br>*Only common tests should be described solely by name; describe more complex techniques in the Methods section.* |
| ☐ | ☒ | A description of all covariates tested |
| ☒ | ☐ | A description of any assumptions or corrections, such as tests of normality and adjustment for multiple comparisons |
| ☐ | ☒ | A full description of the statistical parameters including central tendency (e.g. means) or other basic estimates (e.g. regression coefficient) AND variation (e.g. standard deviation) or associated estimates of uncertainty (e.g. confidence intervals) |
| ☐ | ☒ | For null hypothesis testing, the test statistic (e.g. $F$, $t$, $r$) with confidence intervals, effect sizes, degrees of freedom and $P$ value noted<br>*Give P values as exact values whenever suitable.* |
| ☒ | ☐ | For Bayesian analysis, information on the choice of priors and Markov chain Monte Carlo settings |
| ☒ | ☐ | For hierarchical and complex designs, identification of the appropriate level for tests and full reporting of outcomes |
| ☒ | ☐ | Estimates of effect sizes (e.g. Cohen's *d*, Pearson's *r*), indicating how they were calculated |

*Our web collection on statistics for biologists contains articles on many of the points above.*

## Software and code

Policy information about availability of computer code

Data collection | Tilt series were acquired using SerialEM 3.8 and GMS 2.3. Single particle cryo-EM data were acquired using EPU 3.

Data analysis | For subtomogram averaging and classification we used the following software: Warp 1.0.9, M 1.0.9, Relion 3.1.1, PyTOM 0.994, IMOD 4.10.25. For segmentation we used Eman2 2.91 and for visualization Chimera 1.14.0, ChimeraX 1.3.0. To analyze polysomes we used an in-house developed python package (https://github.com/McHaillet/polysome-stats) that made use of mclogit 0.9.4.2, R 3.6.1, Python 3.8.11, Numpy 1.20.3 and Scipy 1.7.1.
For cryo-EM single particle analysis we used Relion 3.1.1. For atomic model building and assessment we used Isolde 1.0b5, Phenix 1.20.1, Coot 0.9.8.2, Imodfit 1.51, Molprobity 4.5.1, and findMySequence (https://gitlab.com/gchojnowski/findmysequence, not versioned), Mass-spec analysis was performed using MaxQuant 2.0.1.0.

For manuscripts utilizing custom algorithms or software that are central to the research but not yet described in published literature, software must be made available to editors and reviewers. We strongly encourage code deposition in a community repository (e.g. GitHub). See the Nature Portfolio guidelines for submitting code & software for further information.

## Data

Policy information about availability of data

All manuscripts must include a data availability statement. This statement should provide the following information, where applicable:

- Accession codes, unique identifiers, or web links for publicly available datasets
- A description of any restrictions on data availability
- For clinical datasets or third party data, please ensure that the statement adheres to our policy

Data generated in this study are available in the main article, supplementary materials or in public repositories: nos. EMD-15870, EMD-15871, EMD-15872, EMD-15873, EMD-15874, EMD-15875, EMD-15876, EMD-15877, EMD-15878, EMD-15879, EMD-15880, EMD-15884, EMD-15885, EMD-15886, EMD-15887, EMD-15888, EMD-15889, EMD-15890, EMD-15891, EMD-15892, EMD-15893 of EMDB (www.ebi.ac.uk/emdb) and PDB-8B6Z, PDB-8B6L of PDB (www.rcsb.org). The mass spectrometry proteomics data have been deposited to the ProteomeXchange Consortium via the PRIDE68 partner repository with the dataset identifier PXD035475.
In addition, we made use of a previously published atomic models from the PDB (accession codes 5AJO, 4CXG, 4UJE, 6Y0G, 6Y57, 6GZ5, 6Z6L, 6Z6M, 5LZS, 4C0S, 5LZT, 5IZK, 6O85, 5LZZ, 6GZ3, 6GZ4, 6GZ5, 6SXO, 1BN5, 6W6L, 6ENY) and the AlphaFold Protein Structure Database (AF-O00178, AF-P30101). Moreover, we used the following EM densities from the EMDB for analyses: EMDB-2904, EMDB-2908.

# Field-specific reporting

Please select the one below that is the best fit for your research. If you are not sure, read the appropriate sections before making your selection.

☒ Life sciences          ☐ Behavioural & social sciences          ☐ Ecological, evolutionary & environmental sciences

For a reference copy of the document with all sections, see nature.com/documents/nr-reporting-summary-flat.pdf

# Life sciences study design

All studies must disclose on these points even when the disclosure is negative.

| | |
|---|---|
| Sample size | No sample size calculation was performed. Cryo-EM structures were determined from a single sample based on 869 tilt series containing 134,350 particles. Target for cryo-ET subtomogram analysis was at least 100,000 particles, surpassing previous analysis by one order of magnitude allowing for better classification. The rationale for 100,000 particles was that a class representing 1% of intermediates would contain 1,000 particles, which is sufficient to obtain sub-nanometer resolution. Classification was repeated for 2 independent samples (see Replication). MS analysis was performed in technical replicates from one sample. |
| Data exclusions | Tiltseries from thick samples were excluded due to poor signal-to-noise ratio. |
| Replication | Microsome preparation and translocon analysis has been repeated twice from different cell batches with lower acquisition statistics (Extended Figures 4F). The two replicates comprised 31 tomograms (6,101 particles) and 69 tomograms (3,836 particles), respectively. The same translation intermediates were detected in replicates. |
| Randomization | Randomization was not performed for cryo-EM analysis as there is nothing to randomize. |
| Blinding | Blinding is not technically feasible for structure determination. |

# Reporting for specific materials, systems and methods

We require information from authors about some types of materials, experimental systems and methods used in many studies. Here, indicate whether each material, system or method listed is relevant to your study. If you are not sure if a list item applies to your research, read the appropriate section before selecting a response.

## Materials & experimental systems

| n/a | Involved in the study |
|---|---|
| ☐ | ☒ Antibodies |
| ☐ | ☒ Eukaryotic cell lines |
| ☒ | ☐ Palaeontology and archaeology |
| ☒ | ☐ Animals and other organisms |
| ☒ | ☐ Human research participants |
| ☒ | ☐ Clinical data |
| ☒ | ☐ Dual use research of concern |

## Methods

| n/a | Involved in the study |
|---|---|
| ☒ | ☐ ChIP-seq |
| ☒ | ☐ Flow cytometry |
| ☒ | ☐ MRI-based neuroimaging |

## Antibodies

| | |
|---|---|
| Antibodies used | anti-Sec61alpha (Abcam, ab15575, polyclonal; dilution: 1:1000), anti-SSR3 (Sigma Aldrich, hpa014906, polyclonal; dilution: 1:1000), anti-CCDC47 (Abcam, ab241608, polyclonal; dilution: 1:1000). |
| Validation | anti-Sec61alpha: validated by WB of murine dendritic cells (DOI: 10.4049/jimmunol.1302312), anti-SSR3: validated by WB of human A549 cells (DOI: 10.1126/sciadv.abc6364), anti-CCDC47: validated by WB of HEK-293T whole lysate (manufacturer) |

## Eukaryotic cell lines

Policy information about cell lines

| | |
|---|---|
| Cell line source(s) | FreeStyle 293-F cells (ThermoFisher Scientific, R79007), U2OS and HeLa cell originated from ATCC (CVCL_0042 and CVCL_0030 in Cellosaurus.org, respectively) |
| Authentication | Cell lines were not authenticated |
| Mycoplasma contamination | Cell lines tested negative for Mycoplasma contamination |
| Commonly misidentified lines (See ICLAC register) | No commonly misidentified cell lines were used |

