## [Peer Review File · Nature]

Manuscript Title: Visualization of translation and protein biogenesis at the ER membrane

Reviewer Comments & Author Rebuttals

Reviewer Reports on the Initial Version:

Referees' comments:

Referee #1:

The manuscript, "Molecular snapshots of translation and protein translocation at the ER membrane", by Gemmer et al., reports exciting developments in biochemistry and in the cryoelectron tomography (cryoET) fields that have enabled structural characterizations of protein synthesis complexes and ribosome-translocon (sec61) complexes localizing to the periphery of rough endoplasmic reticulum (ER) membranes (rough microsomal preparations).

Their preparations were rapidly isolated from roughly 50 million HEK 293-F cells, and cryoET tilt series were collected on a Talos Arctica microscope equipped with a K2 camera. An integration of publicly available and custom-designed softwares was applied to identify, classify and refine approximately 13 conformationally and/or conformationally distinct sub-classes – some that are directly tethered to the membrane via the translocon and some proximal to the membrane but not close enough to be caught in the process of translocation across the membrane. I am not quite sure what holds the latter set of ribosomes near the membrane surface (more discussion on this would be helpful), but it would seem to be the most logical conclusion (given the isolation procedure) that they may be polysome-associated with the ribosomes that are caught in the act of secretion through the Sec61 complex. Is this what the authors think? Are all the ribosomes monosomes? Is there any evidence of polysomal structure? How do the authors explain their statement that microsomes are known to harbor "hibernating", i.e. non-active, ribosomes? Does the evidence for this derive from ref. 27 or some other source? From what type of functional complex does a hibernating ribosome come from – elongating ribosomes?

Moving ahead, it seems that a main thrust of the present work is that the speed of the isolation resulted in the authors' capture of a greater number of functional ribosome complexes (less hibernating and other non-functional complexes) in the act of translocating secretory proteins through and into ER membranes. Hence, the present work may be expected to better capture more physiologically relevant snapshots of components than previously described efforts by the Hegde (ref. 26, porcine pancreas-detergent solubilized, cryoEM 3.4A) and Förster (ref 27, dog pancreas lysate, native membranes, cryoET 10A) groups. This increase in functional relevance combined with the two-fold increase in resolution would appear to represent a significant technical milestone.

After giving the presented research full consideration, in the context of earlier work on ostensibly the same system (refs 26 and 27), it seems apparent, however, that there are important considerations that the authors should address prior to publication.

My suggestions and comments in this regard are delineated below. Clearly this is a promising and challenging interdisciplinary frontier that is on an exciting trajectory. Making concrete assessments of the material advancements are, however, very important. The average reader may not catch the nuance. It also seems critical to the scientific process that the authors should acknowledge where uncertainties, potential biases and alternative interpretations may exist.

Major comments:

To address this head on, in the writing of the paper, it would be most helpful to the reader if the authors would show the closest thing to the raw data first (e.g. something akin to Fig. 2b). The authors may also wish to highlight further the complex and massively consolidated set of information that is present in SI Fig. 2, which provides all of the real numbers of each complex for the entire paper. While the conversion to percentages of each complex is certainly helpful in terms of simplified presentation, it seems relevant for the reader to keep in mind that the raw data are comprised of sub-classes, with the stated resolutions and numbers of particles to keep things in context.

How was the cryoET data collected in a manner that is considered high-throughput? If another improvement in beam shift rate or grid hole organization is made to increase throughput another 5- to 10-fold, will that be ultra-high throughput? This doesn't seem properly justified, and it's unclear what this means or how it adds value.

Given the very small number of particles and resolution limits in each class (see SI Fig. 2), in its present form it also seems challenging for the reader to understand that there are uncertainties associated with the classification and sorting of ribosomes into the various sub-classes presented. Something that clarifies and touches on this point would be very instructive and helpful. This is a challenging balance, of course, particularly when working to concisely present the work, but it seems relevant and necessary from my perspective (see additional comments below). Perhaps discussion of these points can be added to the main text and/or put into the supplemental information?

Another major issue that the authors should address is the tenor and tone of some of their interpretations about the elongation cycle and its connections to prior literature on the subject. The authors seem eager to convey that they have assembled a nearly complete translation elongation cycle from their ~100K or so picked and sorted particles (8 classes) at 6-10A resolution that is somehow distinct from what is commonly accepted. Some of the complexes they report have never been seen before or are unexpected based on the field's present understanding. What the reader may take away from the work, if left in its current form, is that the present work overturns canon: in actuality, as the authors themselves emphasize, their sample preparations are an earlier snapshot compared to their previous work of a rapidly degrading sample and therefore extreme caution and proper context should be provided to the reader. After all, their findings may potentially reflect spurious outcomes of an ongoing decay process.

The assigned states and conformational changes are not specifically and quantitatively compared with prior work in a meaningful way. Comparative structures from highly purified and/or reconstituted systems exist. Careful in vitro reconstituted reaction studies with highly active material have proven

vital to our understanding of translation, including the work presented and the authors' interpretations (aside from the speculated departures they infer from their data [more on this below]). While I am sure the authors recognize this, it doesn't currently read as such.

For example, more could be done to appropriately convey to the reader that the authors have inferred the ordering of events from this vast array of prior literature. The work presented in refs 10 and 11 – which the authors seem to hold as a solitary benchmark for mammalian translation studies – is based on an enormous body of prior work, including the major translation-factor bound classes presented. The authors should endeavor to be more complete in their referencing and be concrete about what complexes have been foreshadowed by prior work and what complexes have not been hypothesized previously so that it is clear to the reader. In this regard, it should be noted that the authors state on lines 95-97 that an eEF1A ternary complex in an A/T state, with both E and P sites filled (which they call a “decoding-sampling” state in ref. 11) was not observed in ref. 10, but it clearly was. The decoding-sampling state defined in ref. 11 is also distinct from an A/T state, as the A/T state as defined implies specific contacts with the large subunit GTPase activating center and Sarcin-Ricin loop as well as full engagement by the small subunit decoding site. The decoding-sampling state is defined as highly transient, which is not likely to be what is observed in the present analysis but rather the A/T state. In ref. 10, the authors of that identify an eEF1A-bound A/T state complex (8% occupancy), similar to the one reported here in which domain 1 of eEF1A - the G domain - exhibited weak density. It should be clarified that the present data differ from the prior work only by having more robust density for eEF1A and that one explanation may be their purification strategies, which are likely to be more rapid and thus less prone to degradation. The eEF1A complex, with all three domains fully accounted for, has also been directly isolated from rabbit reticulocyte lysate and the structure determined to roughly 4Å by the Hegde/Ramakrishnan groups using a nascent polypeptide pull-down strategy (Cell, 2016; PDBID:5LZS).

Connections to prior literature also seem important to contextualize the findings against expectation. For instance, atomic models of the “decoding-sampling” (which should be the A/T state) and “Pre+” complexes are shown in Fig. 1 using previously published, partial models of rabbit ribosomes (PDBID:4CXG) and X-ray structures of isolated eEF1A isoform 2 bound to GDP that forms intermolecular crystal contacts (PDBID: 4C0S). In the former case, I think the more relevant model would be PDBID:5LZS. In the latter case, it would seem notable that the putative ribosome-bound eEF1A protein in its “GDP” conformation looks so similar to eEF1A2(GDP) found crystallographically packed as a dimer in the asymmetric unit.

Along similar lines, it seems appropriate to point out the similarities and differences between the relative populations of states observed here and in ref. 10. For instance, it is interesting that no complexes are identified in either work in which eEF2 is bound to its cellular substrate, the pre-translocation complex, although they are present in high abundances: in the present work pre-translocation complexes represent ~60% of the ribosome population evidenced, but in ref. 10 the total pre-translocation complex percentage is roughly 40%. The fast GTP hydrolysis by EF-G argument that is cited to explain this notably unexpected result does not take into consideration that GTP hydrolysis by EF-Tu is reported by the same group to be just as fast as for EF-G, yet the former complex represents 23% of the ribosome population observed. To me this seems striking given the view held by some that EF-G “drives” translocation and recently reported evidence that

the rate limiting step of translocation (in bacteria) is an EF-G(GDP) bound complex (Wasserman et al., NSMB, 2016). Oddly, the authors instead find eEF2 trapped on a “POST+” complex for which eEF2 should have negligible affinity. In addition to noting that similar findings have been reported previously (ref. 10), the authors may wish to further clarify that the POST+ state may actually represent an intermediate state of translocation that happens to closely resemble the POST state.

Another striking departure from canon is that the authors report that 35% of their complexes represent an eEF1A-bound “Classical Pre+” complex in which eEF1A is in a GDP-bound conformation (a model derived from a crystal structure of eEF1A2 appears dimerized in the asymmetric unit; see comment above). According to canon, the switch of eEF1A/EF-Tu(bacteria) is understood to trigger rapid release of the factor from the ribosome as aminoacyl-tRNA release occurs. The relevant research in this area - and the departure from expectation – is not noted by the authors, and they instead misquote a molecular dynamics simulation arguing for a power-stroke mechanism of tRNA selection (see additional comments below).

Do the authors possess any direct mechanistic information regarding the functional relevance or homogeneity of any of the complexes presented? While I would imagine much of the information is correct in its overall interpretation, particularly as it related to tRNA occupancies, ribosome rotations, etc it would seem prudent not to overreach given the present resolution limits and the complexities associated with assigning low-resolution densities to specific factors and factor conformations. It is my understanding that relatively low-resolution data, plus relatively small particle numbers, can result in increased probabilities of mixed sub-populations with multiple conformations and/or compositions. This leaves open the possibility of misinterpretation of states, at least in my view.

Another example of overreach is that the authors do not have the ability to know the nucleotide status (GTP or GDP) of the factors they observe. This point is definitive. Their modeling of two distinct conformations of eEF1A is questionable given what is known about the process and what is presented in the paper as support for this interpretation. The notion that eEF1A undergoes conformational changes on the ribosome as a result of GTP hydrolysis and that this hydrolysis mechanism contributes to the proofreading stage of tRNA selection has been known in bacteria for several decades. As stated above, however, in previous bodies of work, eEF1A/EF-Tu is thought to release from aa-tRNA and the ribosome almost immediately after GTP hydrolysis (which is almost immediately followed by inorganic phosphate release). While taking into consideration that there are some groups presenting evidence/hypotheses that eEF1A/EF-Tu may undergo a change to its GDP-bound conformation while on the ribosome (see work from the M. Ehrenberg, C. Knudsen, K. Sanbonmatsu, B. Cooperman, and S. Blanchard labs) – none of which is appropriately referenced – I have very little if any confidence in the authors' interpretation. The model that such conformational changes “drive” aa-tRNA accommodation during proofreading through a power-stroke mechanism has: 1] not been evidenced or stated to occur as suggested by their citation of ref. 32 (the authors' statement in this regard misrepresents the findings presented in that body of work); 2] is antithetical to proofreading mechanisms that are evidenced to increase fidelity in this step of tRNA selection (as reiterated in ref. 32); and 3] is physically implausible for other reasons (more on this below).

If the authors wish to remain focused on this point, they should consider referencing experimental

evidence that supports this notion. They should specifically address the likelihood that the factor density that they observe may not be eEF1A or contain a nucleotide. They may also wish to present a statistical argument that it is eEF1A and not another factor that takes into consideration their present resolution, map quality with low particle numbers and that they are working with complexes isolated from cellular extract. Significantly more effort will need to be made to convince the reader of this point. From my perspective, there may be numerous small GTPases - or proteolytic fragments thereof - that could be bound to the ribosome in the same position.

Further to the point that their factor density assignment is unlikely to be eEF1A, and something potentially related to a non-functional ribosome sub-class, is the 3% class assigned as "Rotated-1 Pre+" in which they also state that a GDP-bound conformation of eEF1A is found. The possibility that this specific complex is off-pathway should be considered for a number of reasons. eEF1A makes cross-subunit contacts that are specific for the unrotated ribosome (see ref. 10 or Shao et al., Cell, 2016) and the state shown has eEF1A bound to a rotated ribosome conformation for which most known stabilizing contacts are not possible. Moreover, this complex has to have a nascent polypeptide linked to it to be placed in the elongation cycle as it is. To which tRNA is the nascent polypeptide linked in this complex? If it is on the incoming, A-site-bound tRNA, then peptide bond formation would have already occurred and hence there is no reason for translation factor eEF1A to be bound at all. If one alternatively infers the nascent peptide is on the P/E hybrid tRNA somehow, this would mean that the authors are suggesting that the nascent peptide has backtracked out of the peptidyltransferase center by 10-15 amino acids. Both scenarios seem highly implausible given what is presently known about the translation machinery. Is there any supporting evidence for either interpretation? In my view, the authors' assignment of factor density to eEF1A(GDP) in this complex casts further doubt on eEF1A(GDP) assignment in the "Classical Pre+" complex. In the manuscript's current form, the elongation cycle presentation, the aforementioned referencing deficiencies and the over-reaching claims noted appear to weaken the work and should be heavily revised before further consideration.

The translocon/secretory machinery findings appear much more novel, concrete, and hypothesis-generating. It's clear that the authors have spent more time digging through the literature in this area and that they are much more comfortable navigating this important space. This is the intended focus of the work after all it would seem. And it showed. The fitting of models based on AlphaFold structures (which may have issues of their own) into potentially weak and amorphous density does seem like a bit of a risky business in my view, given the potential limitations of this experimental model generation approach and the complexity of partially purified extracts, but this is the vanguard. As for the alternative conformation of eEF1A proposed in the elongation cycle, I would think that erring on the side of caution or providing discussion on alternative models may be appropriate.

Other comments: In Fig. 1, panel e is uninterpretable and doesn't seem to relate to the figure legend.

Referee #2:

This study presents a cryo-ET analysis of rough ER microsomes isolated (relatively rapidly) from cultured cells. Two aspects are analysed: (i) ribosomes and their conformations throughout the translation cycle; and (ii) the ER translocation and glycosylation machinery. One main new solid observation related to each aspect is made. Regarding translation, the authors observe a GDP-bound form of eEF1a in which the tRNA is accommodated into the ribosome. This is used to argue that eEF1a participates in a powerstroke mechanism and may provide a proofreading function. Regarding translocation, the authors characterise the proportions of different flavours of translocon based on the associated components. In addition, they fitted AlphaFold models into their density to generate an atomic model of the most abundant translocon. Intriguingly, several unexplained densities are observed, for which the authors suggest some possible explanations.

The aim of direct visualisation of key molecular processes in their native context is an important goal. It has the value of validating inferences drawn from *in vitro* and reconstituted systems, revealing areas where our understanding may be deficient, and in the best cases, revealing something new that had been previously inaccessible or overlooked. This study makes some progress toward these goals, but is not really “*in situ*” as claimed (see below). The most noteworthy observation is probably the eEF1a-GDP state, whereas the other observations are either modest extensions of earlier work, confusing, or of uncertain biological relevance. For these reasons, the study would seem to be insufficiently developed at this stage for publication in *Nature*. In short, it was very difficult to actually discern the “numerous mechanistic insights advancing beyond the study of isolated components” as claimed by the authors in the abstract.

Specific points, in order of importance:

1. A major issue is the use of “*in situ*” to describe the findings. The moment cells are lysed, anything relating to translation cannot realistically be considered *in situ* because all the translation factors are rapidly diluted by a massive level. Furthermore, salt, ATP, GTP, and other metabolite concentrations are completely altered, and no matter how quickly one does the fractionation, the time frames are enormous relative to that of (bio)chemical reactions. Thus, whatever is observed with ribosomes must be considered somewhere between *in vitro* and *in situ*. They are endogenous (i.e., non-recombinant) components, but it is unclear whether they reflect the situation in intact living cells any better than analysing an *in vitro* translation extract. Indeed, ribosome profiling studies have illustrated that after lysis, ribosomes can rapidly change conformations. The membrane is probably closer to “*in situ*” because membrane proteins are confined to the membrane. Nonetheless, some caution is warranted because various things like ER calcium, ER morphology, membrane curvature, and so forth are altered upon lysis. Whether this impacts what is observed is not clear. For these reasons, I would strongly recommend that the authors not use “*in situ*” to describe the situation that is observed.

2. Much is made of the molecular model of the most abundant ER translocon. As far as I can see, the AlphaFold models of TRAP subunits could have been fitted just as well into earlier cryo-ET or cryo-EM maps, and no additional information is gained from the new cryo-ET analysis. While it is nice to have such a model (thanks to the transformative advance represented by AlphaFold!), the actual

advance would be to provide some new insight into what TRAP actually does. Here, the speculation about influencing Sec61 opening is fairly superficial and there is little to support this idea (see next point). Similarly, the only new part of OSTA is the AlphaFold model of RPN2's N-terminal domains fitted into the map. The relevance of this new information is unclear. Thus, I found it very difficult to see how the authors arrive at the conclusion that this model "reveals the molecular framework for signal peptide (SP) membrane insertion and protein N-glycosylation." If possible, they should be clear about what the new insights actually are beyond conclusions drawn from earlier work.

3. The authors note that the TRAP-alpha TMD "associates with the Sec61-alpha hinge-region (Fig. 3f)," which "may facilitate" gating for weak SPs. A supplementary figure showing the experimental data for the putative interaction between the TRAP-alpha TMD and Sec61 would help the reader judge the claim. More importantly however, no support is provided for the claim that this putative interaction facilitates gating. Such support seems necessary, especially since in the very next sentence they upgrade the claim from a possibility to a fact ("the TRAP architecture positions TRAP-alpha to facilitate SP insertion"). Moreover there is prior reason to doubt this claim; the highest-resolution structures of the 5/6 loop of Sec61 currently available (Braunger et al., 2018; Itskanov et al., 2021) show that this loop doesn't change conformation between the ribosome-primed and the Sec62/63-opened state. Admittedly this mode of opening differs from the ribosome-bound, signal-opened state, for which comparably well-resolved maps aren't yet published, but one is left with no reason to think that proximity to this loop would influence gating.

4. The unexplained densities for T1 and the luminal proteins are certainly intriguing, but are not validated. This is the one part of the translocon half of the study that really has the potential to reveal something new. It was therefore a bit disappointing to this reader that they did not do more to identify and validate these proteins.

5. The authors make the observation that engaged and hibernating ribosome-Sec61 complexes are basically the same. It is unclear what this is supposed to mean. Are they claiming that translocons are simply open all the time? Is the implication that there is no plug in the central channel even in the inactive state? Some explanation of the relevance of this observation is warranted. It does not seem likely that active and inactive translocation channels are identical.

6. Related to the previous point, how certain are the authors that the plug domain is where they have assigned it? In Fig. S7H, there is unassigned density in the centre of the channel in both the active and idle maps. Could this be the plug in one or both maps? I believe in the active map, they assign this to the nascent chain. If so, then what is it in the idle map?

7. What does "interplay in situ remains largely uncharted" mean in the Abstract? I think some readers may object to the notion that simply because something has not been seen by EM means it is not charted. Information obtained by other means (crystallography, single-particle EM, etc.) can be validated to be relevant to the in vivo situation using approaches such as mutagenesis, various functional assays, and so forth. Put simply, the authors' implication that most of what we know is based on in vitro studies and the in vivo situation is "uncharted" should probably be softened.

8. The conclusions about the percent of hibernating ribosomes and their bound factors probably

warrant some caveats. If a longer microsome isolation procedure results in markedly higher proportion of empty ribosomes, then clearly ribosomes go into 'hibernation' after cell lysis. It therefore follows that the current purification protocol might have some hibernating ribosomes that existed *in vivo*, and others that occurred post-lysis. The authors have no way of knowing how many are artefacts and how many were that way in the cell prior to lysis.

9. Related to the previous point, how can the authors know whether the difference in hibernating ribosomes they see between ER and soluble ribosomes is statistically or biologically significant? What they describe as a "marked" difference seems fairly small to me, but regardless, the relevance or reproducibility is unclear. Some comment is probably needed here.

10. The authors note a contact between the TRAP- and Sec61-gamma termini and state that this contact may maintain the 3-nm separation between them. This seems unlikely because this contact is negligible compared to the large contact area between TRAP-gamma and the ribosome (Fig. 3d), and that contact alone would fix TRAP at its position. I would suggest removing this explanation and simply note the proximity of the TRAP and Sec61 termini.

11. What is the basis for knowing the GTP/GDP state of the factors? The resolution would seem insufficient to know this from the structure, so this is presumably deduced from earlier conclusions about the nucleotide state of each observed structure. Please clarify.

12. Two minor issues in the Methods: (i) method for freezing the crude microsomes (presumably in liquid N₂) and temperature and length of time of storage (presumably at -80° C) is not specified; (ii) it is stated that 20 ng of microsomes were used for immunoblotting. This seems off by a factor of 1000? Please check and adjust as needed.

13. Worth checking the referencing carefully. For example, on p. 4, lines 18-20, the sentence stating that TRAP supports the insertion of many SPs cites two studies that are not about TRAP. Two of the citations about TRAP being near stoichiometric to Sec61 also did not study TRAP (or for that matter, Sec61). I didn't check all the references, but a double-checking by the authors might be warranted.

Referee #3:

We are not experts in translation or translocation, but are users of cryo-EM and cryo-ET, so we did not comment much on the details of the ribosome states described and their significance.

This study has used subtomogram averaging of ribosomes from purified endoplasmic reticulum vesicles (microsomes) to generate snapshots of ribosome states within HEK cells. It has made extensive use of RELION's 3D classification capabilities to reconstruct the translation/translocation mechanism of ER-associated ribosomes, and other post-translational details, such as N-glycosylation and amino acid isomerisation. Using these methods has enabled visualisation of states that have not been previously seen in purified ribosomes. The methods have also allowed the solving of structures of associated proteins in the ER.

We would like to suggest that the work is at the cutting edge of in situ EM. It is an admittedly “easy” sample (big, much known, abundant) but it is a triumph, nevertheless. Resolutions and map qualities are mostly sufficient (although not mind-blowing, see below). Being able to solve the structure of TRAP is a highlight, and the translocon supercomplexes are potential landmark achievements, at least to us as outsiders. The work is descriptive but is very interesting and will lead to much follow-up work. It also works as an example, spurring on much future work on less-favourable in situ EM in the future.

One of our (minor) concerns is with describing these structures as generated “in situ”. This is more typically applied to lamellae, and while it is clear that their approach of ER vesicle purification is a good approach that has allowed elucidation of many different ribosome states and associated ER protein complexes, we question whether it can truly be described as “in situ”. The purification protocol is described as extremely fast but seems to take more than 20 min, as far as could be gauged from the methods section (sorry if this is a misunderstanding somehow).

As a general comment, the methods are often lacking in detail and it was not always clear how the analyses described in the main text were undertaken. Maybe more detailed descriptions could be provided as a supplement? Practitioners would want to know exactly how certain analyses were performed.

Also, it is not always made particularly clear which subset of particles was used for any one structure. In particular, from reading the main text it gives the impression that the multiple structures of the elongation cycle in Fig. 1 are membrane associated, but when consulting supplementary Fig. S2 it appears these consist of a mixture of both membrane-bound and cytosolic ribosomes.

Finally, it is not clear how the ribosomes were classified as either cytosolic or membrane bound. Was this during the particle picking or by 3D classification? This is important as comparisons are made between the two populations. Looking at Fig. 2 and the mapping back of different states onto the tomogram, it looks like some particles have been determined to be membrane-associated when there is no obvious membrane, and others are coloured to be soluble, when they appear to be sitting on the membrane.

In summary, an exciting piece of cutting-edge structural biology, of the sort that will dominate the next 10-20 years in many other contexts and fields (our own included). Editorial revisions should be able to deal with our comments.

Some specific points in no particular order:

- 1) Figs 2b (with the caveats above) and 4e are beautiful and helpful.
- 2) Fig. 2b mapping back: Can any more insight be gained into the regulation of how these complexes change to meet the post-translational needs of their products?
- 3) It might be good to mention how fast the microsomes purification method is in the main text.

- 4) Is the difference in elongating/hibernating populations in the cytosol and on the ER membrane (Fig. 1e) statistically significant? This can be checked with a two-sample t-test for unpaired samples (e.g. Welch's t-test). Actually, are any of the differences demonstrated in Fig. 1e statistically significant?
- 5) Some resolutions could be described as low given the sample used (vesicles, thin) and the method (W, M and Relion). The authors used a 200 kV instrument, which is unusual for this type of work. The authors' opinion on this would be good to hear.
- 6) Where there any issues with anisotropy due to lack of top views of the membrane bound population due to the missing wedge?
- 7) Fig. 1e needs some more labelling as it is quite difficult to understand the graphs. There is no mention of ES27L in the legend, and this figure is referred to sometime before it is introduced in the main text. Also why is the data only shown for the ES27L population of "cytosolic" ribosomes and not all of those that have been determined to be cytosolic? Following on from this the text describes the hibernating ribosomes at the ER membrane as "markedly higher abundance than their cytosolic counterparts", yet from the graph alone this does not feel very convincing and no numbers have been given.
- 8) Fig. 1 is not consistent in the colouring of A-/P-/E- trRNA between the different states shown.
- 9) There is some discrepancy in the determination of the tomogram thicknesses. In the methods section it states that the average thickness is 184 nm, yet the legend for Fig. S1 states that this is 156 nm.
- 10) The legend for Fig. S1 is not the easiest to follow, and we think that on the figure there is some incorrect labelling and mask 2 is shown to be applied at a stage when it should be mask 1.
- 11) What size fiducials were used?
- 12) Which version of Relion was used? We assume that it is 3.1 to fit in with the Warp/M pipeline, but this needs to be explicitly stated as there are some major differences in subtomogram averaging between Relion versions, especially version 4.
- 13) We feel the term "deep classification" is potentially a bit misleading. This term brings to mind the potential use of deep neural networks for 3D classification, which did not feature in this study.
- 14) In some figures, the agreement between map and model is less than completely convincing, owing to poor local resolution and probably also to the fact that, as far as we can tell from reading the *Methods > Model building* subsection, the models were obtained by rigid-body fitting pre-existing models, with adjustments made only to the TMHs of TRAP. See for example the right panel of Fig. S5d, where several short helices are not well placed in the density. See similarly problematic helices in Supplementary Fig. S9c. The low local resolution is to be expected, given the large effective pixel sizes used for the classification (3.45 Å for ordinary 3D classification and then 6.9 Å for focused

classification) and the heterogeneity of the sample. Comments on why these things do not matter for the conclusions reached would be helpful.

Minor points that the authors might find helpful:

15) > its native interactions, including biogenesis cofactors
(line 58) to

> its native interactions, including those with biogenesis cofactors

16) > were aligned RELION focused on the LSU
(line 308) to

> were aligned in RELION, focused on the LSU

17) > Sec61 can switch from a closed to open conformations
(line 49) to

> Sec61 can switch from a closed to an open conformation

18) > In analogy to the cytosolic hibernating ribosome 28, we also detected a second rotated ribosome state (5%), which features eEF2 while CCDC124 is absent.
(line 85) to

> We also detected a second rotated ribosome state (5%), which features eEF2, and from which CCDC124 is absent, analogous to the cytosolic hibernating ribosome.

As it stands, it reads as if the detection itself (rather than what was detected) was analogous to the cytosolic hibernating ribosome.

19) > While OSTA structure and association with ribosome and Sec61 have been studied in detail in isolation,

> its native interactions, including biogenesis cofactors such as ER chaperones remain elusive.
(line 56) to

> While (the) OSTA's structure and its specific association with the ribosome and Sec61 have been studied extensively, its native interactions, including those with biogenesis cofactors such as ER chaperones, remain elusive.

20) > characteristic for
(line 150) to

> characteristic of

21) > allowed building a complete atomic model
(line 166) to

> allowed the building of a complete atomic model

Or, better yet, just don't use a gerund:

> allowed us to build a complete atomic model

22) Remove "unexplained" from

> unexplained density in the lumen coincides with predicted N-glycosylation sites

(line 185), because you are supplying an explanation ...

23) > feature a lipid-filled distance of 2-3.5 nm between them
(line 190) to
> are separated by 2.0-3.5 nm of lipid density

24) > the glucosyltransferases acting upstream of OSTA or dolichyldiphosphatase I acting downstream OSTA
(line 212), to
> the glucosyltransferases acting upstream of OSTA or the dolichyldiphosphatase I acting downstream of OSTA
(that is, add a second 'the' and a second 'of').

25) > neither atomic models provide acceptable fits
(line 213) to
> neither atomic model provides an acceptable fit

26) > L2-3 and L2-4 are poorly resolved concomitant with the flexibility of the RPN2 N-terminal domain
(line 225) to
> L2-3 and L2-4 are poorly resolved, likely due to the flexibility of the RPN2 N-terminal domain
The word "concomitant" is so often misused in scientific writing. If in doubt, leave it out.

27) Comma-separate the thousands on line 314 (i.e. 134,350) and on line 337 (i.e. 5,554).

28) > D1-3 indicate the domains eEF1a domains 1-3
to
> D1-3 indicate eEF1a domains 1-3

29) > A, P and E indicate ribosomal aminoacyl, peptidyl, and exit site, respectively
(line 392) to
> A, P and E indicate the ribosomal aminoacyl, peptidyl, and exit sites, respectively

30) Change "domain" to "domains" on line 422.

31) Replace the hyphen in "false-positives" (line 444) with a space: "false positives".

32) Not sure why one of the arrows in Fig. 1 is dashed.

33) Capitalise PDB on line 393.

34) Inconsistent application of the serial comma (the so-called "Oxford comma", sorry, this is now getting pedantic):

On line 392:
> A, P and E

(serial comma absent)

> aminoacyl, peptidyl, and exit

(serial comma present)

35) also:

> The arrangement of Sec61, TRAP, OSTA, and T1

(line 208)

36) Inconsistent hyphenation of "cofactor"

> stoichiometric and sub-stoichiometric *cofactors*

(line 49)

> biogenesis *cofactors* such as ER chaperones

(line 58)

> Sec61 associates with distinct *co-factors*

(line 50)

We suggest no hyphenation.

37) Use an actual β in line 416 (i.e. β hairpin), maybe with a hyphen (β -hairpin).

38) Remove the hyphen from "re-positioning" in line 183, i.e. write "repositioning".

39) > pseudosymmetrical (line 190) to > pseudosymmetric.

Author Rebuttals to Initial Comments:

Point-by-point reply to reviewers

We thank the reviewers for their thoughtful comments, which we considered in thoroughly revising of our manuscript. To address the comments, we added a substantial amount of additional data and performed additional analysis of the data. In summary, we added the following data with the listed main conclusions:

- (i) polysome analysis of the tomographic data: all reviewers asked about polysomes and hence we performed a neighbor-based analysis of ribosomal particles. In particular, this analysis provides insights into the organization of the different translocon complexes.
- (ii) comparative analysis of ER-microsomes from stressed cells: DTT stress underscores the physiological relevance of the hibernating state and validates the polysome analysis. It also clarifies the positioning of the Sec61 plug in the hibernating Sec61-OSTA-TRAP translocon.
- (iii) tomography data of cryo-FIB milled human cells: the translation state analysis is consistent with the *ex vivo* analysis.
- (iv) proteomics data of microsomes: the data indicate the most abundant proteins in the sample. In particular, eEF1A and eEF2 are highly abundant, while other structurally GTPases are not present in amount comparable to those of ribosomal proteins.
- (v) Single particle cryo-EM structure of classical PRE+ state: the map (~2.9 Å resolution overall and up to 3.5 Å for eEF1a) unambiguously identifies eEF1a and reveals residue-level interactions.

Below we address the specific points of the reviewers in detail (our replies in *italic*).

Referee 1:

1. I am not quite sure what holds the latter set of ribosomes near the membrane surface (more discussion on this would be helpful), but it would seem to be the most logical conclusion (given the isolation procedure) that they may be polysome-associated with the ribosomes that are caught in the act of secretion through the Sec61 complex. Is this what the authors think?

In the revised manuscript we performed a detailed analysis of ribosome neighbor configurations, which shows different 3D arrangements for cytosolic and ER-bound polysomes. The analysis shows that it is very unlikely cytosolic and ER-translocon bound ribosomes (Fig. 2E) coincide in a polysome. Thus, cytosolic ribosomes are likely due to imperfect separation of ER or they may be loosely tethered to the ER by mRNAs.

2. Are all the ribosomes monosomes? Is there any evidence of polysomal structure?

As all reviewers were curious about polysome structures we added a neighborhood analysis of ribosome particles (Fig. 2D-F, S11, S12), which reveals that most ribosomes are part of polysomes, as expected from their elongating state. Consistently, ribosomes in a hibernating state do not show a distinctive neighborhood distribution (Fig. 2D).

3. How do the authors explain their statement that microsomes are known to harbor “hibernating”, i.e. non-active, ribosomes? Does the evidence for this derive from ref. 27 or some other source?

Microsomes from dog pancreas are commonly used for biochemical assays, even available commercially. Indeed ref. 27 contains the most comprehensive analysis of the associated ribosomes to our knowledge. In these microsomes most ribosomes not only lack the tRNA in the P-site but also eEF2 (i.e., these are idle and not ‘hibernating’), which is most likely a consequence of preparation. We clarify the term ‘non-active’ in the revised text.

4. From what type of functional complex does a hibernating ribosome come from – elongating ribosomes?

We understand the question as whether the hibernating state may be a result of preparation, potentially converting elongating ribosomes to hibernating ones. In the revision we added ribosome analysis from microsomes originating from DTT-stressed cells, which display almost exclusively hibernating ribosomes on the membrane. Thus, the observed hibernating ribosomes are likely a long-lived state in the cell, although we cannot completely rule out formation of hibernating ribosomes during purification (see also point 6 of reviewer #2).

Major comments:

5. To address this head on, in the writing of the paper, it would be most helpful to the reader if the authors would show the closest thing to the raw data first (e.g. something akin to Fig. 2b). The authors may also wish to highlight further the complex and massively consolidated set of information that is present in SI Fig. 2, which provides all of the real numbers of each complex for the entire paper. While the conversion to percentages of each complex is certainly helpful in terms of simplified presentation, it seems relevant for the reader to keep in mind that the raw data are comprised of sub-classes, with the stated resolutions and numbers of particles to keep things in context.

We added a short paragraph opening the main text with summarizing the analysis. Moreover, we added a representation of a segmented tomogram and back-mapped intermediate states to Fig. 1 to provide better context and included the absolute particle numbers of ribosomal intermediates in Fig. 1A.

To provide better insight into the raw data we also prepared additional supplementary figures, such as Fig. S10.

6. How was the cryoET data collected in a manner that is considered high-throughput? If another improvement in beam shift rate or grid hole organization is made to increase throughput another 5- to 10-fold, will that be ultra-high throughput? This doesn't seem properly justified, and it's unclear what this means or how it adds value.

The comment refers to the term ‘high-throughput’ in the abstract, which may indeed hold room for improvement. The main point of our analysis is the large amount of data (altogether >1,000 tiltseries now make it into this manuscript, which is unprecedented to our knowledge)

rather than the acquisition rate. In the revised version we now refer to ‘extensive classification’ to capture the essence.

7. Given the very small number of particles and resolution limits in each class (see SI Fig. 2), in its present form it also seems challenging for the reader to understand that there are uncertainties associated with the classification and sorting of ribosomes into the various sub-classes presented. Something that clarifies and touches on this point would be very instructive and helpful. This is a challenging balance, of course, particularly when working to concisely present the work, but it seems relevant and necessary from my perspective (see additional comments below). Perhaps discussion of these points can be added to the main text and/or put into the supplemental information?

We added a detailed statistical analysis (see also reply to point 4 of reviewer #3) to provide error bars and significance measures, indicating that particle numbers are not limiting. We estimate the technical uncertainties of 3D classification at 5-15% determined by repeating our processing workflow 3 times. Moreover, we added the results of a combined subtomogram analysis of our large data set (~135,000 particles) with two smaller data sets of ER derived vesicles (~3,000 and 7,000 particles, respectively), which demonstrates that, although distribution of elongation states varies among samples, the overall trend is consistent (Fig. 5C).

8. Another major issue that the authors should address is the tenor and tone of some of their interpretations about the elongation cycle and its connections to prior literature on the subject. The authors seem eager to convey that they have assembled a nearly complete translation elongation cycle from their ~100K or so picked and sorted particles (8 classes) at 6-10A resolution that is somehow distinct from what is commonly accepted. Some of the complexes they report have never been seen before or are unexpected based on the field’s present understanding. What the reader may take away from the work, if left in its current form, is that the present work overturns canon: in actuality, as the authors themselves emphasize, their sample preparations are an earlier snapshot compared to their previous work of a rapidly degrading sample and therefore extreme caution and proper context should be provided to the reader. After all, their findings may potentially reflect spurious outcomes of an ongoing decay process.

Our data do not cover a complete elongation cycle because very short-lived states are unlikely to be detected, but our results highlight the key rate-limiting steps. In the revised manuscript we discuss the possible impact of sample preparation in more detail. Importantly, to address the potential effect of sample preparation on translation state we have added cellular tomography data from human cell lines (Fig. S5). Due to the much lower throughput of cellular imaging, the number of in situ (sensu stricto) ribosome particles (5,818) is substantially lower than that of ex vivo particles, which restricts classification depth. Nevertheless, the data clearly indicate the abundant – and unexpected – classical PRE+ state and its abundance is comparable with the distribution of intermediates from our ex vivo data (Fig. 5C) (see previous point 7).

9. The assigned states and conformational changes are not specifically and quantitatively compared with prior work in a meaningful way. Comparative structures from highly purified and/or reconstituted systems exist. Careful in vitro reconstituted reaction studies with highly active material have proven vital to our understanding of translation, including the work presented and the authors' interpretations (aside from the speculated departures they infer from their data [more on this below]). While I am sure the authors recognize this, it doesn't currently read as such.

We adapted our Fig. S3, which now illustrates the agreements and deviations of our reconstructions with previously published high-resolution structures of ribosomal intermediate states. We also added a chart to visually compare abundance of intermediates between our samples and the sample from Behrmann et al. (Fig. S5D).

10. For example, more could be done to appropriately convey to the reader that the authors have inferred the ordering of events from this vast array of prior literature. The work presented in refs 10 and 11 – which the authors seem to hold as a solitary benchmark for mammalian translation studies – is based on an enormous body of prior work, including the major translation-factor bound classes presented. The authors should endeavor to be more complete in their referencing and be concrete about what complexes have been foreshadowed by prior work and what complexes have not been hypothesized previously so that it is clear to the reader. In this regard, it should be noted that the authors state on lines 95-97 that an eEF1A ternary complex in an A/T state, with both E and P sites filled (which they call a “decoding-sampling” state in ref. 11) was not observed in ref. 10, but it clearly was. The decoding-sampling state defined in ref. 11 is also distinct from an A/T state, as the A/T state as defined implies specific contacts with the large subunit GTPase activating center and Sarcin-Ricin loop as well as full engagement by the small subunit decoding site. The decoding-sampling state is defined as highly transient, which is not likely to be what is observed in the present analysis but rather the A/T state. In ref. 10, the authors of that identify an eEF1A-bound A/T state complex (8% occupancy), similar to the one reported here in which domain 1 of eEF1A - the G domain - exhibited weak density. It should be clarified that the present data differ from the prior work only by having more robust density for eEF1A and that one explanation may be their purification strategies, which are likely to be more rapid and thus less prone to degradation. The eEF1A complex, with all three domains fully accounted for, has also been directly isolated from rabbit reticulocyte lysate and the structure determined to roughly 4Å by the Hegde/Ramakrishnan groups using a nascent polypeptide pull-down strategy (Cell, 2016; PDBID:5LZS).

In the revised manuscript we phrased this passage more precisely. Specifically, we describe that the post-hydrolysis state, but not the decoding-sampling or decoding-recognition state were captured in Behrmann et al.'s analysis. We concede that we have been imprecise in stating that we observed eEF1a in the A/T state: the eEF1a-tRNA ternary complex is in a stage of codon sampling. Our reconstruction almost perfectly matches the structure of the decoding-sampling state from Budkevich et al. (4CXG, required translation of eEF1a•tRNA into our map is 0.7 Å), while it only poorly resembles structures of the decoding-recognition state from Hegde et al. (5LZS, shift[eEF1a•tRNA] = 7.4 Å) and the post-hydrolysis state from Behrmann et al. (EMDB-2908, shift[eEF1a•tRNA] = 7.5 Å for D2/D3). Moreover, in our reconstruction,

nucleobases A1824 and A1825, which reorient to bind to the tRNA upon recognition, were not observed in the characteristic flipped-out conformation, whereas densities for flipped-out A1824/A1825 is clearly visible in our classical-pre reconstructions. Consequently, our reconstruction of the ribosome-bound eEF1a-tRNA ternary complex represents a state prior to codon recognition and GTP-hydrolysis and thus we assigned it the decoding-sampling state. We illustrated our findings in Fig. S6A,B in the revised manuscript.

We emphasize that our sample is a cell lysate, meaning that the ribosomes present in the sample are mostly ribosomes testing non cognate tRNA (those being much more likely than finding the correct one in the cell). For these non-cognate tRNA the GTP hydrolysis on eEF1a is precisely of low efficiency, therefore also supporting the conclusion that the abundant state we observed is likely to mostly contain eEF1a in a GTP-bound conformation.

11. Connections to prior literature also seem important to contextualize the findings against expectation. For instance, atomic models of the “decoding-sampling” (which should be the A/T state) and “Pre+” complexes are shown in Fig. 1 using previously published, partial models of rabbit ribosomes (PDBID:4CXG) and X-ray structures of isolated eEF1A isoform 2 bound to GDP that forms intermolecular crystal contacts (PDBID: 4C0S). In the former case, I think the more relevant model would be PDBID:5LZS. In the latter case, it would seem notable that the putative ribosome-bound eEF1A protein in its “GDP” conformation looks so similar to eEF1A2(GDP) found crystallographically packed as a dimer in the asymmetric unit.

We believe there may have been a misunderstanding about the major feature of the eEF1A conformation in the classical PRE+ state: we observe eEF1A in conformation, which we now refer to as ‘extended’. This extended quaternary structure has also been observed for GDP-bound eEF1A2 and the bacterial EF-Tu homolog. In contrast, the model 5LZS displays eEF1A in the compact quaternary structure, with the bound antibiotic possibly inhibiting the conformational change to the extended form. The extended quaternary structure of 4C0S is unlikely to be affected by crystal contacts, as it is well established for the prokaryotic homologs (e.g., ref. 26 and Berchtold et al. Nature 365:126-32 (1993)). In the revised manuscript, the nomenclature of ‘extended’ and ‘compact’ conformation clarifies the main aspect of eEF1A in the classical PRE+ state.

12. Along similar lines, it seems appropriate to point out the similarities and differences between the relative populations of states observed here and in ref. 10. For instance, it is interesting that no complexes are identified in either work in which eEF2 is bound to its cellular substrate, the pre-translocation complex, although they are present in high abundances: in the present work pre-translocation complexes represent ~60% of the ribosome population evidenced, but in ref. 10 the total pre-translocation complex percentage is roughly 40%. The fast GTP hydrolysis by EF-G argument that is cited to explain this notably unexpected result does not take into consideration that GTP hydrolysis by EF-Tu is reported by the same group to be just as fast as for EF-G, yet the former complex represents 23% of the ribosome population observed. To me this seems striking given the view held by some that EF-G “drives” translocation and recently reported evidence that the rate limiting step of translocation (in bacteria) is an EF-G(GDP) bound complex (Wasserman et al., NSMB,

2016). Oddly, the authors instead find eEF2 trapped on a “POST+” complex for which eEF2 should have negligible affinity. In addition to noting that similar findings have been reported previously (ref. 10), the authors may wish to further clarify that the POST+ state may actually represent an intermediate state of translocation that happens to closely resemble the POST state.

We thank the reviewer for this comment and renamed the “Post+” state to “translocation intermediate”. We emphasize that we believe to have eEF2 captured in a GDP-bound state, as we observe the switch I loop to be disordered (Fig. S6B), which would be consistent with the kinetic study.

Regarding the comparison to GTP-hydrolysis of eEF1A we refer to point 10, where we point out the abundance of non-cognate peptidyl-tRNAs in the cell, which are not triggering efficient hydrolysis.

13. Another striking departure from canon is that the authors report that 35% of their complexes represent an eEF1A-bound “Classical Pre+” complex in which eEF1A is in a GDP-bound conformation (a model derived from a crystal structure of eEF1A2 appears dimerized in the asymmetric unit; see comment above). According to canon, the switch of eEF1A/EF-Tu(bacteria) is understood to trigger rapid release of the factor from the ribosome as aminoacyl-tRNA release occurs. The relevant research in this area - and the departure from expectation – is not noted by the authors, and they instead misquote a molecular dynamics simulation arguing for a power-stroke mechanism of tRNA selection (see additional comments below).

Regarding the crystal structure of eEF1A(2) and its possible crystal contacts we refer to point 11: the relevant point is the extended quaternary structure of eEF1A, which is also well established for EF-Tu•GDP.

We refer to the additional comments in points 15 and 16 for the discussion of the Classical Pre+ complex.

14. Do the authors possess any direct mechanistic information regarding the functional relevance or homogeneity of any of the complexes presented? While I would imagine much of the information is correct in its overall interpretation, particularly as it related to tRNA occupancies, ribosome rotations, etc it would seem prudent not to overreach given the present resolution limits and the complexities associated with assigning low-resolution densities to specific factors and factor conformations. It is my understanding that relatively low-resolution data, plus relatively small particle numbers, can result in increased probabilities of mixed sub-populations with multiple conformations and/or compositions. This leaves open the possibility of misinterpretation of states, at least in my view.

We are aware of these limitations and applied gold standard methods to limit their effect on our conclusions, as typically used in relevant literature in the field. First the fact that we acquire tomography data gives us 3D information on each subvolume itself, which supports an improved classification for limited resolution and number of particles. In addition, we checked that the classification algorithms had converged by using a sufficient number of iterations to obtain stable classes. Several masks were also tested at each step to limit the

effect of the choice of the mask itself. We added more details on these steps in the methods section and expended the statistical analysis (see also responses to reviewer #3). Regarding functional relevance we point out the added polysome analysis: the states assigned to elongation occur in polysomes, whereas hibernating ribosomes do not.

15. Another example of overreach is that the authors do not have the ability to know the nucleotide status (GTP or GDP) of the factors they observe. This point is definitive. Their modeling of two distinct conformations of eEF1A is questionable given what is known about the process and what is presented in the paper as support for this interpretation. The notion that eEF1A undergoes conformational changes on the ribosome as a result of GTP hydrolysis and that this hydrolysis mechanism contributes to the proofreading stage of tRNA selection has been known in bacteria for several decades. As stated above, however, in previous bodies of work, eEF1A/EF-Tu is thought to release from aa-tRNA and the ribosome almost immediately after GTP hydrolysis (which is almost immediately followed by inorganic phosphate release). While taking into consideration that there are some groups presenting evidence/hypotheses that eEF1A/EF-Tu may undergo a change to its GDP-bound conformation while on the ribosome (see work from the M. Ehrenberg, C. Knudsen, K. Sanbonmatsu, B. Cooperman and S. Blanchard labs) – none of which is appropriately referenced – I have very little if any confidence in the authors' interpretation. The model that such conformational changes “drive” aa-tRNA accommodation during proofreading through a power-stroke mechanism has: 1] not been evidenced or stated to occur as suggested by their citation of ref. 32 (the authors' statement in this regard misrepresents the findings presented in that body of work); 2] is antithetical to proofreading mechanisms that are evidenced to increase fidelity in this step of tRNA selection (as reiterated in ref. 32); and 3] is physically implausible for other reasons (more on this below).

We have now replaced “GTP bound” state by “compact” conformation and “GDP bound” state by “extended” conformation, which match our observations but do not rely on the exact nucleotide state of the complexes. We refer to above point 10 for the rationale of an abundant GTP bound decoding complex. The data added in the revision (see next point) provide evidence that eEF1a remains bound to the ribosome during accommodation and we have added the suggested citations (Morse et al. Proc Natl Acad Sci USA 117:3610-20 (2020); Liu, W. et al. ACS Chem Biol 9:2421-31 (2014); Jeong et al. Proc Natl Acad Sci USA 113:13744-9 (2016)).

Regarding the notion of eEF1a hydrolysis actively driving proofreading we have decided to remove the discussion of ref. 32 and the ‘powerstroke’ concept because this mechanistic model does not rely on precisely the same eEF1a conformations that we observe. Thus, we leave the elucidation of proofreading energetics for future research based on our observations.

16. If the authors wish to remain focused on this point, they should consider referencing experimental evidence that supports this notion. They should specifically address the likelihood that the factor density that they observe may not be eEF1A or contain a nucleotide. They may also wish to present a statistical argument that it is eEF1A and not another factor that takes into consideration their present resolution, map quality

with low particle numbers and that they are working with complexes isolated from cellular extract. Significantly more effort will need to be made to convince the reader of this point. From my perspective, there may be numerous small GTPases - or proteolytic fragments thereof - that could be bound to the ribosome in the same position.

We have removed statements about nucleotide states and replaced it by “compact” or “extended” conformation, which matches our observations. In the revision, we included a polysome analysis which clearly maps the PRE+ state to polysomes, indicating involvement in active elongation (Fig. S11D). We also added proteomics data from our microsome preparation, which demonstrates that eEF1a is 50- to 4,000-fold more abundant than its structurally related counterparts (Fig. S4). In the range of abundance of ribosomal proteins, no GTPase candidate other than eEF1a could explain the observed factor density in the classical pre+ state. The comparative fit of all GTPase candidates to the observed density also indicates that eEF1a explains the density best.

Most importantly, we have performed cryo-EM single particle analysis of the classical PRE+ state (Fig. 1 E, Supp. Figure S7). The resolution is sufficient to unambiguously identify large side chains in eEF1a domain 3. Thus, it is definitely eEF1a that associates with the ribosome in the classical PRE+ state.

17. Further to the point that their factor density assignment is unlikely to be eEF1A, and something potentially related to a non-functional ribosome sub-class, is the 3% class assigned as “Rotated-1 Pre+” in which they also state that a GDP-bound conformation of eEF1A is found. The possibility that this specific complex is off-pathway should be considered for a number of reasons. eEF1A makes cross-subunit contacts that are specific for the unrotated ribosome (see ref. 10 or Shao et al., Cell, 2016) and the state shown has eEF1A bound to a rotated ribosome conformation for which most known stabilizing contacts are not possible. Moreover, this complex has to have a nascent polypeptide linked to it to be placed in the elongation cycle as it is. To which tRNA is the nascent polypeptide linked in this complex? If it is on the incoming, A-site-bound tRNA, then peptide bond formation would have already occurred and hence there is no reason for translation factor eEF1A to be bound at all. If one alternatively infers the nascent peptide is on the P/E hybrid tRNA somehow, this would mean that the authors are suggesting that the nascent peptide has backtracked out of the peptidyltransferase center by 10-15 amino acids. Both scenarios seem highly implausible given what is presently known about the translation machinery. Is there any supporting evidence for either interpretation? In my view, the authors’ assignment of factor density to eEF1A(GDP) in this complex casts further doubt on eEF1A(GDP) assignment in the “Classical Pre+” complex. In the manuscript’s current form, the elongation cycle presentation, the aforementioned referencing deficiencies and the over-reaching claims noted appear to weaken the work and should be heavily revised before further consideration.

Please see point 16 for the unambiguous identification of eEF1a. The new ribosome neighborhood analysis shows that the Rotated-1 Pre+ complex is found on polysomes, indicating that it represents an actively translating ribosome. We note that the rotated-1 Pre state with the same tRNA sites occupied, but without any elongation factor, was also

previously described in the polysome analysis by Behrmann et al. The only difference here is the presence of a similar state also in complex with the extended conformation of eEF1a.

Since there is no previous report of eEF1A in an extended conformation bound to the ribosome, all interactions between eEF1A and the ribosome that have previously been analyzed refer to the compact conformation of the eEF1a and cannot possibly exclude that eEF1a in its extended conformation is able to make other interactions with the ribosomal subunits. In the extended conformation eEF1A is interacting with the LSU mostly via its domain 2 and 3. In the compact conformation of eEF1A additional interactions with the SSU are mediated by the domain 2, while in our extended conformation, domain 2 has moved towards the A site and instead domain 1 is facing the SSU. However, domain 1 remains at distance from SSU, precluding direct interactions, and remaining compatible with the SSU rotation. The newly added cryo-EM SPA data show residue-level interactions of eEF1a domains D2 and D3 with the SRL (Fig. 1E, Supp. Figure S7).

We think that the only possible situation in this conformation is that the polypeptide chain is bound to the A-site tRNA and the extended eEF1a is not yet released. Overall, our data suggest that the extended eEF1a conformation is compatible with different ribosomal conformations possibly explaining that eEF1A could stay longer on the ribosome than observed in bacteria (where the switch loops for instance seem to contribute to the detachment of EF-Tu at the later steps of its conformational change).

18. The translocon/secretory machinery findings appear much more novel, concrete, and hypothesis-generating. It's clear that the authors have spent more time digging through the literature in this area and that they are much more comfortable navigating this important space. This is the intended focus of the work after all it would seem. And it showed. The fitting of models based on AlphaFold structures (which may have issues of their own) into potentially weak and amorphous density does seem like a bit of a risky business in my view, given the potential limitations of this experimental model generation approach and the complexity of partially purified extracts, but this is the vanguard. As for the alternative conformation of eEF1A proposed in the elongation cycle, I would think that erring on the side of caution or providing discussion on alternative models may be appropriate.

We have improved and nuanced our conclusions on the role of eEF1A in the elongation cycle suggested by our data in the new version of the manuscript and thank for the constructive comments.

19. Other comments: In Fig. 1, panel e is uninterpretable and doesn't seem to relate to the figure legend.

The comparison of distributions of intermediate states has been revised and moved to supplement (Fig. S5D).

Referee #2

This study presents a cryo-ET analysis of rough ER microsomes isolated (relatively rapidly) from cultured cells. Two aspects are analysed: (i) ribosomes and their conformations throughout the translation cycle; (ii) the ER translocation and glycosylation machinery. One main new solid observation related to each aspect is made. Regarding translation, the authors observe a GDP-bound form of eEF1a in which the tRNA is accommodated into the ribosome. This is used to argue that eEF1a participates in a powerstroke mechanism and may provide a proofreading function. Regarding translocation, the authors characterise the proportions of different flavours of translocon based on the associated components. In addition, they fitted AlphaFold models into their density to generate an atomic model of the most abundant translocon. Intriguingly, several unexplained densities are observed, for which the authors suggest some possible explanations.

The aim of direct visualisation of key molecular processes in their native context is an important goal. It has the value of validating inferences drawn from in vitro and reconstituted systems, revealing areas where our understanding may be deficient, and in the best cases, revealing something new that had been previously inaccessible or overlooked. This study makes some progress toward these goals, but is not really “in situ” as claimed (see below). The most noteworthy observation is probably the eEF1a-GDP state, whereas the other observations are either modest extensions of earlier work, confusing, or of uncertain biological relevance. For these reasons, the study would seem to be insufficiently developed at this stage for publication in Nature. In short, it was very difficult to actually discern the “numerous mechanistic insights advancing beyond the study of isolated components” as claimed by the authors in the abstract.

We rephrased the sections of the abstract (see also point 2). The last sentence now conveys the major message of the manuscript as: “Collectively, we visualize ER-bound polysomes with its coordinated downstream machinery and its response to cellular stress.”

In short, we believe the experiments and analyses added to the manuscript make the unique contribution of our work to the field more apparent.

Specific points, in order of importance:

1. A major issue is the use of “in situ” to describe the findings. The moment cells are lysed, anything relating to translation cannot realistically be considered in situ because all the translation factors are rapidly diluted by a massive level. Furthermore, salt, ATP, GTP, and other metabolite concentrations are completely altered, and no matter how quickly one does the fractionation, the time frames are enormous relative to that of (bio)chemical reactions. Thus, whatever is observed with ribosomes must be considered somewhere between in vitro and in situ. They are endogenous (i.e., non-recombinant) components, but it is unclear whether they reflect the situation in intact living cells any better than analysing an in vitro translation extract. Indeed, ribosome profiling studies have illustrated that after

lysis, ribosomes can rapidly change conformations. The membrane is probably closer to “in situ” because membrane proteins are confined to the membrane. Nonetheless, some caution is warranted because various things like ER calcium, ER morphology, membrane curvature, and so forth are altered upon lysis. Whether this impacts what is observed is not clear. For these reasons, I would strongly recommend that the authors not use “in situ” to describe the situation that is observed.

Instead of the term ‘in situ’, we now use ‘ex vivo’ and ‘near native’ in the manuscript. We also added ‘in situ’ (sensu stricto) data from frozen hydrated FIB-milled human cells. Albeit to lower resolution, these data confirm the presence of similar states as the decoding and the PRE+ (Fig. S5) (see also response to question 7 of reviewer #1).

2. Much is made of the molecular model of the most abundant ER translocon. As far as I can see, the AlphaFold models of TRAP subunits could have been fitted just as well into earlier cryo-ET or cryo-EM maps, and no additional information is gained from the new cryo-ET analysis. While it is nice to have such a model (thanks to the transformative advance represented by AlphaFold!), the actual advance would be to provide some new insight into what TRAP actually does. Here, the speculation about influencing Sec61 opening is fairly superficial and there is little to support this idea (see next point). Similarly, the only new part of OSTA is the AlphaFold model of RPN2’s N-terminal domains fitted into the map. The relevance of this new information is unclear. Thus, I found it very difficult to see how the authors arrive at the conclusion that this model “reveals the molecular framework for signal peptide (SP) membrane insertion and protein N-glycosylation.” If possible, they should be clear about what the new insights actually are beyond conclusions drawn from earlier work.

We clarify the “molecular framework for signal peptide (SP) membrane insertion and protein N-glycosylation” in the revision and rephrase the complete sentence to “The complete atomic structure of the most abundant ER translocon variant comprising the protein-conducting channel Sec61, the translocon-associated protein complex (TRAP) and the oligosaccharyltransferase complex A (OSTA) reveals an intricate interaction network of TRAP with other translocon components enabling its function early in nascent chain processing”.

TRAP literally provides a frame that positions the fibronectin fold of TRAPa at the exit of Sec61. Specially, this ‘frame’ constructed by the tetrameric TRAP complex positions TRAPa near the Sec61 hinge, which interacts with the fibronectin ‘BC’ and ‘FG’ loops.

As explained in more detail in point 3, our data rules out an active ‘switching’ of Sec61 conformation by TRAP, which has been the largely hypothesized function of this complex. Thus, TRAP’s main role appears to provide protein-protein interactions, which are reversible, even in the context of polysomes. In the revised manuscript we also add a polysome analysis, which shows that TRAP-Sec61 translocons are preferentially encountered early on in translation, while TRAP-OSTA-Sec61 or multipass translocons tend to occur at later stages of translation. Overall, TRAP appears to have notable affinity for Sec61, and only dissociates in multipass polysomes.

To comment on the resolution requirements to fit TRAP unambiguously (which is viewed differently by reviewer #1): the point is that our map visualizes the individual transmembrane

helices, loops in the lumen and glycosylation sites, which allows unambiguous and near-complete model building, which previous maps did not allow.

3. The authors note that the TRAP-alpha TMD "associates with the Sec61-alpha hinge-region (Fig. 3f)," which "may facilitate" gating for weak SPs. A supplementary figure showing the experimental data for the putative interaction between the TRAP-alpha TMD and Sec61 would help the reader judge the claim. More importantly however, no support is provided for the claim that this putative interaction facilitates gating. Such support seems necessary, especially since in the very next sentence they upgrade the claim from a possibility to a fact ("the TRAP architecture positions TRAP-alpha to facilitate SP insertion"). Moreover there is prior reason to doubt this claim; the highest-resolution structures of the 5/6 loop of Sec61 currently available (Braunger et al., 2018; Itskanov et al., 2021) show that this loop doesn't change conformation between the ribosome-primed and the Sec62/63-opened state. Admittedly this mode of opening differs from the ribosome-bound, signal-opened state, for which comparably well-resolved maps aren't yet published, but one is left with no reason to think that proximity to this loop would influence gating.

Apparently, the mechanistic model has not been fully clear: It is not claimed that TRAP α induces a change of the 5/6 loop conformation and Sec61 gate opening, and it indeed does not do so. We provide additional analyses to clarify this point: the multipass translocon does not display a SP and the lateral Sec61 gate is closed (Fig. S10). Importantly, the lateral gate is closed irrespective of TRAP binding for the SP-less multipass translocon, indicating that TRAP α is not sufficient to open the lateral gate. We rephrased the passage as: "In this position TRAP α may constitute a 'handle' supporting SPs in opening the lateral gate, in particular those SPs with weak helical propensity. Depictions of experimental data and the corresponding model of TRAP α and Sec61 has been added to Fig. S14E in the revised manuscript.

4. The unexplained densities for T1 and the luminal proteins are certainly intriguing, but are not validated. This is the one part of the translocon half of the study that really has the potential to reveal something new. It was therefore a bit disappointing to this reader that they did not do more to identify and validate these proteins.

We are working on a validation, which is highly challenging biochemically and time-consuming as the reviewer will appreciate. We believe the data in our manuscript are sufficiently exciting to be shared at this point and not to delay by follow-up experiments that are the subject of another PhD thesis complementing decades of research on the ER-translocon complex.

5. The authors make the observation that engaged and hibernating ribosome-Sec61 complexes are basically the same. It is unclear what this is supposed to mean. Are they claiming that translocons are simply open all the time? Is the implication that there is no plug in the central channel even in the inactive state? Some explanation of the relevance of this observation is warranted. It does not seem likely that active and inactive translocation channels are identical.

We clarify this point in the revision: the statement of an 'open' conformation referred to the lateral gate, not the plug. In the revision, we now distinguish between lateral gate and plug

conformation. We analyzed the plug of the hibernating ribosome-Sec61 complex in more detail with the help additional data (DTT treatment), which we discuss in more detail in point 6. Indeed, while the active and inactive translocons do not differ in the lateral gate, they do differ in the plug.

6. Related to the previous point, how certain are the authors that the plug domain is where they have assigned it? In Fig. S7H, there is unassigned density in the centre of the channel in both the active and idle maps. Could this be the plug in one or both maps? I believe in the active map, they assign this to the nascent chain. If so, then what is it in the idle map?

In the revised manuscript we added tomograms from microsomes of DTT-stressed HEK cells, which primarily show the hibernating ribosome-Sec61 complex. These additional particles allowed us to improve the resolution of the inactive complex such that we can resolve the relatively subtle difference of the plug. Indeed, the Sec61 plug closes in the idle map, while it is open in the active map (Fig. S9 in revised manuscript).

7. What does “interplay in situ remains largely uncharted” mean in the Abstract? I think some readers may object to the notion that simply because something has not been seen by EM means it is not charted. Information obtained by other means (crystallography, single-particle EM, etc.) can be validated to be relevant to the in vivo situation using approaches such as mutagenesis, various functional assays, and so forth. Put simply, the authors’ implication that most of what we know is based on in vitro studies and the in vivo situation is “uncharted” should probably be softened.

We rephrased the passage to “insights into their interplay in the native membrane remain limited”.

8. The conclusions about the percent of hibernating ribosomes and their bound factors probably warrant some caveats. If a longer microsome isolation procedure results in markedly higher proportion of empty ribosomes, then clearly ribosomes go into ‘hibernation’ after cell lysis. It therefore follows that the current purification protocol might have some hibernating ribosomes that existed in vivo, and others that occurred post-lysis. The authors have no way of knowing how many are artefacts and how many were that way in the cell prior to lysis.

We addressed the physiological significance of the observed hibernating ribosomes by analyzing ribosomes from ER-derived microsomes of DTT-stressed cells. Upon inducing the UPR, almost all ribosomes are detected in hibernating states. Nevertheless, the reviewer is correct that we cannot completely exclude the formation of some hibernating ribosomes after lysis. We reflect the uncertain error bar of hibernating ribosome abundance by the statement: “We cannot completely rule out induction of some hibernating ribosomes by lysis, which must be considered when interpreting the relative abundances.”

9. Related to the previous point, how can the authors know whether the difference in hibernating ribosomes they see between ER and soluble ribosomes is statistically or biologically significant? What they describe as a “marked” difference seems fairly

small to me, but regardless, the relevance or reproducibility is unclear. Some comment is probably needed here.

Regarding the biological relevance of ER-associated hibernating ribosomes we refer to response 8.

Since reviewer #3 also asked for a statistical significance analysis we refer to response to major point 4 of referee #3. We have moved the comparison of soluble / ER-bound hibernating ribosomes to Fig. S5D. We thank all reviewers for stumbling across the erroneous 'marked difference', as indeed we rather intended make the opposite point that the distributions are similar.

We have also assessed the reproducibility of the translation state abundances with different samples (see point 7 of reviewer #1) (Fig. S5E).

10. The authors note a contact between the TRAP- and Sec61-gamma termini and state that this contact may maintain the 3-nm separation between them. This seems unlikely because this contact is negligible compared to the large contact area between TRAP-gamma and the ribosome (Fig. 3d), and that contact alone would fix TRAP at its position. I would suggest removing this explanation and simply note the proximity of the TRAP and Sec61 termini.

Removed.

11. What is the basis for knowing the GTP/GDP state of the factors? The resolution would seem insufficient to know this from the structure, so this is presumably deduced from earlier conclusions about the nucleotide state of each observed structure. Please clarify.

We refer to our answer to point 11 of reviewer #1 about the nucleotide states, which were indeed inferred from fitted higher resolution structures. We now refer to compact and extended eEF1A conformations as this major change of quaternary structure apparently had not been fully clear.

12. Two minor issues in the Methods: (i) method for freezing the crude microsomes (presumably in liquid N₂) and temperature and length of time of storage (presumably at -80° C) is not specified; (ii) it is stated that 20 ng of microsomes were used for immunoblotting. This seems off by a factor of 1000? Please check and adjust as needed.

Added and corrected.

13. Worth checking the referencing carefully. For example, on p. 4, lines 18-20, the sentence stating that TRAP supports the insertion of many SPs cites two studies that are not about TRAP. Two of the citations about TRAP being near stoichiometric to Sec61 also did not study TRAP (or for that matter, Sec61). I didn't check all the references, but a double-checking by the authors might be warranted.

Corrected.

Referee #3

General 1: One of our (minor) concerns is with describing these structures as generated “in situ”. This is more typically applied to lamellae, and while it is clear that their approach of ER vesicle purification is a good approach that has allowed elucidation of many different ribosome states and associated ER protein complexes, we question whether it can truly be described as “in situ”. The purification protocol is described as extremely fast but seems to take more than 20 min, as far as could be gauged from the methods section (sorry if this is a misunderstanding somehow).

We changed the wording to ‘ex vivo’ and have also included some ‘in situ’ (sensu stricto) data from cryo-FIB milled human cells in the revised manuscript.

General 2: As a general comment, the methods are often lacking in detail and it was not always clear how the analyses described in the main text were undertaken. Maybe more detailed descriptions could be provided as a supplement? Practitioners would want to know exactly how certain analyses were performed.

We expanded our methods section and included more specific details.

General 3: Also, it is not always made particularly clear which subset of particles was used for any one structure. In particular, from reading the main text it gives the impression that the multiple structures of the elongation cycle in Fig. 1 are membrane associated, but when consulting supplementary Fig. S2 it appears these consist of a mixture of both membrane-bound and cytosolic ribosomes.

We clarified this point in text and figures and now point out that the translation state analysis included all particles.

Finally, it is not clear how the ribosomes were classified as either cytosolic or membrane bound. Was this during the particle picking or by 3D classification? This is important as comparisons are made between the two populations. Looking at Fig. 2 and the mapping back of different states onto the tomogram, it looks like some particles have been determined to be membrane-associated when there is no obvious membrane, and others are coloured to be soluble, when they appear to be sitting on the membrane.

Ribosomes were separated according to their features at the ribosomal tunnel exit by 3D classification focused on that area (Fig. S2, mask 1). Ribosomes bound to the Sec61-TRAP-, Sec61-TRAP-OST-, and multipass-translocon, which also displayed strong signal for ER membrane, were considered membrane-bound and ribosomes associated with EBP1(MetAP-like)/ES27L were considered soluble. In the revised manuscript we improved classification of soluble vs. ambiguous classes. We clarified the terminology in the revision: in addition to the membrane-bound ribosomes, which bind to four different types of ER-translocons, we identify a clearly cytosolic class with EBP1 at the exit tunnel, and an ‘unassigned’ pool that we cannot assign unambiguously to membrane or cytosol.

Regarding Fig. 2B, we previously prepared the segmented representation from a tomogram containing several, adjacent ER-vesicles which proved difficult for neural-network-based segmentation tools. We now chose a tomogram with clearer membrane signal and segmented the membranes manually to avoid confusion and provide the best interpretability. Moreover, we included central slices and the respective segmentations of 3 representative tomograms in the supplement (Fig. S13, including the one presented in Fig. 2B) and added different shades to the LSU and SSU to aid interpretation.

In summary, an exciting piece of cutting-edge structural biology, of the sort that will dominate the next 10-20 years in many other contexts and fields (our own included). Editorial revisions should be able to deal with our comments.

Some specific points in no particular order:

1. Figs 2b (with the caveats above) and 4e are beautiful and helpful.

Encouraged by the comment on Figure 2b and the curiosity of referees #1 and #2 we have carried out a more detailed analysis of polysomes (Figs. 2D-F, S12, S13 in the revised manuscript).

2. Fig. 2b mapping back: Can any more insight be gained into the regulation of how these complexes change to meet the post-translational needs of their products?

See above: yes, we have carried out a more detailed polysome analysis and refer to point 1. and 2. of reviewer #1 for a summary.

3. It might be good to mention how fast the microsome purification method is in the main text.

~1h. Added to the main text.

4. Is the difference in elongating/hibernating populations in the cytosol and on the ER membrane (Fig. 1e) statistically significant? This can be checked with a two-sample t-test for unpaired samples (e.g. Welch's t-test). Actually, are any of the differences demonstrated in Fig. 1e statistically significant?

The essence of Fig 1e is to demonstrate that the distribution of ribosomal intermediate states is similar in soluble and membrane-bound ribosomes, and we rather meant to say the opposite. We thank the reviewer for pointing this out (see also reviewer #2, point 9). Nevertheless, we moved this figure to the supplement in favor of the more informative polysome analysis.

We did address statistical testing in new figures we added (see figures trailing/leading state, translation state abundance in polysomes). For all significance analysis we used a multinomial logistic regression model in R, instead of the suggested test. This model deals specifically with counts of different classes as a response variable (nominal variables), which we found appropriate as both predictor and response variables are nominal in our dataset. Additionally, the model was also able to include random effects, such as collection date and tomogram

number to test for co-variation, which is also the essence of the suggested Welch's T-test. Figure 1e has been moved to Fig. S5D.

5. Some resolutions could be described as low given the sample used (vesicles, thin) and the method (W, M and Relion). The authors used a 200 kV instrument, which is unusual for this type of work. The authors' opinion on this would be good to hear.

The primary aim of our study had been to obtain good statistics to classify distinct states with high accuracy. Our analysis indicates that the resolution of the features of interest in this study (translocon, ribosomal factors) were primarily limited by structural variability (e.g., relative rotation of LSU with respect to translocon), suggesting that acquiring data at a Krios would not have provided major additional biological insights (which is what we have done to unambiguously identify eEF1a in the revision). Overall resolutions, typically dominated by the core structure of the ribosome would certainly have profited from a Krios and might have allowed shattering resolution records, but the impact on the ribosome-bound factors would have been minor.

Altogether, more than 1,000 tilt series have been acquired for this study (869 of microsomes from unstressed cells and 212 of stressed cells, not including replicates and the in situ tomograms), requiring 25+ microscope days. Only few institutes worldwide provide this amount of Krios time.

6. Where there any issues with anisotropy due to lack of top views of the membrane bound population due to the missing wedge?

Classification of ribosomes focused on the ribosomal exit tunnel yielded a comparatively large class of 'unassigned' particles (see above, general 4), which likely contains particles with poor membrane signal as a result of the missing wedge. In contrast to soluble (EBP1-bound-) ribosomes, membrane-bound ribosomes were prone to preferred orientations due to the restriction by the ER membrane, which, however, did not affect the refinement of the particles in M.

7. Fig. 1e needs some more labelling as it is quite difficult to understand the graphs. There is no mention of ES27L in the legend, and this figure is referred to sometime before it is introduced in the main text. Also why is the data only shown for the ES27L population of "cytosolic" ribosomes and not all of those that have been determined to be cytosolic? Following on from this the text describes the hibernating ribosomes at the ER membrane as "markedly higher abundance than their cytosolic counterparts", yet from the graph alone this does not feel very convincing and no numbers have been given.

Associated with the improved classification of soluble ribosomes (see above, general 4), we redefined two classes (soluble and unassigned) and renamed the MetAP-ES27L-containing population to "soluble". Statistical analysis has been added. The sentence "markedly higher abundance than their cytosolic counterparts" has been removed (see point 4).

8. Fig. 1 is not consistent in the colouring of A-/P-/E- tRNA between the different states shown.

tRNAs are color-coded according to their positions in respect of a complete elongation cycle. The dashed arrow indicates reiteration of the cycle, resetting the color-code for tRNAs. Added to the figure legend.

9. There is some discrepancy in the determination of the tomogram thicknesses. In the methods section it states that the average thickness is 184 nm, yet the legend for Fig. S1 states that this is 156 nm.

Corrected.

10. The legend for Fig. S1 is not the easiest to follow, and we think that on the figure there is some incorrect labelling and mask 2 is shown to be applied at a stage when it should be mask 1.

Corrected.

11. What size fiducials were used?

10 nm. Added to methods section.

12. Which version of Relion was used? We assume that it is 3.1 to fit in with the Warp/M pipeline, but this needs to be explicitly stated as there are some major differences in subtomogram averaging between Relion versions, especially version 4.

We expanded the methods section. Indeed, it has been Relion version 3.1.1.

13. We feel the term "deep classification" is potentially a bit misleading. This term brings to mind the potential use of deep neural networks for 3D classification, which did not feature in this study.

changed to "extensive".

14. In some figures, the agreement between map and model is less than completely convincing, owing to poor local resolution and probably also to the fact that, as far as we can tell from reading the *Methods > Model building* subsection, the models were obtained by rigid-body fitting pre-existing models, with adjustments made only to the TMHs of TRAP. See for example the right panel of Fig. S5d, where several short helices are not well placed in the density. See similarly problematic helices in Supplementary Fig. S9c. The low local resolution is to be expected, given the large effective pixel sizes used for the classification (3.45 Å for ordinary 3D classification and then 6.9 Å for focused classification) and the heterogeneity of the sample. Comments on why these things do not matter for the conclusions reached would be helpful.

In the revision we refined the atomic models for the Sec61-TRAP-OSTA translocon in Isolde/Coot/phenix and provide a detailed description in the materials and methods. This model will also be submitted to the PDB and resolves any major clashes (see also PDB validation). Models used for assignment of molecular states were not refined due to moderate changes with respect to the fitted, previously published models (e.g., eEF2 in previous Fig. S5d).

Minor points that the authors might find helpful:

15. > its native interactions, including biogenesis cofactors (line 58) to
> its native interactions, including those with biogenesis cofactors

Done.

16. were aligned RELION focused on the LSU (line 308) to
> were aligned in RELION, focused on the LSU

Done.

17. Sec61 can switch from a closed to open conformations (line 49) to
> Sec61 can switch from a closed to an open conformation

This formulation was deliberate since a spectrum of open conformations has been reported in detergent-solubilized isolates in the cited references.

18. In analogy to the cytosolic hibernating ribosome 28, we also detected a second rotated ribosome state (5%), which features eEF2 while CCDC124 is absent.
(line 85) to
> We also detected a second rotated ribosome state (5%), which features eEF2, and from which CCDC124 is absent, analogous to the cytosolic hibernating ribosome. As it stands, it reads as if the detection itself (rather than what was detected) was analogous to the cytosolic hibernating ribosome.

Done.

19. > While OSTA structure and association with ribosome and Sec61 have been studied in detail in isolation, its native interactions, including biogenesis cofactors such as ER chaperones remain elusive.
(line 56) to
> While (the) OSTA's structure and its specific association with the ribosome and Sec61 have been studied extensively, its native interactions, including those with biogenesis cofactors such as ER chaperones, remain elusive.

Done.

20. > characteristic for
(line 150) to
> characteristic of

Done.

21. > allowed building a complete atomic model
(line 166) to
> allowed the building of a complete atomic model
Or, better yet, just don't use a gerund:
> allowed us to build a complete atomic model

Done.

22. Remove "unexplained" from
> unexplained density in the lumen coincides with predicted N-glycosylation sites
(line 185), because you are supplying an explanation ...

Done.

23. feature a lipid-filled distance of 2-3.5 nm between them
(line 190) to
> are separated by 2.0-3.5 nm of lipid density

Done.

24. the glucosyltransferases acting upstream of OSTA or dolichyldiphosphatase I acting
downstream OSTA
(line 212), to
> the glucosyltransferases acting upstream of OSTA or the dolichyldiphosphatase I
acting downstream of OSTA
(that is, add a second 'the' and a second 'of').

Done.

25. neither atomic models provide acceptable fits
(line 213) to
> neither atomic model provides an acceptable fit

Done.

26. L2-3 and L2-4 are poorly resolved concomitant with the flexibility of the RPN2 N-
terminal domain
(line 225) to
> L2-3 and L2-4 are poorly resolved, likely due to the flexibility of the RPN2 N-
terminal domain
The word "concomitant" is so often misused in scientific writing. If in doubt, leave it
out.

Done.

27. Comma-separate the thousands on line 314 (i.e. 134,350) and on line 337 (i.e. 5,554).

Done.

28. D1-3 indicate the domains eEF1a domains 1-3
to
> D1-3 indicate eEF1a domains 1-3

Done.

29. A, P and E indicate ribosomal aminoacyl, peptidyl, and exit site, respectively
(line 392) to
> A, P and E indicate the ribosomal aminoacyl, peptidyl, and exit sites, respectively

Done.

30. Change "domain" to "domains" on line 422.

Done.

31. Replace the hyphen in "false-positives" (line 444) with a space: "false positives".

Done.

32. Not sure why one of the arrows in Fig. 1 is dashed.

See point 8. The dashed arrow indicates a reiterating elongation cycle.

33. Capitalise PDB on line 393.

Done.

34. Inconsistent application of the serial comma (the so-called "Oxford comma", sorry, this is now getting pedantic):
On line 392:
> A, P and E
(serial comma absent)
> aminoacyl, peptidyl, and exit
(serial comma present)

Done (serial comma present).

35. also:
> The arrangement of Sec61, TRAP, OSTA, and T1
(line 208)

Done.

36. Inconsistent hyphenation of "cofactor"
> stoichiometric and sub-stoichiometric *cofactors*
(line 49)
> biogenesis *cofactors* such as ER chaperones
(line 58)
> Sec61 associates with distinct *co-factors*
(line 50)
We suggest no hyphenation.

Done.

37. Use an actual β in line 416 (i.e. β hairpin), maybe with a hyphen (β -hairpin).

Done.

38. Remove the hyphen from "re-positioning" in line 183, i.e. write "repositioning".

Done.

39. pseudosymmetrical (line 190) to > pseudosymmetric.

Done.

Reviewer Reports on the First Revision:

Referees' comments:

Referee #1 (Remarks to the Author):

The revised manuscript “Molecular snapshots of translation and protein translocation at the ER membrane” by Gemmer et al., appears to have taken significant strides towards responding to this reviewer. The most notable efforts pertain to the toning down of conclusions made based on their relatively low-resolution reconstructions relating to 1] the observed ‘PRE+’ state, which they assign as containing eEF1A in an ‘extended’ conformation, not previously observed on the ribosome, and 2] the observed ‘decoding-sampling’ state, which they interpret as a tRNA-selection intermediate preceding codon-anticodon recognition and GTPase activation.

The authors have made substantial efforts to assign protein density within the ‘PRE+’ state to eEF1A, which was challenged during initial review. This includes new proteomics data shown in Fig. S4, in-situ cryo-ET of human cells shown in Fig. S6 where they appear to again see density in the A site of the classical-PRE complex containing three tRNAs, comparative modeling of GTPases present in their sample into the cryo-EM density, and single-particle analysis of isolated ribosomes to achieve a higher resolution structure of this intermediate. With this later effort, the presence of eEF1A could be more confidently assigned based on side-chain densities that appear to match the sequence and structure of the eEF1A2(GDP) model observed crystallographically. Collectively, these additions bring the paper to a higher standard of excellence that all readers in the translation field will appreciate.

However, similarly to how the authors describe eEF2 bound to ‘hibernating’ ribosomes as non-translating ribosome populations directly resulting from cellular stress and isolation artifacts, I would suggest that the authors approach their placement of the two PRE+ complexes within the canonical translation cycle with a bit more humility and skepticism. While the conversion of eEF1A to a GDP-bound conformation on the ribosome would make sense given previous literature, the authors isolation conditions appear to give rise to an abundant pool of both ternary complex and eEF1A, which may simply bind to the ribosomes that are present – like eEF2 to non-functional ribosomes to populate the hibernating class.

Additionally, the author’s interpretation of the observed ‘codon-sampling’ state as arising from bound near- and non-cognate ternary complexes has implications for the authors’ assignment of the PRE+ state and its placement in the elongation cycle. The lifetime of near- and non-cognate ternary complex on the ribosome must be short-lived enough so as to not inhibit translation in the cell. In bacteria, both experimental and theoretical measurements indicate the near- and non-cognate ternary complex lifetime on the ribosome during active protein synthesis to be significantly below 10 ms. For the authors’ assignment of non-cognate ternary complex bound to 22% of the population of ribosomes to be correct, this would require ternary complex to be present in their isolated sample at significant excess over ribosomes.

If ternary complex is indeed that abundant, then translation would proceed during the isolation step, while GTP is being depleted from the cell lysate, giving rise to an abundance of eEF1A(GDP), which

could bind to the ribosomes that are already present as an off-pathway event. This interpretation seems equally plausible to consider as the on-pathway model that is suggested by Fig. 1 and the authors' interpretations. While the on-pathway model is indeed attractive, the authors do show in the mass spec, and their modeling (Fig.S6), that a number of other GTPases are also present in their samples, which also come quite close to fitting the density. I would thus suggest that the authors present the on-pathway model as an attractive interpretation consistent with literature that needs to be further investigated.

Referee #2 (Remarks to the Author):

The direct visualisation of ribosome-translocon complexes *ex vivo* and *in situ* provides a terrific complement to the wealth of earlier biochemical and structural work on reconstituted systems and purified proteins. This work is performed to a high standard as far as I can tell, and getting views of key biological processes such as translation and protein translocation in near-native contexts is important and of wide interest. For these reasons, I remain supportive of publication.

The manuscript continues to over-interpret the data with regard to potential function. I feel this is really not necessary, and often causes more problems than is helpful and risks being wrong or misleading the non-expert reader. My suggestions, focused on the translocon parts of the paper, can be addressed for the most part with some minor changes to the text, and possibly a small amount of added analysis.

Interpretation issues:

1. The authors observe that signal-like densities remain at the lateral gate on hibernating ribosomes, even after a 2-hour DTT treatment. The authors' explanation for this is not plausible. They suggest that these densities are signal peptides of terminated, uncleaved preproteins, and that they remain at the lateral gate because they cannot dissociate until they have been cleaved ("The Sec61-bound SP is likely sterically inaccessible to the signal peptidase complex while the ribosome is bound, which may be required for dissociation," lines 209-211). However, the structure of signal peptidase referenced by the authors shows that the SPase catalytic site cannot approach close enough to the lateral gate for cleavage to occur before dissociation; instead dissociation of the signal from Sec61 must occur prior to cleavage, and such dissociation is well known to occur readily upon sufficient elongation, termination or puromycin release of the nascent chain from the ribosome (under conditions where the ribosome would not dissociate from Sec61). For example, stalled truncations of preprolactin are efficiently cleaved by SPase once the nascent chain elongates past ~140 aa (Mothes et al. 1994), indicating that the signal diffuses away from Sec61 far enough to access SPase when it is not tethered via a short nascent chain to the ribosome. This is also seen in numerous earlier studies releasing the nascent chain from the ribosome with puromycin (e.g., Fig. 1E of PMID 7628015, or Fig. 2 of PMID 7650000). Many cleaved signal peptides then remain in the membrane until they are further processed by signal peptide peptidase (Lemberg & Martoglio 2002). Therefore it remains unknown why there is a signal-like density in ribosome-translocon complexes which have been hibernating on an hour-long timescale, which is surely plenty of time for signal dissociation to

have occurred. The authors should probably be more cautious and simply state that they don't know why there is density there, because the idea of it being signal peptide is probably wrong.

2. Lines 35-36 and lines 228-229: the authors claim that the interactions they modeled between TRAP and Sec61 are what enables TRAP to increase the gating efficiency of weakly gating SPs ("[The model] reveals an intricate interaction network of TRAP with other translocon components enabling its function early in nascent chain processing") and ("the TRAP architecture positions TRAPa to facilitate SP insertion"). They offer no evidence to support the assertion that the TRAP-alpha interactions they have modeled mediate TRAP function, despite privileging this claim with a place in the Abstract. Without such evidence, other mechanistic hypotheses appear equally plausible; for example TRAP's function could be enabled by interactions between Sec61 and any of TRAP's more than 100 unmodeled amino acids. Structural studies generally rely on follow-up experiments like genetic perturbations to test mechanistic hypotheses. Absent such follow-up the authors can only speculate or offer a hypothesis. A well-framed hypothesis can be very useful, but should be clearly presented as such. The manuscript should leave a non-specialist reader with no doubt that TRAP's mechanism remains unknown.

3. If the authors do choose to hypothesize about TRAP's mechanism, they should do so precisely, whereas the current explanation offered appears vague and conflicted. At present, they speculate that "In this position ["near the luminal end of its TMH"] TRAPa may constitute a 'handle' supporting SPs in opening the lateral gate" (lines 225-226). What does this mean? How can a luminal part of TRAP-alpha, which is inaccessible to a cytoplasmic SP, be described as a "handle" for that SP? And what does it mean for TRAP-alpha to "support SPs in opening the lateral gate"? It initially appeared to me that the authors were describing an allosteric model, by which TRAPa's contacts with the Sec61 hinge could be communicated to distal sites (the gate and/or plug) and thereby make gate opening more favourable. The apparent model was that TRAP-alpha and SPs exerted a cooperative effect, which would explain why TRAP alone was insufficient to open the gate. And yet in the rebuttal the authors disclaim the idea "that TRAPa induces a change of the 5/6 loop conformation". But if the modeled TRAP-alpha-Sec61 hinge interaction does not have any effect on that part of Sec61, then how can it have an effect on any other part of Sec61, such as the gate? This contradiction needs to be resolved if a good hypothesis is to be offered.

4. If the authors do want to propose that the TRAPa-Sec61 hinge interaction mediates TRAP function, then the only logically consistent hypothesis I can think of is the one described above: TRAPa cooperates allosterically with the SP to facilitate gating. This hypothesis would be particularly attractive if the conformation of Sec61 differed in any systematic way between the classes with and without TRAP, or between the present TRAP-rich dataset and prior EMDB maps where TRAP had low occupancy. The manuscript as written makes no such comparisons, which seems like a missed opportunity. Regardless, the authors really cannot claim to have deduced what TRAP does or how it works from the structural model, so a more cautious tone is warranted.

5. The authors argue that Sec61 is closed in translocons with MP and TRAP because TRAP's gate-stimulating effect is not strong enough to open the gate without an SP. But the MP also interacts with Sec61, so it seems equally possible that Sec61 remains closed because the MP has a gate-inhibiting effect, instead of TRAP's gate-stimulating effect being weak. All the comparison made

shows is that TRAP does not stimulate gate opening in the presence of MP. The authors' claim would only be directly supported if there were an abundant class of Sec61-TRAP translocons without SPs, and there is not. Care should be taken not to overinterpret the data.

6. Line: 232: "TRAP can dissociate from Sec61 to warrant the biogenesis of multipass proteins." This implies that TRAP's continued presence at multipass translocons would interfere with the biogenesis of multipass proteins. No evidence to support this claim is provided. The data could nonetheless yield useful insights into why TRAP is partially depleted from MP translocons and not others. How abundant are Sec61-TRAP-MP translocons compared to Sec61-MP translocons? (These numbers are omitted from Figure 2A and should be included.) Are Sec61-TRAP-MP translocons more enriched than Sec61-MP translocons in the trailing ribosomes? (If so this would hint at competition between TRAP and MP components.) Are there clashes between the TRAP and MP structures? Does the combination of TRAP and MP in the TRAP-MP class appear to induce any structural changes compared to the TRAP or MP classes?

Presentation issues:

1. Abstract: the Sec61-TRAP-OSTA model "reveals an intricate interaction network of TRAP with other translocon components." This is an overstatement. In the two paragraphs describing TRAP (212-232), only two interactions with other translocon components are described, and both are with the same complex (TRAP-gamma with Sec61-gamma, and TRAP-alpha with Sec61-alpha). A network of two or four interactions does not seem "intricate."

2. Introduction: TRAP's "interaction with Sec61 and the ribosome remain unknown." This misrepresents the literature from even the authors' own work. Pfeiffer et al. 2017 showed that TRAP-gamma's cytoplasmic helical hairpin is "bound to the ribosome via large subunit ribosomal RNA expansion segments (rRNA ES) and ribosomal protein L38," and that the luminal domain is bound "to the ER-luminal loop of the 'hinge' region between the N- and C-terminal halves of Sec61 α ." The text should state that these facts were already known. As a related point, subsequent sections should explain what is new about the present work. The new information about TRAP appears to be an additional contact between TRAP-gamma and Sec61-gamma, the location of the TRAP-alpha TMD, the slightly modified AlphaFold models, and the observation that TRAP is insufficient to stimulate gate opening in the absence of an SP.

3. Figure 3E supports the claim that TRAP-gamma and Sec61-gamma interact by showing that there is continuous density between them at a reasonable threshold. And yet the adjacent panels 3D and 3F do not show any density, despite being designed to support similar claims about other interactions. Densities should be shown for each, particularly given how uncertain interpreting >6 Å maps can be. If the density does not support some interactions as clearly as others, the authors should say so.

4. In the abstract, main text, and figure legends, the atomic model of the Sec61-TRAP-OSTA complex is described as "complete." This is an overstatement. It is true that the model has higher coverage than past models; it is exciting to see all chains represented and all predicted globular domains fitted to density. But that coverage is not complete; there are hundreds of unmodeled amino acids and

unmodeled PTMs. Even if one excludes disordered regions that may never be resolved in a cryo-EM map, there remain unmodeled densities in these maps, such as T1, L1, and L2. It would also seem incorrect to describe the TRAP model as complete, since there remains an unmodeled density adjacent to the cytoplasmic hairpin of TRAP-gamma (visible in Figure S15C and previous maps such as EMD-4315), not to mention >100 unmodeled amino acids. Indeed the authors themselves call the TRAP model merely "near-complete" in the rebuttal, contradicting themselves. This is not to fault the authors' work; completeness is rare in cryo-EM structures, and what matters is whether the maps suffice to answer biological questions.

5. Figure 2D: This schematic would be slightly clearer if the UAG were rotated 180° and placed on the left side of the mRNA instead of the right; as currently drawn it appears backward.

6. Figure 3: the caption incorrectly describes the Sec61-TRAP-OSTA translocon as "the ER translocon." It would be more accurate to retain the term used in the abstract: "the most abundant ER translocon variant."

7. Figure S15A: The authors show monomeric AlphaFold DB models with pLDDTs, but according to the methods these were not used in model building, and instead Colabfold was used to generate multimer predictions. Confidence scores should be shown for those predictions since they were the actual input to model building. Both pLDDT and PAE should be shown, since it is the PAE that shows how confidently the subunits could be placed with respect to one another. The authors should also specify which version of Colabfold and AF2 (single-chain or Multimer) was used.

8. Figure S11F,G,I: Typographical error: "Mulitpass."

Referee #3 (Remarks to the Author):

The authors have done a very good job dealing with our previous comments. Since we are not ribosome or translation experts, the other two reviewers need to comment whether the biological insights gained are sufficiently exciting for their field. From a technical perspective, I am satisfied that this study represents a major advance and showcases what is possible with 1000+ tilt series these days, and what will be possible.

Author Rebuttals to First Revision:

Point-by-point reply

We thank the reviewers for their thoughtful and positive comments. Below we respond to the remaining points and how we have addressed them in the manuscript (in *italic*):

Referee #1:

The revised manuscript “Molecular snapshots of translation and protein translocation at the ER membrane” by Gemmer et al., appears to have taken significant strides towards responding to this reviewer. The most notable efforts pertain to the toning down of conclusions made based on their relatively low-resolution reconstructions relating to 1] the observed ‘PRE+’ state, which they assign as containing eEF1A in an ‘extended’ conformation, not previously observed on the ribosome, and 2] the observed ‘decoding-sampling’ state, which they interpret as a tRNA-selection intermediate preceding codon-anticodon recognition and GTPase activation.

The authors have made substantial efforts to assign protein density within the ‘PRE+’ state to eEF1A, which was challenged during initial review. This includes new proteomics data shown in Fig. S4, in-situ cryo-ET of human cells shown in Fig. S6 where they appear to again see density in the A site of the classical-PRE complex containing three tRNAs, comparative modeling of GTPases present in their sample into the cryo-EM density, and single-particle analysis of isolated ribosomes to achieve a higher resolution structure of this intermediate. With this later effort, the presence of eEF1A could be more confidently assigned based on side-chain densities that appear to match the sequence and structure of the eEF1A2(GDP) model observed crystallographically. Collectively, these additions bring the paper to a higher standard of excellence that all readers in the translation field will appreciate.

1. However, similarly to how the authors describe eEF2 bound to ‘hibernating’ ribosomes as non-translating ribosome populations directly resulting from cellular stress and isolation artifacts, I would suggest that the authors approach their placement of the two PRE+ complexes within the canonical translation cycle with a bit more humility and skepticism. While the conversion of eEF1A to a GDP-bound conformation on the ribosome would make sense given previous literature, the authors isolation conditions appear to give rise to an abundant pool of both ternary complex and eEF1A, which may simply bind to the ribosomes that are present – like eEF2 to non-functional ribosomes to populate the hibernating class.

We agree with the reviewer’s general recommendation to avoid overinterpretation and to clarify hypothetical character and need for additional studies, where appropriate. We adopted the tone in several instances detailed below and conclude the section on the PRE+ structure with the sentence: “This observation is compatible with a possible role of eEF1a in post-hydrolysis proofreading³⁰⁻³² and remains to be further investigated with complementary methods.”

Reviewer #1’s comparison of the PRE+ state to the hibernating eEF2-bound ribosomes indicates that one important piece of evidence strongly supporting the on-pathway interpretation of PRE+ has not come across: the polysome analysis, which has also not been listed in the reviewer’s recap of additional data. We concede that we should have discussed

aspects of the polysome analysis already in the section on ribosome states to clarify the entirely different 'social behavior' of PRE+ ribosomes compared to hibernating eEF2-bound particles, while the previous version of the manuscript covered the entirety of this analysis later.

The analysis of the class of the particles neighboring each ribosome particle shows that the PRE+ classes behave fundamentally differently from the hibernating eEF2-bound particles (Figure S12 D-F in previous manuscript). While the two PRE+ states position adjacent to the other 6 elongating ribosome states, the hibernating eEF2 particles are vastly underrepresented in the neighborhood of elongating particles. Thus, the analysis shows that the PRE+ states occur on polyribosomes together with the other 6 elongation states, while the hibernating eEF2-bound particles have a strong tendency to seclude. In addition, the fact that the PRE+ state is also observed in large amounts *in situ*, in lamellae data, where it is also observed on polysomes, supports the interpretation that this state is not arising from a purification artifact but instead does represent a native conformation of the functional elongating ribosome.

To clarify the position-driven assignment of PRE+ states to the elongation cycle we discuss the neighborhood analysis of the ribosome states early on in the section "Ribosomal intermediate states and their relative spatial distribution".

We refer to point #3 for the incorporation of specific points in the PRE+ section.

2. Additionally, the author's interpretation of the observed 'codon-sampling' state as arising from bound near- and non-cognate ternary complexes has implications for the authors' assignment of the PRE+ state and its placement in the elongation cycle. The lifetime of near- and non-cognate ternary complex on the ribosome must be short-lived enough so as to not inhibit translation in the cell. In bacteria, both experimental and theoretical measurements indicate the near- and non-cognate ternary complex lifetime on the ribosome during active protein synthesis to be significantly below 10 ms.

We acknowledge the point of the reviewer and we have now modified the text to "We propose that these conformations likely correspond to the decoding and PRE+ states [...]" to clarify that the functional context is a model.

We point out that our results are not in disagreement with previous results from kinetics or computational studies. Indeed our technique will always visualize a pool of ribosomes at a given time in the cell (lamella data) and the abundance of the population reflects their relative lifetime with respect to the other populations. The abundance of the decoding step in this data for instance only reflects the fact that it exists for longer in the cell than the other states occurring even faster, or more often during the elongation cycle.

3. If ternary complex is indeed that abundant, then translation would proceed during the isolation step, while GTP is being depleted from the cell lysate, giving rise to an abundance of eEF1A(GDP), which could bind to the ribosomes that are already present as an off-pathway event. This interpretation seems equally plausible to consider as the on-pathway model that is suggested by Fig. 1 and the authors' interpretations. While the on-pathway model is indeed

attractive, the authors do show in the mass spec, and their modeling (Fig.S6), that a number of other GTPases are also present in their samples, which also come quite close to fitting the density. I would thus suggest that the authors present the on-pathway model as an attractive interpretation consistent with literature that needs to be further investigated.

As explained in point #1 we have emphasized that we are suggesting a model in the revision, and we have also incorporated the residual possibility of alteration of PRE+ during purification.

We have now modified Fig. S8 to show in detail how our SPA data does allow us to unambiguously identify eEF1A on the purified ribosome sample and exclude any other protein with similar fold present in the MS data. In addition, we have run FindMySequence neural network (Chojnowski et al, IUCRJ 2022) on the best resolved part of the SPA density (domain 3) probing the entire human proteome. This test also independently identified the protein in this density to be eEF1A with a convincing E-value of 1.3×10^{-10} ($\ll 10^{-7}$ recommended for trustworthy result). However, we concede that we cannot fully exclude that during the purification the factor originally bound at the ribosome GAC in the PRE+ state disassociated from the complex and eEF1a might then have re bound these ribosomes during the quick purification. Therefore, we have now re-worded and nuanced our conclusions as asked by reviewer #1:

"We propose that the classical PRE+ state follows the previously described decoding-like state^{24,27}, where eEF1a still adopts a compact conformation immediately after GTP hydrolysis. While we cannot fully rule out that another factor observed at this site in situ might have been displaced by eEF1A (Fig. S5) during the purification, the occurrence of the classical PRE+ intermediate state indicates the possibility that eEF1a remains bound to the ribosome during conformational switching to the extended form. This observation is compatible with a possible role of eEF1a in post-hydrolysis proofreading²⁸⁻³⁰ and remains to be further investigated with complementary methods."

Referee #2:

4. The authors observe that signal-like densities remain at the lateral gate on hibernating ribosomes, even after a 2-hour DTT treatment. The authors' explanation for this is not plausible. They suggest that these densities are signal peptides of terminated, uncleaved preproteins, and that they remain at the lateral gate because they cannot dissociate until they have been cleaved ("The Sec61-bound SP is likely sterically inaccessible to the signal peptidase complex while the ribosome is bound, which may be required for dissociation," lines 209-211). However, the structure of signal peptidase referenced by the authors shows that the SPase catalytic site cannot approach close enough to the lateral gate for cleavage to occur before dissociation; instead dissociation of the signal from Sec61 must occur prior to cleavage, and such dissociation is well known to occur readily upon sufficient elongation, termination or puromycin release of the nascent chain from the ribosome (under conditions where the ribosome would not dissociate from Sec61). For example, stalled truncations of preprolactin are efficiently cleaved by SPase once the nascent chain elongates past ~140 aa (Mothes et al. 1994), indicating that the signal diffuses away from Sec61 far enough to access SPase when it is not tethered via a short nascent chain to the ribosome. This is also seen in numerous earlier

studies releasing the nascent chain from the ribosome with puromycin (e.g., Fig. 1E of PMID 7628015, or Fig. 2 of PMID 7650000). Many cleaved signal peptides then remain in the membrane until they are further processed by signal peptide peptidase (Lemberg & Martoglio 2002). Therefore it remains unknown why there is a signal-like density in ribosome-translocon complexes which have been hibernating on an hour-long timescale, which is surely plenty of time for signal dissociation to have occurred. The authors should probably be more cautious and simply state that they don't know why there is density there, because the idea of it being signal peptide is probably wrong.

We agree with reviewer #2 that assignment of the helix in the open gate remains to be investigated further and re-phrase the passage as: "Since SPs can be cleaved co-translationally⁴⁴⁻⁴⁶ this helix might correspond to a pool of cleaved SPs or to an unknown specific peptide."

5. Lines 35-36 and lines 228-229: the authors claim that the interactions they modeled between TRAP and Sec61 are what enables TRAP to increase the gating efficiency of weakly gating SPs ("[The model] reveals an intricate interaction network of TRAP with other translocon components enabling its function early in nascent chain processing") and ("the TRAP architecture positions TRAP α to facilitate SP insertion"). They offer no evidence to support the assertion that the TRAP- α interactions they have modeled mediate TRAP function, despite privileging this claim with a place in the Abstract. Without such evidence, other mechanistic hypotheses appear equally plausible; for example TRAP's function could be enabled by interactions between Sec61 and any of TRAP's more than 100 unmodeled amino acids. Structural studies generally rely on follow-up experiments like genetic perturbations to test mechanistic hypotheses. Absent such follow-up the authors can only speculate or offer a hypothesis. A well-framed hypothesis can be very useful, but should be clearly presented as such. The manuscript should leave a non-specialist reader with no doubt that TRAP's mechanism remains unknown.

We agree with emphasizing that our mechanistic interpretation of the TRAP structure is hypothetical. To this end we removed the mechanistic hypothesis of TRAP α from the abstract and emphasized the hypothetical character in the main text. Also reflecting points 6-8, the passage now reads:

"Cellular and biochemical studies indicate that TRAP is required for the biogenesis of proteins that exhibit SPs with weak helical propensity due to glycine and proline residues^{14,47,48}. Preproteins with pronounced hydrophobic helical SPs are subject to stronger pulling forces than TRAP-dependent preproteins of, e.g., insulin or prion protein^{47,48}, presumably due to their lower affinity of their SPs for the lateral gate. While the structure of TRAP-Sec61 does not provide an obvious mechanism of action for TRAP, it allows to formulate a hypothesis. When SPs traverse Sec61 head-on and enter the lumen, they contact the luminal TRAP α domain⁴⁹. The growing nascent chain pushing against the Sec61 hinge-bound TRAP α domain may then open the Sec61 lateral gate via an allosteric mechanism and expose its hydrophobic surface to accommodate the SP. The relevant intermediates would be too short-lived to be captured in our ensemble analysis. Further studies will be required to evaluate the precise mechanistic function of the TRAP interactions revealed in this study."

6. If the authors do choose to hypothesize about TRAP's mechanism, they should do so precisely, whereas the current explanation offered appears vague and conflicted. At present,

they speculate that "In this position ["near the luminal end of its TMH"] TRAPa may constitute a 'handle' supporting SPs in opening the lateral gate" (lines 225-226). What does this mean? How can a luminal part of TRAP-alpha, which is inaccessible to a cytoplasmic SP, be described as a "handle" for that SP? And what does it mean for TRAP-alpha to "support SPs in opening the lateral gate"? It initially appeared to me that the authors were describing an allosteric model, by which TRAPa's contacts with the Sec61 hinge could be communicated to distal sites (the gate and/or plug) and thereby make gate opening more favourable. The apparent model was that TRAP-alpha and SPs exerted a cooperative effect, which would explain why TRAP alone was insufficient to open the gate. And yet in the rebuttal the authors disclaim the idea "that TRAPa induces a change of the 5/6 loop conformation". But if the modeled TRAP-alpha-Sec61 hinge interaction does not have any effect on that part of Sec61, then how can it have an effect on any other part of Sec61, such as the gate? This contradiction needs to be resolved if a good hypothesis is to be offered.

We clarified the hypothesis as outlined in point 5 by pointing out its allosteric nature. We suggest that the allosteric mechanism is activated upon binding of the substrate and hence the resulting conformational state would be transient, which the re-worded hypothesis conveys.

7. If the authors do want to propose that the TRAPa-Sec61 hinge interaction mediates TRAP function, then the only logically consistent hypothesis I can think of is the one described above: TRAPa cooperates allosterically with the SP to facilitate gating. This hypothesis would be particularly attractive if the conformation of Sec61 differed in any systematic way between the classes with and without TRAP, or between the present TRAP-rich dataset and prior EMDB maps where TRAP had low occupancy. The manuscript as written makes no such comparisons, which seems like a missed opportunity. Regardless, the authors really cannot claim to have deduced what TRAP does or how it works from the structural model, so a more cautious tone is warranted.

As outlined in point 5 we have formulated the hypothesis of an allosteric switch more clearly in the revision. We have also compared various available Sec61 structures and do not observe a significant change of the hinge and hence we suggest that a conformational change is transiently induced by the substrate. The hypothetical character of this model is emphasized in the text.

8. The authors argue that Sec61 is closed in translocons with MP and TRAP because TRAP's gate-stimulating effect is not strong enough to open the gate without an SP. But the MP also interacts with Sec61, so it seems equally possible that Sec61 remains closed because the MP has a gate-inhibiting effect, instead of TRAP's gate-stimulating effect being weak. All the comparison made shows is that TRAP does not stimulate gate opening in the presence of MP. The authors' claim would only be directly supported if there were an abundant class of Sec61-TRAP translocons without SPs, and there is not. Care should be taken not to overinterpret the data.

We agree that an inhibitory effect of MP is also a possible scenario. While we kept the finding that TRAP can associate with laterally closed Sec61, we removed the conclusion that TRAP is insufficient to open the Sec61 lateral gate from the manuscript.

9. Line: 232: "TRAP can dissociate from Sec61 to warrant the biogenesis of multipass proteins." This implies that TRAP's continued presence at multipass translocons would interfere with the biogenesis of multipass proteins. No evidence to support this claim is provided. The data could nonetheless yield useful insights into why TRAP is partially depleted from MP translocons and not others. How abundant are Sec61-TRAP-MP translocons compared to Sec61-MP translocons? (These numbers are omitted from Figure 2A and should be included.) Are Sec61-TRAP-MP translocons more enriched than Sec61-MP translocons in the trailing ribosomes? (If so this would hint at competition between TRAP and MP components.) Are there clashes between the TRAP and MP structures? Does the combination of TRAP and MP in the TRAP-MP class appear to induce any structural changes compared to the TRAP or MP classes?

We included the Sec61-TRAP-MP and Sec61-MP abundances in Figure 2A. Moreover, we compared the abundance of TRAP-Sec61-MP vs Sec61-MP in leading and trailing particles. The slight enrichment of Sec61-TRAP-MP translocons (Supplementary Figure S13C) is not statistically significant and hence not covered in the results.

Finally, we do not observe clashes between TRAP and MP, albeit TRAP α is immediately adjacent to MP (McGilvray et al, eLife 2020). Thus, minor changes of TRAP-Sec61-MP arrangement, for example induced by cofactor binding, might indeed result in clashes with TRAP α and induce dissociation, but further data would be required to conclusively address the mechanism of TRAP dissociation and its function. Hence, we now removed this sentence on TRAP dissociation from the manuscript.

Presentation issues:

10. Abstract: the Sec61-TRAP-OSTA model "reveals an intricate interaction network of TRAP with other translocon components." This is an overstatement. In the two paragraphs describing TRAP (212-232), only two interactions with other translocon components are described, and both are with the same complex (TRAP-gamma with Sec61-gamma, and TRAP-alpha with Sec61-alpha). A network of two or four interactions does not seem "intricate."

We rephrased this passage to: "The near-complete atomic structure [...] highlights specific interactions of TRAP with other translocon components [...]"

11. Introduction: TRAP's "interaction with Sec61 and the ribosome remain unknown." This misrepresents the literature from even the authors' own work. Pfeffer et al. 2017 showed that TRAP-gamma's cytoplasmic helical hairpin is "bound to the ribosome via large subunit ribosomal RNA expansion segments (rRNA ES) and ribosomal protein L38," and that the luminal domain is bound "to the ER-luminal loop of the 'hinge' region between the N- and C-terminal halves of Sec61 α ." The text should state that these facts were already known. As a related point, subsequent sections should explain what is new about the present work. The new information about TRAP appears to be an additional contact between TRAP-gamma and Sec61-gamma, the location of the TRAP-alpha TMD, the slightly modified AlphaFold models, and the observation that TRAP is insufficient to stimulate gate opening in the absence of an SP.

We changed the introductory sentence to “Low-resolution studies revealed interactions of TRAP with ribosome rRNA expansion segments and ribosomal subunit protein 38e (RPL38e)¹⁶, but molecular details lack in the absence of an atomic structure of TRAP.” In the revision we also put more emphasis on the novel findings.

12. Figure 3E supports the claim that TRAP-gamma and Sec61-gamma interact by showing that there is continuous density between them at a reasonable threshold. And yet the adjacent panels 3D and 3F do not show any density, despite being designed to support similar claims about other interactions. Densities should be shown for each, particularly given how uncertain interpreting >6 Å maps can be. If the density does not support some interactions as clearly as others, the authors should say so.

We adjusted the close-ups, all of which now depict the molecular models placed into the experimentally determined density maps.

13. In the abstract, main text, and figure legends, the atomic model of the Sec61-TRAP-OSTA complex is described as "complete." This is an overstatement. It is true that the model has higher coverage than past models; it is exciting to see all chains represented and all predicted globular domains fitted to density. But that coverage is not complete; there are hundreds of unmodeled amino acids and unmodeled PTMs. Even if one excludes disordered regions that may never be resolved in a cryo-EM map, there remain unmodeled densities in these maps, such as T1, L1, and L2. It would also seem incorrect to describe the TRAP model as complete, since there remains an unmodeled density adjacent to the cytoplasmic hairpin of TRAP-gamma (visible in Figure S15C and previous maps such as EMD-4315), not to mention >100 unmodeled amino acids. Indeed the authors themselves call the TRAP model merely "near-complete" in the rebuttal, contradicting themselves. This is not to fault the authors' work; completeness is rare in cryo-EM structures, and what matters is whether the maps suffice to answer biological questions.

The term ‘complete’ has been removed or replaced by ‘near-complete’.

14. Figure 2D: This schematic would be slightly clearer if the UAG were rotated 180° and placed on the left side of the mRNA instead of the right; as currently drawn it appears backward.

Done

15. Figure 3: the caption incorrectly describes the Sec61-TRAP-OSTA translocon as "the ER translocon." It would be more accurate to retain the term used in the abstract: "the most abundant ER translocon variant."

Done

16. Figure S15A: The authors show monomeric AlphaFold DB models with pLDDTs, but according to the methods these were not used in model building, and instead Colabfold was used to generate multimer predictions. Confidence scores should be shown for those predictions since they were the actual input to model building. Both pLDDT and PAE should

be shown, since it is the PAE that shows how confidently the subunits could be placed with respect to one another. The authors should also specify which version of Colabfold and AF2 (single-chain or Multimer) was used.

We added the prediction model from Colabfold that was used as input for refinements, as well as plots showing sequence coverage, the pLDDT values per residue position and PAE. We also denoted versions used.

17. Figure S11F,G,I: Typographical error: "Muiltipass."

Done

Reviewer Reports on the Second Revision:

Referees' comments:

Referee #1 (Remarks to the Author):

The manuscript appears to be evolving in an overall positive direction. I attempt to summarize the concrete advances of this body of work.

Others, including this group, have published previously on microsomes using cryo-ET methods. The results obtained are globally congruent with what has been shown previously. However, the authors report less sample degradation. The new technologies utilized also afford more systematic analyses and higher resolution results (ca. 4-10 Angstrom- a table should be included alongside Fig. S3 to indicate each complexes' resolution).

Others have also reported conformational changes in eEF1A on the ribosome during active translation after GTP hydrolysis - see Berhmann et al. Cell 2015 for instance- albeit at lower resolution.

The presented advances therefore center on the benefits of faster lysis techniques for the same sample preparations and the application of direct detector technology and image processing strategies and the eEF1A intermediates they observe using cryoET methods, unlike the single-particle analysis (SPA) methods used by others.

The findings therefore reflect a systematic approach towards advancing the cryoET approach, important steps forward in this fast-emerging area. The additional experimental information provided during the review process has strengthened the conclusion that factor-bound intermediates exist and may be part of the translation elongation cycle, but issues remain in this regard.

The authors focus their attention in this work on what they argue to be two new insights from their study regarding the translation mechanism. In the first half of the paper, they focus on what they refer to as a "comprehensive" overview of the translation elongation cycle including specific intermediates that have not been evidenced previously. The second half of the paper focuses on co-translational transport across the ER membranes. My current read is that the authors' molecular insights in both areas remain speculative.

Regarding the translation mechanism, which I am more familiar with, I am unconvinced that it is reasonable to state that the complexes they arrange into an elongation cycle is "comprehensive" in nature. While the new SPA cryoEM data solidify identification of the factor as eEF1A, for reasons that I elaborate below, I remain skeptical of the conclusions surrounding the placement and relevance of the factor-bound intermediates they refer to as 1] near-cognate ternary complexes engaged with ribosomes (decoding-sampling) and 2] eEF1(GDP)-bound complexes (PRE+ and Rotated-1 PRE+).

Based on previous literature from multiple groups in bacteria and the work reported by Berhmann et al. Cell 2015 in human referenced above, the concept that eEF1A adopts undergoes conformation changes on the ribosome germane to the proofreading process seems quite likely. What makes little biological sense is why eEF1A would stay bound to tRNA and the ribosome for so long after peptide bond formation. While the review has steered clear of more problematic claims, further clarification on why the authors see something different than Berhmann et al is needed. The specific distinctions of the eEF1A conformational changes they see compared to that of Berhmann et al. Cell 2015 also seems relevant to include.

Summary of key issues:

1] In describing the elongation cycle in Figure 1a, “comprehensive” (as it is referred to in the abstract) is an overstatement as there are known intermediates along the pathway that are not observed and there are surely intermediates that exist that have yet to be discovered. For instance, there is no complex defined with eEF1 in GTP-bound, activated state and there are no eEF2-bound translocation intermediates. The cycle shown is thus partially comprehensive compared to what has already been discerned by others.

2] Upon reviewing the literature and the methods section of this paper more carefully, I restate again some need for caution regarding the mechanistic relevance of the factor-bound intermediates identified -- defined as Decoding-sampling, PRE+ and Rotated-1 PRE+. These complexes may be an artifact of the isolation methods/imaging conditions. This concern is rationalized based on the following considerations.

The estimated cellular rates of translation based on biochemical studies and live-cell measurements by the Vale, Tanenbaum and Stasevich groups conclude that translation elongation occurs at a rate of 3-5 per second. How then do the authors manage to isolate factor-bound complexes after massively diluting cell extracts for an hour or more, followed by pelleting of their microsomes/ribosomes out of those conditions over the course of 10s of minutes. Something is occurring that dramatically stabilizes the observed factor-bound complexes or to unusually populate factor-bound complexes. On line 134, the authors state that 66% of their ribosomes are factor-bound whereas the previous study by the Spahn group (Behrmann Cell 2015) reports just 8% factor bound.

First principles stipulate that decoding-sampling complexes are functionally defined as exhibiting highly transient lifetimes – on the order of 1-50 milliseconds. The transient nature of this state is required for rapid turnover during the selection process. Given the large excess of near and non-cognate tRNAs in the cell for each codon, longer residence times would be strongly inhibitory to translation elongation. While I accept the authors interpretation that they see a mixture of states in their cellular tomography studies that look to be similar to what they call decoding-sampling complexes (at ca. 12 Angstrom with 2300 particles), it’s quite difficult to say much more. In diluted extracts, it is difficult to understand how near- and non-cognate complexes remain fixed after dilution.

Similar considerations can be deduced for post-hydrolysis eEF1A intermediates. eEF1A should only be bound for a maximum of ca. 200-300 ms (3-5 per second /estimated translation rate).

Could something be favoring factor binding to ribosomes in the experiments performed? Does

translation continue in their prepared extracts after rapid and significant dilution, in the presence of no added nucleotide, no energy regeneration systems, no cellular polyamines and only 10 mM salt? Cellular conditions are roughly 200 mM monovalent ions, dominated by potassium, and contain polyamines. The methods section states that microsomes and ribosomes were isolated in 10 mM Hepes pH 7.4, 1 mM MgCl₂, 0.5 mM DTT and 250 mM sucrose. The previous report by Spahn and co-workers reports using 25 mM Hepes pH 7.2, 110 mM KOAc, 2.5 mM Mg(OAc)₂, 1 mM EGTA, 1 mM DTT plus protease and RNAase inhibitors.

Ionic strength considerations are important because eEF1A, like other eukaryotic translation factors, is a basic protein (pI ~9) and tRNA and the ribosome are of course both highly negatively charged. In the absence of salt and GTP, abundant translation factors should therefore fix themselves to the ribosome. Whether low salt conditions prevent dissociation or promote binding after dissociation remains to be determined.

3] While supportive that factor-bound intermediates also exist in the cell, the in-cell tomographic studies (Figure S5) are at 16 Angstrom resolution and contain just 1950 particles. If we agree that eEF1A's assignment was non-trivial at 4-10 Angstrom, and thus required SPA analyses to be certain, we can agree that it is more speculative at 16 Angstrom to call the factor that is bound to the ribosome in the cell eEF1A. All that can be concluded is that translating ribosomes in PRE-state conformations in cells also have something bound to them. It may or may or may not be eEF1A.

While it may be the case that the snapshots obtained represent what the authors claim them to be - bona fide translation elongation intermediates, given the aforementioned considerations I will continue advocate for more caution regarding their physiological relevance of their findings as they related to eEF1A and question their placement of the observed intermediates on the elongation pathway, particularly the rotated, PRE+ class. The authors' new single-particle analyses indeed support the interpretation that the factor is eEF1A bound to the A site of the ribosome in an extended conformation. It remains to be determined, however, whether and how it is relevant to the translation elongation cycle, and why they see something different than Behrmann et al.. Particular concern relates to the rotated PRE+ conformation, where eEF1A is suggested to remain bound to the ribosome after peptide bond formation has occurred (therefore an inhibited state of translation, however temporary it may be).

It should therefore be made more clear to the reader that the cycle presented in figure 1 is speculative rather than comprehensive. This could be done by changing their language in the abstract and main text and revising the Figure 1 title to make sure this impression is made. Absent these changes, the general reader, and perhaps even translation-focused scientists may think the findings to be truth rather than observations.

It should also be more openly stated that the authors' presented order of events in translation elongation is the logical progression given what has been defined previously in the literature through in fully reconstituted systems and analogous investigations by others.

Referee #2 (Remarks to the Author):

The changes have improved the paper. I would urge the authors to carefully consider their functional speculations in preparing the final manuscript - generally better to be a bit reserved than be proven wrong later.

Author Rebuttals to Second Revision:

Point-by-point reply

We thank reviewers for thoroughly going through our manuscript and their helpful comments. Reviewer #1 has some remaining concerns and we are grateful to the detailed explanation and constructive suggestions to incorporate those into the manuscript. In essence, reviewer #1 emphasizes that isolation conditions may impact some of the states detected as well as their abundance, and that it must be avoided that readers are left with the impression that figure 1A comprehensively captures *the* elongation cycle. Reviewer #1's suggestion is further clarifying the speculative nature of the suggested cycle. Reviewer #2 recommends further caution with functional speculation. We agree with both suggestions and summarize the main changes to the manuscript:

- We removed 'comprehensively' from the abstract to avoid misconceptions.
- We rearranged the manuscript such that we first summarize the experimental findings on ribosome intermediates and their relative spatial distributions, including possible limitations due to preparation, prior to interpreting them in the context of a model for the elongation cycle and available structural and biochemical data. Thus, the 'on-pathway' hypothesis becomes clearer as reviewer #1 previously encouraged in the context of the PRE+ state.
- We re-labeled previous Figure 1A (now 1D) as 'Observed active intermediates positioned in model of human elongation cycle' to clarify its speculative aspects and added Figure S7 to position the detected states in a model comprising further states not detected in our study.
- We moved the low-abundance eEF1a-bound Rot-1+ state from the model of the elongation cycle to the supplement as it is indeed difficult to reconcile with peptide bond formation and the lack of support through in situ data at this point.
- We further changed the wording at several instances to unequivocally distinguish results and hypotheses.
- We cited the work on the multipass-translocon by the Hegde and Keenan labs, and the work on bacterial ribosome populations in situ by the Mahamid lab, all 3 recently published in Nature.

We address the points in detail below with our replies in blue italic.

Referee #1:

The manuscript appears to be evolving in an overall positive direction. I attempt to summarize the concrete advances of this body of work.

Others, including this group, have published previously on microsomes using cryo-ET methods. The results obtained are globally congruent with what has been shown previously. However, the authors report less sample degradation. The new technologies utilized also afford more systematic analyses and higher resolution results (ca. 4-10 Angstrom- a table should be included alongside Fig. S3 to indicate each complexes' resolution).

The table is provided in the Nature template, which may not have been accessible to the reviewer. We now include it as 'supplementary material' as well.

Others have also reported conformational changes in eEF1A on the ribosome during active translation after GTP hydrolysis - see Berhmann et al. Cell 2015 for instance- albeit at lower resolution.

The presented advances therefore center on the benefits of faster lysis techniques for the same sample preparations and the application of direct detector technology and image processing strategies and the eEF1A intermediates they observe using cryoET methods, unlike the single-particle analysis (SPA) methods used by others.

The findings therefore reflect a systematic approach towards advancing the cryoET approach, important steps forward in this fast-emerging area. The additional experimental information provided during the review process has strengthened the conclusion that factor-bound intermediates exist and may be part of the translation elongation cycle, but issues remain in this regard.

The authors focus their attention in this work on what they argue to be two new insights from their study regarding the translation mechanism. In the first half of the paper, they focus on what they refer to as a “comprehensive” overview of the translation elongation cycle including specific intermediates that have not been evidenced previously. The second half of the paper focuses on co-translational transport across the ER membranes. My current read is that the authors’ molecular insights in both areas remain speculative.

Regarding the translation mechanism, which I am more familiar with, I am unconvinced that it is reasonable to state that the complexes they arrange into an elongation cycle is “comprehensive” in nature. While the new SPA cryoEM data solidify identification of the factor as eEF1A, for reasons that I elaborate below, I remain skeptical of the conclusions surrounding the placement and relevance of the factor-bound intermediates they refer to as 1] near-cognate ternary complexes engaged with ribosomes (decoding-sampling) and 2] eEF1(GDP)-bound complexes (PRE+ and Rotated-1 PRE+).

Based on previous literature from multiple groups in bacteria and the work reported by Berhmann et al. Cell 2015 in human referenced above, the concept that eEF1A adopts undergoes conformation changes on the ribosome germane to the proofreading process seems quite likely. What makes little biological sense is why eEF1A would stay bound to tRNA and the ribosome for so long after peptide bond formation. While the review has steered clear of more problematic claims, further clarification on why the authors see something different than Berhmann et al is needed. The specific distinctions of the eEF1A conformational changes they see compared to that of Berhmann et al. Cell 2015 also seems relevant to include.

Summary of key issues:

1] In describing the elongation cycle in Figure 1a, “comprehensive” (as it is referred to in the abstract) is an overstatement as there are known intermediates along the pathway that are not observed and there are surely intermediates that exist that have yet to be discovered. For instance, there is no complex defined with eEF1 in GTP-bound, activated state and there are no eEF2-bound translocation intermediates. The cycle shown is thus partially comprehensive compared to what has already been discerned by others.

We agree with the reviewer and removed the term ‘comprehensively’ from the abstract.

Furthermore, we restructured the text to better separate the discussion of the specific detected intermediates and to clarify that these do not cover the entire elongation cycle. To emphasize the

speculative character and the role of prior work in the interpretation we start the paragraph with: “To further analyze the polysome-associated ribosomal classes, we attempted to position them in the context of the elongation cycle as modelled based on knowledge from previous in vitro reconstitution work”, and further “7 states are consistent with prior structural or biochemical data”.

Figure 1D (previously Figure 1A) is now also named more clearly: “Observed active intermediates positioned in model of human elongation cycle”. Overall, we avoided to use any element of language that may lead a reader to think that we claim to recapitulate all intermediates of a full elongation cycle. In addition, we add Figure S7, where we place our detected ribosome states in the context of the elongation cycle proposed by Behrmann et al.

2] Upon reviewing the literature and the methods section of this paper more carefully, I restate again some need for caution regarding the mechanistic relevance of the factor-bound intermediates identified -- defined as Decoding-sampling, PRE+ and Rotated-1 PRE+. These complexes may be an artifact of the isolation methods/imaging conditions. This concern is rationalized based on the following considerations.

The estimated cellular rates of translation based on biochemical studies and live-cell measurements by the Vale, Tanenbaum and Stasevich groups conclude that translation elongation occurs at a rate of 3-5 per second. How then do the authors manage to isolate factor-bound complexes after massively diluting cell extracts for an hour or more, followed by pelleting of their microsomes/ribosomes out of those conditions over the course of 10s of minutes. Something is occurring that dramatically stabilizes the observed factor-bound complexes or to unusually populate factor-bound complexes. On line 134, the authors state that 66% of their ribosomes are factor-bound whereas the previous study by the Spahn group (Behrmann Cell 2015) reports just 8% factor bound.

First principles stipulate that decoding-sampling complexes are functionally defined as exhibiting highly transient lifetimes – on the order of 1-50 milliseconds. The transient nature of this state is required for rapid turnover during the selection process. Given the large excess of near and non-cognate tRNAs in the cell for each codon, longer residence times would be strongly inhibitory to translation elongation. While I accept the authors interpretation that they see a mixture of states in their cellular tomography studies that look to be similar to what they call decoding-sampling complexes (at ca. 12 Angstrom with 2300 particles), it's quite difficult to say much more. In diluted extracts, it is difficult to understand how near- and non-cognate complexes remain fixed after dilution.

Similar considerations can be deduced for post-hydrolysis eEF1A intermediates. eEF1A should only be bound for a maximum of ca. 200-300 ms (3-5 per second /estimated translation rate).

Could something be favoring factor binding to ribosomes in the experiments performed? Does translation continue in their prepared extracts after rapid and significant dilution, in the presence of no added nucleotide, no energy regeneration systems, no cellular polyamines and only 10 mM salt?

Cellular conditions are roughly 200 mM monovalent ions, dominated by potassium, and contain polyamines. The methods section states that microsomes and ribosomes were isolated in 10 mM Hepes pH 7.4, 1 mM MgCl₂, 0.5 mM DTT and 250 mM sucrose. The previous report by Spahn and co-workers reports using 25 mM Hepes pH 7.2, 110 mM KOAc, 2.5 mM Mg(OAc)₂, 1 mM EGTA, 1 mM DTT plus protease and RNAase inhibitors.

Ionic strength considerations are important because eEF1A, like other eukaryotic translation factors, is a basic protein (pI ~9) and tRNA and the ribosome are of course both highly negatively charged. In the absence of salt and GTP, abundant translation factors should therefore fix themselves to the ribosome. Whether low salt conditions prevent dissociation or promote binding after dissociation remains to be determined.

We agree that our isolation protocol, like any preparation involving cell lysis, may introduce some alteration of the ribosomal populations. As the reviewer points out, it is difficult to predict the effect of specific purification conditions on ribosomal populations. Therefore, we attempted to compare our populations to a small in situ sample, which supports that the high amount of factor bound population is in principle relevant and not due to a major artifact in purification (as in situ ~70% of the ribosomes are factor bound and ~66% in our isolation data). The correlation of the major intermediates detected in situ and ex vivo do not indicate drastic changes of factor binding either. Nevertheless, we cannot exclude the possibility of some degree of post lysis exchange of the factors, hence we now clearly indicated in the manuscript the possible limitations of our observations due to isolation conditions with the sentence: "Nevertheless, we stress that lysis and the isolation conditions may affect intermediate structures and their abundance, which may eventually be overcome when higher resolution is achievable for cryo-FIB/ET studies of human cells".

Until a large in situ dataset at side-chain resolution can be analyzed, there is a value in integrating data from different isolation protocols as each may better preserve different ribosomal populations, as exemplified by the complementarity between our work and Behrmann et al. 2015. Furthermore, isolates offer the possibility of correlation with orthogonal analytical approaches such as mass-spectrometry analysis, as we did in this study.

We thank the reviewer for the comment on the decoding state. In the re-organized manuscript, we describe this state as 'an unrotated state, with clear densities for the tRNAs in the P- and E-sites and the eEF1a-tRNA ternary complex'. Therefore, we assigned it to a decoding population. We note that comparison with previously determined decoding EM maps indicates that the eEF1 position in our map is closest to EMD-2623 (Budkevich et al. 2014), where eEF1A GTP hydrolysis was inhibited, rather than to EMD-2908 (Behrmann et al. 2015), where eEF1A was presumably observed post hydrolysis (see FigS7A,B). The interpretation of this population possibly being testing non cognate tRNAs is now clearly presented as a speculation.

Regarding the relative abundance of decoding complexes we note here that, although a maximum lifetime of 200-300ms is short per se, this lifetime should be considered relatively to that of other elongation intermediates because purification (or in situ FIB data) will provide a snapshot of an entire ribosomal population at one given timepoint. The complexes having a longer lifetime relatively to others shall be over-represented. Complexes with slightly shorter lifetime but occurring very frequently may also be observed. Proofreading is a known rate limiting step of elongation and therefore corresponding complexes (decoding and PRE, including eEF1A bound complexes) will have a lifetime that is longer when compared to other elongation intermediates.

We also note that the assigned positions of the 3 most abundant states we observe in purified sample as well as in situ data are consistent with the known elongation rate-limiting steps: decoding and PRE+ correspond to the 2 steps of proof-reading, while rotated-2 precedes translocation. Therefore, we think that our data appears in reasonable agreement with respect to the previously established kinetics of

elongation. Similarly, a recent in situ structural study performed in bacteria also identified a decoding state and an A,P state (corresponding to classical PRE without E site) as the most abundant ones, corresponding to the proofreading steps (Xue et al, Nature 2022). In this case however, the A,P state is not bound by any factor – which is different from our data on human PRE+ (both purified and in situ) and suggests differences at this second proofreading step in eukaryotes compared to what is established in bacteria.

Regarding the comparison of the specific isolation protocols used by Behrmann et al and our study we agree that ionic strength is an important parameter that may result in differences of ribosome intermediates, but we think that the size exclusion step introduced in Behrmann et al. may also have contributed substantially to loss of factor bound complexes. Compared to the in situ factor binding (>70%) we see a similar amount of factor binding in our isolation study (66%). If we assume that the pre-lysis factor binding has been similar in the study by (Behrmann Cell 2015) also performed in HEK cells, their isolation protocol, seem to have resulted in a notably larger loss of factor bound states. Behrmann et al introduced this size exclusion step to meet the requirements of sample homogeneity for cryo-EM single particle analysis, ,while the use of cryo-ET allowed us to avoid this extra purification step. Therefore, we think that both approaches and results are complementary to increase our insights on the human ribosomal elongation cycle.

3] While supportive that factor-bound intermediates also exist in the cell, the in-cell tomographic studies (Figure S5) are at 16 Angstrom resolution and contain just 1950 particles. If we agree that eEF1A's assignment was non-trivial at 4-10 Angstrom, and thus required SPA analyses to be certain, we can agree that it is more speculative at 16 Angstrom to call the factor that is bound to the ribosome in the cell eEF1A. All that can be concluded is that translating ribosomes in PRE-state conformations in cells also have something bound to them. It may or may or may not be eEF1A.

We agree with this comment and we have acknowledged this in the previous manuscript version: “While we cannot rule out that another factor observed at this site in situ might have been displaced by eEF1A (Fig. S6) during the purification, the occurrence of the classical PRE+ intermediate state indicates the possibility that eEF1a remains bound to the ribosome during conformational switching to the extended form.” and later “remains to be further investigated with other methods”. As the first sentence appeared misleading we now further changed it to “While we cannot rule out that other factors observed at this site in situ could have been displaced by eEF1A (Fig. S6) during the purification, the occurrence of the eEF1A bound classical PRE+ state in purified samples indicates the possibility that eEF1a may remain bound to the ribosome during conformational switching to the extended form.”

Our reasoning is that, in situ, a factor (or different factors) is bound to the major classical PRE state observed. This was not observed before and it also does not occur in situ in bacteria (Xue et al. Nature 2022). The identity of the factor or factors bound to the PRE state in human indeed cannot be determined at the resolution achieved in our in situ data. In our purified sample, SPA identified eEF1A in an extended conformation at this place. Our observation of extended eEF1A bound to classical PRE complex (and notably the arrangement visualized in SPA) shows that in principle the extended conformation of human eEF1A does not result in clashes with the ribosome that would exclude the fact that it can remain bound. Therefore, our data indicates that extended eEF1A may be the factor, or one of the factors, bound to the PRE+ complex visualized in situ. Hence, altogether our data “indicates the possibility that eEF1A may remain bound to the ribosome during conformational switching to the extended form”, and “the functional

relevance of a possible eEF1A-bound classical PRE+ state remains to be further investigated with complementary methods.”

While it may be the case that the snapshots obtained represent what the authors claim them to be -bona fide translation elongation intermediates, given the aforementioned considerations I will continue advocate for more caution regarding their physiological relevance of their findings as they related to eEF1A and question their placement of the observed intermediates on the elongation pathway, particularly the rotated, PRE+ class. The authors’ new single-particle analyses indeed support the interpretation that the factor is eEF1A bound to the A site of the ribosome in an extended conformation. It remains to be determined, however, whether and how it is relevant to the translation elongation cycle, and why they see something different that Behrmann et al.. Particular concern relates to the rotated PRE+ conformation, where eEF1A is suggested to remain bound to the ribosome after peptide bond formation has occurred (therefore an inhibited state of translation, however temporary it may be).

It should therefore be made more clear to the reader that the cycle presented in figure 1 is speculative rather than comprehensive. This could be done by changing their language in the abstract and main text and revising the Figure 1 title to make sure this impression is made. Absent these changes, the general reader, and perhaps even translation-focused scientists may think the findings to be truth rather than observations.

It should also be more openly stated that the authors’ presented order of events in translation elongation is the logical progression given what has been defined previously in the literature through in fully reconstituted systems and analogous investigations by others.

We agree with the reviewer that at this point we have no in situ data supporting the physiological relevance of the low abundance rotated-1 PRE+ conformation, which may then also be the result of post lysis re-association of extended eEF1 to ribosomes present on polysomes. Therefore, we have now removed this state from our assignment to the model of the elongation cycle.

We regret that the presentation of our observed states has been misleading. We have now reorganized the manuscript to follow a more logical path, which in short is: we observed 10 different classes. 8 of them are present on polysomes, which is in principle indicative of elongation activity (while the 2 others are hibernating). Comparison with in situ data indicates good correlation with isolates, but limits of isolation approach do apply. Among the 8 states observed on polysomes, we do not interpret the elongation intermediate (rot1 PRE+) as relevant. For the 7 other states observed on polysomes, we compared our maps with previously described ribosomal states and interpreted their possible succession in a model of the human elongation cycle.

As mentioned above, we removed the word “comprehensively” from the abstract, we modified the title of Figure 1D (previously 1A) to: “Observed active intermediates positioned in model of human elongation cycle” and adjusted the manuscript order and text in this direction.

Referee #2:

The changes have improved the paper. I would urge the authors to carefully consider their functional speculations in preparing the final manuscript - generally better to be a bit reserved than be proven wrong later.

We agree with reviewer #2, and we reorganized the manuscript to focus more on the exact conclusions drawn from the data. We strictly limited functional speculation and clearly indicated it as such. Regarding the previously discussed hypothesis of TRAP mechanism we note that we additionally mention an alternative hypothesis from a recent biorxiv manuscript of the Paavilainen lab (Karki et al), which puts membrane reshaping of the TRAP complex forward as a mechanism for TRAP function.

Author Rebuttals to Third Revision:

Referees' comments:

Referee #1 (Remarks to the Author):

The authors efforts to further revise their manuscript have again led to substantial improvements in logical flow and readability. Prior efforts are also more appropriately acknowledged; more speculative claims have been tempered or removed, leaving the major advancements to be more easily understood and consumed.

I have no further recommendations at this time.